# Higgs-Confinement Transitions in QCD from Symmetry Protected Topological Phases

Thomas T. Dumitrescu and Po-Shen Hsin

*Mani L. Bhaumik Institute for Theoretical Physics,*
*Department of Physics and Astronomy,*
*University of California, Los Angeles, CA 90095, USA*

In gauge theories with fundamental matter there is typically no sharp way to distinguish confining and Higgs regimes, e.g. using generalized global symmetries acting on loop order parameters. It is standard lore that these two regimes are continuously connected, as has been explicitly demonstrated in certain lattice and continuum models. We point out that Higgsing and confinement sometimes lead to distinct symmetry protected topological (SPT) phases – necessarily separated by a phase transition – for ordinary global symmetries. We present explicit examples in 3+1 dimensions, obtained by adding elementary Higgs fields and Yukawa couplings to QCD while preserving parity P and time reversal T. In a suitable scheme, the confining phases of these theories are trivial SPTs, while their Higgs phases are characterized by non-trivial P- and T-invariant theta-angles $\theta_f, \theta_g = \pi$ for flavor or gravity background gauge fields, i.e. they are topological insulators or superconductors. Finally, we consider conventional three-flavor QCD (without elementary Higgs fields) at finite $U(1)_B$ baryon-number chemical potential $\mu_B$, which preserves P and T. At very large $\mu_B$, three-flavor QCD is known to be a completely Higgsed color superconductor that also spontaneously breaks $U(1)_B$. We argue that this high-density phase is in fact a gapless SPT, with a gravitational theta-angle $\theta_g = \pi$ that safely co-exists with the $U(1)_B$ Nambu-Goldstone boson. We explain why this SPT motivates unexpected transitions in the QCD phase diagram, as well as anomalous surface modes at the boundary of quark-matter cores inside neutron stars.

December 28, 2023

# 1 Introduction and Summary

## 1.1 Gauge Theory Phases and Higgs-Confinement Continuity

Phases of gauge theories play an important role throughout physics. In the standard model of particle physics, the $SU(3)_c$ color gauge fields of QCD are responsible for the confinement of quarks into color-neutral hadrons. By contrast, the Standard Model Higgs field spontaneously breaks – or Higgses – the electroweak $SU(2) \times U(1)$ gauge symmetry down to the diagonal $U(1)_{\text{e.m.}}$ gauge group of electromagnetism,[1] which realizes a Coulomb phase. In condensed matter physics, BCS superconductors are described by a composite Higgs field of electric charge two (describing Cooper paired electrons) that Higgses the $U(1)_{\text{e.m.}}$ electromagnetic gauge group down to its $\mathbb{Z}_2$ subgroup.[2]

---

[1] See for instance [1] for a textbook discussion.

[2] See for instance [2] and references therein.

It has long been appreciated [3–5] that gauge theory phases in 3+1 dimensions can be characterized using expectation values of large electric and magnetic loop operators, known as Wilson and 't Hooft loops. For instance, the confining phase of pure $SU(3)_c$ gauge theory is characterized by the rapid exponential decay of large fundamental Wilson loops, known as the area law,[3] while the Higgs phase of $U(1)_{\text{e.m.}}$ is characterized by an area law for 't Hooft loops.[4] In both cases the area law signals the presence of finite-tension strings: electric confining strings and magnetic vortex strings, respectively.

More recently, the notion of generalized global symmetries [6] (reviewed in [7,8]), which act on extended defects, operators, and excitations, has made the understanding of gauge theory phases based on large loop order parameters fully compatible with the Landau paradigm, wherein phases are characterized by broken and unbroken global symmetries. The generalized symmetries most relevant to gauge theories in 3+1d are one-form symmetries, which act on one-dimensional loop operators and dynamical strings.[5] For instance, pure $SU(3)_c$ gauge theory has a $\mathbb{Z}_3$ one-form symmetry associated with the center of the gauge group. It can be thought of as a fully covariant version of the $\mathbb{Z}_3$ center symmetry that arises when one considers the theory on a circle, or equivalently at finite temperature $T$. This symmetry acts on Wilson loops and it is unbroken in the confining phase at low $T$ (where it characterizes the confining strings), but spontaneously broken in the deconfined high-$T$ phase [5, 15, 6]. Similarly, any $U(1)_{\text{e.m.}}$ gauge theory with only electrically charged matter fields has a magnetic one-form symmetry that detects the magnetic flux of 't Hooft loops. This symmetry is unbroken in the Higgs phase; it is spontaneously broken in the Coulomb phase, with the massless photon furnishing the requisite Nambu-Goldstone boson (NGB) [6].

The existence of one-form symmetries is predicated on the absence of certain physically allowed electrically or magnetically charged particles. For instance, the $\mathbb{Z}_3$ one-form symmetry of pure $SU(3)_c$ gauge theory is explicitly broken by dynamical matter in the fundamental representation, which can end confining strings and screen Wilson loops. Similarly, the magnetic one-form symmetry in $U(1)_{\text{e.m.}}$ gauge theory is explicitly broken if there are dynamical

---

[3] In more detail, the area law states that a loop operator supported along a suitably large curve $C$ decays as $e^{-\sigma A(C)}$, with $A(C)$ the minimal area enclosed by $C$. The constant $\sigma$ is the string tension.

[4] By contrast, the Coulomb phase of $U(1)_{\text{e.m.}}$ is characterized by a perimeter law $e^{-MP(C)}$ for large Wilson and 't Hooft loops supported on the curve $C$. Here $P(C)$ is the perimeter of the curve, and $M$ is some scheme-dependent mass scale, which can be set to any value (including $M = 0$) by a suitable counterterm.

[5] If both one-form symmetries and ordinary zero-form symmetries are present, they can act on each other in several ways, giving rise to a two-group global symmetry (see for instance [9–12], as well as the review [8] for further references). Moreover, there are situations in which zero- and one-form symmetries act projectively – or fractionalize – on suitable higher-dimensional defects and excitations, which is an important ingredient in 't Hooft anomaly matching, as was recently discussed in [13,14].

magnetic monopoles, which can end magnetic vortex strings. In the absence of one-form symmetries there is no known sharp diagnostic of confinement. Indeed, it has been shown via lattice simulations [16, 17] that QCD with massive fundamental quarks does not display a sharp deconfinement phase transition at finite temperature $T$.[6] However this does not exclude the possibility of a more subtle diagnostic at $T = 0$, which we will encounter below.

A closely related fact is that theories without one-form symmetries do not admit a sharp distinction between confining and Higgs phases. In typical examples, both regimes realize a unique, fully gapped vacuum, without any dynamical degrees of freedom or symmetry breaking at long distances.[7] In the Higgs regime this can be achieved using scalar fields in the fundamental representation of the gauge group,[8] as long as there are sufficiently many such fields. Note that Higgsing can occur at parametrically weak coupling, in which case one can safely discuss the Higgs field (which may be fundamental or composite) and its vacuum expectation value (vev). However, caution is advised when referring to this regime as a Higgs phase, since it may be possible to smoothly deform the theory to the confining regime without encountering a phase transition, as suggested by the generalized Landau paradigm. We will refer to this scenario as Higgs-confinement continuity.[9] It is standard to describe the Higgs regime in terms of gauge non-invariant fields, most prominently the Higgs field itself. By contrast, the confining regime is more naturally described using gauge-invariant composites. In simple cases, it is possible to match the gapped excitations in the two regimes as well, but this is not necessary for Higgs-confinement continuity, which only refers to the absence of a phase transition.[10]

---

[6] If the quarks are massless, there is a phase transition associated with the restoration of the spontaneously broken chiral symmetry.

[7] In condensed matter parlance, there is no topological order or long-range entanglement. Later we will relax these requirements.

[8] More precisely, it requires Higgs fields that transform faithfully under the center of the gauge group (e.g. a two-index symmetric $\mathbf{6}_s$ in $SU(3)_c$ gauge theory), which necessarily break the one-form center symmetry. Note that some gauge theories, e.g. conventional QED with charge-one electrons, or QCD with an even number of colors, have the feature that fermion parity $(-1)^F$ is part of the gauge group. Thus bosonic Higgs fields (fundamental or composite) cannot faithfully represent $(-1)^F$, so that weakly-coupled Higgsing at most reduces the gauge group to a $\mathbb{Z}_2$ subgroup. Despite this feature, such theories do not possess a one-form symmetry, because there are fermions transforming in the fundamental representation. Since such theories are bosonic, more can be learned by formulating them on manifolds $M_4$ without a spin structure, if one twists the dynamical gauge fields by the second Stiefel-Whitney class $w_2(M_4) \in H^2(M_4, \mathbb{Z}_2)$.

[9] Other authors have used "Higgs-confinement complementarity" [18] and "quark-hadron continuity" [19].

[10] In general the massive particle spectrum within a given phase may undergo drastic, and even sharp, changes as we dial parameters, e.g. a stable particle may become unstable and decay. This phenomenon is particularly well studied for BPS particles in supersymmetric theories (starting with [20, 21]), where it is often referred to as wall crossing, because the changes occur abruptly at sharply defined walls in parameter space. Importantly, wall crossing does not in general imply a phase transition, i.e. the low-energy effective

Foundational results that underpin our understanding of Higgs-confinement continuity were obtained in [22, 23], where it was shown in some explicit lattice models that the two regimes are in fact parts of the same phase.[11] The exploration of Higgs-confinement continuity in 3+1d gauge theories was initiated in [18, 24]. Since then the phenomenon has been observed in copious examples (in diverse dimensions), among which supersymmetric theories furnish a particularly rich set (as reviewed for instance in [25]). In such theories it is often possible to display the continuity of Higgs and confining regimes explicitly, in part because the parameters that are being dialed are holomorphic, so that there are no singularities or phase transitions in real codimension one. All of the above has elevated Higgs-confinement continuity to the status of standard lore.

## 1.2 Higgsing and Confinement as Symmetry Protected Topological (SPT) Phases

In this paper we will explore examples where the standard Higgs-confinement lore summarized above breaks down, because the two regimes furnish distinct symmetry protected topological (SPT) phases,[12] or SPTs – a notion we briefly review in section 1.2.1 below, with a more detailed discussion in section 2. In section 1.2.2 we discuss examples of SPT-enforced Higgs-confinement transitions in a class of models we term Higgs-Yukawa-QCD theories. A full account appears in sections 3, 4, and 5.

### 1.2.1 SPT Review

SPTs can be usefully understood from different complementary points of view:

- The modern notion of SPT phases originated in condensed matter physics. Early examples include the Haldane phases for spin chains in 1+1d (see [27, 28]). Quintessential examples in 3+1d (which will play an starring role throughout this paper) are topological insulators and superconductors, see [29–40] for a highly incomplete sampling of the vast literature on this subject. The general notion of SPT phases emerged in [41–47], see [48] and [49] for reviews from a condensed matter and QFT perspective.

---

theory (including background fields) is typically smooth across the wall.

[11] It is interesting that the rigorous proof of continuity in [22] relies on inequalities that follow from positivity of the Euclidean lattice measure. Positivity of the path integral measure will also make an appearance below.

[12] Even though we focus on examples in 3+1d, this mechanism is very general. With the benefit of hindsight it is likely that other transitions in diverse dimensions can be similarly interpreted. An example in 2+1d, pointed out to us by Zohar Komargodski, appears in section 4.2 of [26].

In general, SPT phases are fully gapped phases of matter with only short-range entanglement, i.e. they do not have any long range dynamics (gapless or topological) and they do not display any symmetry breaking – much like the featureless, gapped Higgs and confined phases described above. Below we review two important examples: (i) SPT phases with Chern-Simons effective action in 2+1d which describe the long-distance properties of integer quantum Hall systems; (ii) time-reversal invariant fermionic SPT phases in 3+1d, which characterize topological insulators and superconductors.

- SPT phases are closely related to the notion of anomaly inflow in QFT [50]. An SPT phase can be thought of as an anomaly inflow theory for non-dynamical background fields associated with global symmetries, which gives rise to anomalous (and therefore necessarily non-trivial) edge modes if studied in the presence of a boundary.

- Mathematically, SPT phases have been characterized [51–53] as deformation classes of unitary, invertible TQFTs with symmetry.[13] Another characterization of invertible gapped phases (involving suitable spectra in generalized cohomology theories) was proposed in [55–57].

Note that, unlike many field-theoretic analyses, the SPTs considered in condensed matter do not in general assume relativistic invariance, e.g. they can be meaningful on a lattice. This will be important in our discussion of QCD at finite baryon density (see section 1.3 below).

*Chern-Simons SPT in 2+1 Dimensions*

Consider a 2+1d theory with an global $U(1)$ zero-form flavor symmetry. Assume that the theory is trivially gapped, with a unique vacuum and no long-range entanglement; in particular, the $U(1)$ symmetry is not spontaneously broken. It then follows that the low-energy effective action is a local functional of the background gauge field $A$ associated with the $U(1)$ symmetry. This effective action may contain a Chern-Simons term, which (in Euclidean signature) reads[14]

$$S_E \supset \frac{ik}{4\pi} \int_{M_3} A \wedge dA \,, \qquad k \in \mathbb{Z} \,. \tag{1.1}$$

Let us highlight several important features,[15] which have close analogues for all SPTs:

---

[13] The notion of invertible TQFTs, which essentially describe topological actions for non-dynamical background fields, first appeared in [54].

[14] Here $S_E$ is the Euclidean action, so that the path-integral weight is $e^{-S_E}$. We take $M_3$ to be a spin manifold.

[15] See for instance [58, 59] for a detailed discussion of Chern-Simons terms for background fields.

- Invariance under large $U(1)$ background gauge transformations on closed manifolds $M_3$ quantizes the Chern-Simons level $k \in \mathbb{Z}$.[16]

- In the presence of $U(1)$ background fields, the Chern-Simons term contributes a non-trivial phase to the Euclidean partition function on closed $M_3$.

- If $M_3$ has a boundary $\partial M_3 = M_2$, there are anomalous edge modes on $M_2$ that cancel the $U(1)$ anomaly inflow from the bulk Chern-Simons term onto the boundary.[17]

- If we dial the continuous parameters of the 2+1d bulk theory without closing its gap,[18] then the quantized Chern-Simons level $k \in \mathbb{Z}$ can only jump discontinuously across a first-order bulk phase transition to a different gapped SPT phase. Conversely, any jump in $k$ necessarily signals a bulk phase transition, which may be either first order (if the gap does not close) or second order (if the gap closes). This phase transition cannot be characterized in terms of any known symmetry breaking, and hence it falls outside the Landau paradigm and its generalizations.[19]

- The Chern-Simons term with quantized level is a valid local counterterm, which must be fixed once and for all, e.g. by specifying a UV regularization scheme. The value of $k$ in a given phase then depends on the choice of scheme. By contrast, the amount $\Delta k$ by which $k$ jumps between two gapped phases is scheme independent. A simple example with $\Delta k = \pm q^2$ is a free Dirac fermion of $U(1)$ charge $q$, whose real mass $m \in \mathbb{R}$ is dialed through $m = 0$ (where the gap closes), leading to a second order transition there.

Since the $U(1)$ symmetry (together with the bulk gap) is responsible for ensuring these properties, the background Chern-Simons term above is an SPT protected by that symmetry.

---

[16] In the context of 2+1d quantum Hall physics, this means we are dealing with the integer quantum Hall effect, rather than the fractional one. In the presence of a gap, the latter requires topological order described by a non-invertible TQFT (with long-range entanglement). By taking the variation of the level-$k$ Chern-Simons term in (1.1) with respect to the background gauge field $A$, one deduces that the quantum Hall conductance is $\sigma_{xy} = k$ in unites of $e^2/h$, with $e$ the electron charge and $h$ Planck's constant (see [60–64]).

[17] In this example the boundary theory on $M_2$ is necessarily gapless and symmetry preserving, but more general SPT boundaries can also be gapped or spontaneously break some symmetries.

[18] Here we are assuming infinite volume and no boundary. If present, the boundary may well be gapless.

[19] By contrast, gapped phases with topological order, whose long-distance dynamics is described by a non-invertible TQFT, can be thought of as symmetry-breaking phases for suitable generalized symmetries (either exact or emergent) whose associated topological defects are furnished by the TQFT.

*Time-Reversal Invariant Fermion SPTs in 3+1 Dimensions*

A prototypical example of an SPT-enforced phase transition in 3+1d is furnished by a single 2-component Weyl fermion $\psi_\alpha$ , with $\alpha = 1, 2$ a left-handed spinor index,[20]

$$\mathscr{L}_{\text{Weyl}} = -i\overline{\psi}\overline{\sigma}^\mu D_\mu \psi - \frac{m}{2}\left(\psi\psi + \overline{\psi}\overline{\psi}\right) , \qquad m \in \mathbb{R} . \tag{1.2}$$

Here $D_\mu$ is the spinor covariant derivative on a (generally curved) spacetime manifold $M_4$. The reality of the (in principle complex) mass $m$ is enforced by the following orientation-reversing symmetries,[21]

$$\mathsf{T} : \psi_\alpha(t, \vec{x}) \to \psi^\alpha(-t, \vec{x}) , \qquad \mathsf{CP} : \psi_\alpha(t, \vec{x}) \to i\overline{\psi}^{\dot\alpha}(t, -\vec{x}) , \qquad \mathsf{T}^2 = (\mathsf{CP})^2 = (-1)^F . \tag{1.3}$$

Since the theory in (1.2) is Lorentz invariant, it necessarily has an unbroken $\mathsf{CPT}$ symmetry. We can therefore focus either on $\mathsf{T}$ or on $\mathsf{CP}$; we choose to focus on $\mathsf{T}$.

Let us analyze the behavior of the theory as a function of $m$:

- If $m > 0$ we can regularize the theory (i.e. choose counterterms) in such a way that the path integral measure is positive, resulting in a trivial SPT.

- If $m < 0$, integrating out the massive fermion generates a non-trivial gravitational theta-angle $\theta_g$ for the background metric,

$$S_E \supset \frac{i\theta_g}{384\pi^2} \int_{M_4} \text{Tr}\left(R \wedge R\right) , \qquad \theta_g = \pi , \tag{1.4}$$

  with $R$ the Riemann curvature two-form. Since $\theta_g \sim \theta_g + 2\pi$, it follows that $\theta_g = \pi$ is $\mathsf{T}$-invariant. Thus $\theta_g = \pi$ constitutes a non-trivial SPT protected by time-reversal symmetry. This SPT can be detected by examining the sign of the partition function on Riemannian spacetime manifolds $M_4$.[22] This SPT describes the bulk of a 3+1d topological superconductor in condensed matter physics, where it is typically diagnosed by examining the thermal Hall conductivity on the 2+1d boundary (see section 2.3.3

---

[20] We use Wess and Bagger conventions [65] for 2-component spinors. Note that a single 2-component Weyl fermion $\psi_\alpha$ is equivalent to a 4-component Majorana fermion $\Psi_M = (\psi_\alpha, \overline{\psi}^{\dot\alpha})$ in 3+1 dimensions.

[21] Since a single Weyl fermion does not have a charge conjugation symmetry, what is called $\mathsf{CP}$ here (for consistency with the rest of the paper) could just as well be called $\mathsf{P}$. Note also that we use the symbol $\mathsf{T}$ for time-reversal symmetry, while $T$ is reserved for temperature.

[22] We take $M_4$ to be an oriented four-manifold equipped with a spin structure. If $\theta_g = \pi$, the phase of the partition function on such an $M_4$ is $(-1)^{\sigma/16}$, with $\sigma \in 16\mathbb{Z}$ the signature of the spin four-manifold $M_4$, e.g. this phase is non-trivial if $M_4$ is a K3 surface, for which $\sigma = 16$.

for further details). Unlike the partition function on closed spacetime manifolds, this criterion is also available in situations where Lorentz invariance is explicitly broken, as is typically the case in condensed matter physics, and also in section 1.3 below.

The classification of topological superconductors via $\theta_g = 0, \pi$ applies if the spacetime manifold $M_4$ is oriented. A more refined $\mathbb{Z}_{16}$ classification can be obtained by using the orientation-reversing T-symmetry to place the theory on unorientable spacetime manifolds $M_4$, or alternatively by examining the anomalous realization of T-symmetry on 2+1d boundaries, see for instance [66–68, 52, 57, 49, 69].

In this paper we will focus on oriented spacetime manifolds. A more refined analysis of Higgs-confinement transitions using SPTs on unorientable manifolds will appear in a companion paper [70].

- At $m = 0$ the fermion is massless and there is a second-order phase transition. In more general interacting theories the transition can also be first order. As in the Chern-Simons example discussed above, this phase transition cannot be characterized in terms of Landau-type symmetry breaking. Rather, it is enforced by the distinct SPT phases at positive and negative $m$.

Note that T-symmetry, which forces $m \in \mathbb{R}$ and quantizes the (otherwise continuous) gravitational theta-angle to the values $\theta_g = 0, \pi$, is crucial. If we explicitly break T by allowing complex masses $m \in \mathbb{C}$, we can interpolate from positive to negative $m$ without encountering the massless point at $m = 0$.

A variant of the preceding discussion involves integrating out T-symmetric negative-mass fermions that are also charged under a flavor symmetry $G_f$. This can give rise to a theta-angle $\theta_f = \pi$ for the $G_f$ flavor background gauge fields,[23] which constitutes an SPT protected by T and $G_f$.

The case $G_f = U(1)$ corresponds to a (fermionic) topological insulator in condensed matter physics. It can be realized by a single 4-component Dirac fermion $\Psi_D$, which (throughout this paper) we will represent by a pair $\psi_\alpha, \chi_\alpha$ of 2-component Weyl fermions, so that $\Psi_D = (\psi_\alpha, \overline{\chi}^{\dot\alpha})$. The free Dirac Lagrangian is then given by

$$
\begin{aligned}
\mathscr{L}_{\text{Dirac}} &= -i\overline{\Psi}_D \gamma^\mu D_\mu \Psi_D - m\overline{\Psi}_D \Psi_D \\
&= -i\overline{\psi}\overline{\sigma}^\mu D_\mu \psi - i\overline{\chi}\overline{\sigma}^\mu D_\mu \chi - m(\psi\chi + \overline{\psi}\overline{\chi}) , \qquad m \in \mathbb{R} .
\end{aligned}
\tag{1.5}
$$

---

[23] The periodicity of $\theta_f$, and hence its non-trivial T-invariant midpoint, are dictated by the global form of the flavor symmetry group $G_f$.

If $\Psi_D = (\psi_\alpha, \overline{\chi}^{\dot{\alpha}})$ has unit $U(1)$ charge, then $\psi_\alpha, \chi_\alpha$ have charges $(+1)$ and $(-1)$, respectively. Taking $m > 0$ to be the trivial SPT implies that $m < 0$ has background theta-angle $\theta_f = \pi$ for the $U(1)$ flavor symmetry, while $\theta_g = 2\pi = 0$ (modulo $2\pi$), because the masses of both Weyl fermions flip sign.

### 1.2.2 SPT-Enforced Higgs-Confinement Transitions in Higgs-Yukawa-QCD

In this section we analyze some examples of non-Abelian gauge theories in 3+1d with both fermionic quarks and elementary scalar Higgs fields in the fundamental representation. These theories can be thought of as deformations of QCD with $SU(N)_c$ gauge group to which we add elementary Higgs fields in the $SU(N)_c$ fundamental. Importantly, these Higgs fields couple to the quarks via Yukawa couplings. We broadly refer to this class of models as Higgs-Yukawa-QCD theories, with schematic Lagrangian

$$\mathscr{L}_{\text{Higgs-Yukawa-QCD}} = \mathscr{L}_{\text{QCD}} + \mathscr{L}_{\text{Higgs}} + \mathscr{L}_{\text{Yukawa}} \ . \tag{1.6}$$

These theories do not possess any known one-form (or other generalized) symmetries, but they do have conventional flavor and spacetime zero-form symmetries. Importantly, all of the vector-like examples we consider will possess an anti-unitary time-reversal symmetry $\mathsf{T}$,[24] which enforces reality of the quark masses and Yukawa couplings, and can lead to $\mathsf{T}$-invariant SPTs of the kind reviewed in section 1.2.1 above.

Within this class of Higgs-Yukawa-QCD models, we will analyze explicit examples with $SU(2)_c$ and $SU(3)_c$ gauge group (see below), whose Higgs and confining regimes are both fully gapped and featureless at long distances, but which are nevertheless separated by at least one phase transition. These Higgs-confinement phase transitions, which are not expected within the generalized Landau paradigm, are enforced by the fact that the two regimes constitute distinct SPT phases for the unbroken global zero-form symmetries:

(C) The confining phase is continuously connected to QCD without Yukawa couplings.[25] In the presence of a common, positive Dirac quark mass $m_q > 0$, QCD can be regulated in such a way that its path integral measure is positive on all Euclidean spacetime manifolds $M_4$ [71–73].[26] Since the partition function $Z[M_4] > 0$ is then also positive,

---

[24] In fact all of our examples will also have a parity symmetry $\mathsf{P}$.

[25] See section 3 for a review.

[26] Here we take $M_4$ to be a spin manifold, and the statement of positivity also holds in the presence of background fields for the vector-like flavor symmetries that survive in the presence of a non-zero quark mass $m_q$. As we review in section 3.4, positivity of the QCD path integral measure depends on a choice of

the $m_q > 0$ confining phase constitutes the trivial reference SPT relative to which all other SPT phases are determined.

(H) In the Higgs phase, which extends to arbitrarily weak gauge coupling and can thus be analyzed reliably, the Yukawa couplings $\mathscr{L}_{\text{Yukawa}}$ in (1.6) contribute Majorana masses for the quarks, in addition to the Dirac mass $m_q$ that is already present in the confining phase. While the Yukawa couplings ensure that all fermions are massive in the Higgs phase, some real mass eigenvalues can flip sign relative to the confining phase.

This leads to the $\theta_g = \pi$ gravitational SPT or the $\theta_f = \pi$ flavor SPT (or both) in the Higgs phase, which is therefore distinct from the trivial confining phase, where $\theta_g = \theta_f = 0$. Note that $\theta_g = \pi$ requires an odd number of Weyl fermions to flip sign, while $\theta_f = \pi$ requires fermions in suitable $G_f$ representations to flip sign.

Since the confining (C) and the Higgs (H) phases above are distinct SPTs, there must be at least one phase transition between them (which may be first or second order). We can thus use SPTs to distinguish these two phases,[27] even though they are identical as far as the generalized Landau paradigm is concerned.

Some additional comments are in order:

1.) Since the non-trivial $\theta_g, \theta_f = \pi$ SPTs above give rise to signs in the Euclidean partition function $Z[M_4]$, they can only arise if the positivity of the path integral measure that holds deep in the confining phase (thanks to [71–73], see above) is ruined. Precisely this is accomplished by the Yukawa couplings in (1.6).

2.) The SPT-enforced phase transitions discussed above occur at zero temperature, $T = 0$. At finite temperature, SPTs protected by only zero-from symmetries are unstable: the distinct SPTs that exist at $T = 0$ generically become continuously connected to the trivial SPT at finite temperature [75].[28] This observation is consistent with our remarks on finite-temperature QCD in section 1.1 above.

---

regularization scheme, which (among other things) trivializes all SPT counterterms when $m_q > 0$. See [74] for a detailed discussion.

[27] As we have emphasized, which SPT is trivial constitutes an arbitrary choice of scheme. In QCD, enforcing positivity of the path integral measure for $m_q > 0$ is natural, leading to a trivial SPT in the confined phase.

[28] This can be illustrated using a free fermion of mass $m$: finite $T$ amounts to compactifying on $S^1_\beta$ with $\beta = T^{-1}$ and anti-periodic fermion boundary conditions, so that the smallest Matsubara frequency (or Kaluza-Klein mass) $m(T)$ satisfies $(m(T))^2 = m^2 + (T/2)^2$. At $T > 0$ we can thus interpolate between positive and negative $m$ without closing the gap.

3.) The idea that certain Higgs phases can be interpreted as SPTs was recently explored in [76, 77].[29] These authors analyzed Abelian gauge theories with (exact or emergent) magnetic one-form symmetries,[30] as well as zero-form symmetries, and considered mixed SPTs protected by both kinds of symmetries.[31] By contrast, all of our examples are $SU(N)$ gauge theories with fundamental matter, which only possess zero-form symmetries.[32] For this reason, the SPT phases we encounter are more akin to those in [74], whose authors analyzed the T-invariant SPTs that can arise in QCD (without elementary Higgs fields or Yukawa couplings) as one dials the quark mass $m_q \in \mathbb{R}$.

### 1.2.3 Example: $SU(2)$ Higgs-Yukawa-QCD

This model is in part motivated by [24], and will be further discussed in section 4. We start with $SU(2)_c$ QCD with $N_f = 1$ Dirac flavor, which amounts to two Weyl flavors,

$$(\psi_\alpha)_i^a \ . \tag{1.7}$$

Here $a = 1, 2$ is an $SU(2)_c$ color index, and $i = 1, 2$ is an $SU(2)_f$ flavor index, i.e. the Weyl quarks transform in the bifundamental $(\mathbf{2}, \mathbf{2})$ of $SU(2)_c \times SU(2)_f$. To this theory we add a real scalar Higgs field, which also transforms in the $(\mathbf{2}, \mathbf{2})$ representation of $SU(2)_c \times SU(2)_f$,

$$h_a^i = (h_a^i)^\dagger \ . \tag{1.8}$$

---

[29] The statement that a given phase is a non-trivial SPT depends on a choice of SPT counterterms that must be made once and for all; only relative SPT jumps are scheme-independent. The authors of [76, 77] worked in a scheme that is natural from the point of view of the lattice models they analyzed.

[30] The setup of [76, 77] can be generalized to non-Abelian gauge theories whose gauge group $G$ is not simply connected, e.g. $G = SO(3)$. Such theories have a magnetic one-form symmetry given by $\pi_1(G)$.

[31] SPT phases protected by only one-form symmetries were discussed in [9, 6]. They give rise to a refined notion of confinement in non-Abelian gauge theories without (or with suitably restricted) matter fields, which possess an electric one-form symmetry associated with the center of the gauge group.

[32] Another distinction is that the examples in [76, 77] only involve bosonic fields, while our examples involve fermions. However, fermions are not necessary to engineer a $\pi$-jump in $\theta_g, \theta_f$. For instance, we can use an axion field – a compact boson $a \sim a + 2\pi$, which can be thought of as a dynamical version of $\theta_g$ or $\theta_f$, with $a$ normalized to match their periodicity on non-spin manifolds. (However, unlike the standard QCD axion, we do not couple $a$ to the instanton density of the dynamical gauge fields.) By dialing a suitable T- and P-invariant axion potential $V(a)$ (e.g. dial $\lambda \in \mathbb{R}$ in $V(a) = \lambda \cos a$), we can engineer a phase transition from $a = 0$ to $a = \pi$ (neither of which spontaneously break T or P, because $a$ does not couple to the dynamical instanton density) that is accompanied by a corresponding SPT jump in $\theta_f, \theta_g$.

We take the quarks to have a common positive mass,[33]

$$\mathscr{L}_{\text{QCD}} \supset -\frac{m_q}{2}\varepsilon_{ab}\varepsilon^{ij}\psi_i^a\psi_j^b \ , \qquad m_q > 0 \ , \tag{1.9}$$

which preserves both the vector-like $SU(2)_f$ symmetry and time-reversal symmetry $\mathsf{T}$.

In addition to suitable kinetic terms for all the fields, as well as the quark masses (1.9), we add the following terms to the Lagrangian:

- A scalar potential for the Higgs field $h_a^i$,

$$V(h) = M_h^2|h|^2 + \lambda|h|^4 \ , \qquad M_h^2 \in \mathbb{R} \ , \qquad \lambda > 0 \ . \tag{1.10}$$

  Here $|h|^2$ is the $SU(2)_c \times SU(2)_f$ invariant norm of $h_a^i$.

- $SU(2)_c \times SU(2)_f$ invariant Yukawa couplings,[34]

$$\mathscr{L}_{\text{Yukawa}} = \frac{1}{2}\left(y_1 h_a^i h_b^j + y_2 h_a^j h_b^i\right)\psi_i^a\psi_j^b + (\text{h.c.}) \ , \qquad y_1, y_2 > 0 \ . \tag{1.11}$$

  Requiring $\mathsf{T}$-symmetry only imposes $y_1, y_2 \in \mathbb{R}$; for now we further restrict to the case $y_1, y_2 > 0$.[35] In the presence of these Yukawa couplings the path integral measure is no longer positive, which raises the possibility of non-trivial SPT phases.

We now explore the phase diagram of the theory (sketched in figure 1) as a function of the Higgs mass squared $M_h^2 \in \mathbb{R}$:

(C) When $M_h^2 \gg \Lambda_{\text{QCD}}^2$ (with $\Lambda_{\text{QCD}}$ the $SU(2)_c$ strong-coupling scale), we can integrate out the heavy Higgs to obtain $SU(2)_c$ QCD with $N_f = 1$ flavor and a positive quark mass $m_q > 0$,[36] deformed by irrelevant operators suppressed by powers of $M_h$. Following section 1.2.2, we choose a scheme in which this confining phase is the trivial SPT.

---

[33] Here $\varepsilon_{ab}, \varepsilon^{ij}$ are anti-symmetric $SU(2)$ invariant symbols, which can be used to raise and lower fundamental $SU(2)$ indices.

[34] These are irrelevant dimension five operators, so that $y_1, y_2 \sim 1/\Lambda_{\text{UV}}$. This means that we can treat $y_1, y_2$ as small parameters and work to leading order. For the purpose of this example, we ignore operators of dimension six or higher, i.e. we tune their coefficients to be suitably small as needed.

[35] The other cases will be considered in section 4. As we will show there the signs of $y_1, y_2$ are meaningful in this model, and changing them gives rise to different SPT phases.

[36] Following [71–73], we take $m_q$ and $M_h^2$ to be bare mass parameters, defined with respect to a suitable UV cutoff.

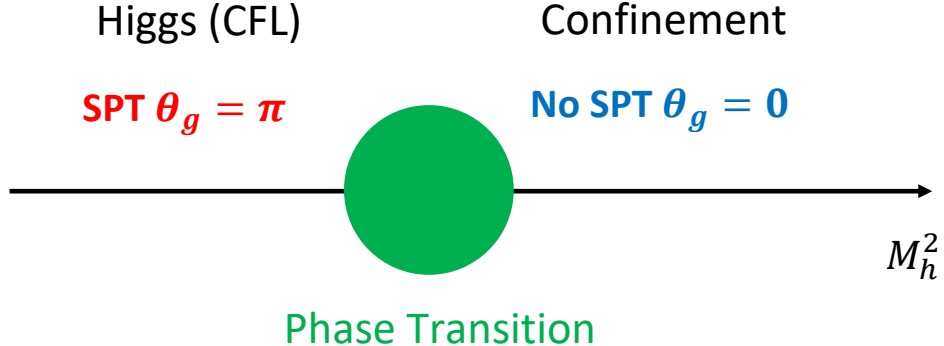

Figure 1: Phase diagram of $SU(2)_c$ Higgs-Yukawa-QCD with one Dirac quark, as a function of the Higgs mass squared $M_h^2 \in \mathbb{R}$.

(H) When $M_h^2 \ll -\Lambda_{\mathrm{QCD}}^2$, the Higgs field acquires a color-flavor locking (CFL) vev,

$$h_a^i = v\delta_a^i \,, \qquad v > 0 \,. \tag{1.12}$$

Here we take $v$ to be sufficiently large,[37] in part to ensure that we are at weak $SU(2)_c$ gauge coupling. The CFL vev (1.12) completely Higgses the $SU(2)_c$ gauge symmetry; the $SU(2)_f$ flavor symmetry is preserved by mixing with the gauge symmetry. Under this unbroken $SU(2)_f$, the Weyl fermions decompose into a triplet $\mathbf{3}$ and a singlet $\mathbf{1}$.[38]

Substituting the Higgs vev (1.12) into the Yukawa couplings (1.11), we find that all fermions are massive. However, the mass of the $\mathbf{3}$ triplet Weyl fermions has flipped sign relative to the confined phase (C), while the $\mathbf{1}$ singlet mass has not. Since this is an odd number of sign flips, the Higgs phase has $\theta_g = \pi$ and is thus a non-trivial SPT.[39]

Thus the Yukawa couplings $y_1, y_2 > 0$ induce an SPT jump, and hence force an unexpected Higgs-Confinement phase transition, in accordance with the general discussion in section 1.2.2.

Two further comments are in order:

- The authors of [24] considered the same theory without Yukawa couplings, $y_1, y_2 = 0$,

---

[37] More precisely, we need $v \gg \Lambda_{\mathrm{QCD}}$ to ensure weak coupling and $y_1, y_2 v^2 \gtrsim m_q$ to trigger the SPT jump.

[38] To see this, note that the Higgs vev $h_a^i \sim \delta_a^i$ diagonally identifies $SU(2)_c$ and $SU(2)_f$ indices. Thus the Weyl fermions $(\psi_\alpha)_i^a \to (\psi_\alpha)_i^j$ become a $\mathbf{2} \otimes \mathbf{2} = \mathbf{3} \oplus \mathbf{1}$ of the unbroken $SU(2)_f$.

[39] As we explain in section 4, there is also an SPT for the flavor symmetry group, but the details involve its global structure, which is $SO(3)_f = SU(2)_f/\mathbb{Z}_2$.

and concluded that the Higgs and confining regimes appear to be continuously connected. Indeed there is no SPT jump in this case, and hence no need for a phase transition. This illustrates the fact that the SPT jump is entirely driven by the fermion masses, which are not intrinsically linked to the strong non-Abelian gauge dynamics that is at play in the Higgs-confinement crossover regime.[40]

- The SPT phases above are protected by time-reversal symmetry $\mathsf{T}$. If $\mathsf{T}$ is explicitly broken, the Yukawa couplings in (1.11) can be complex, $y_1, y_2 \in \mathbb{C}$. This makes it possible to smoothly connect positive and negative fermion masses (and hence $\theta_g = 0$ and $\theta_g = \pi$) without encountering a phase transition, as in the free massive fermion examples in section 1.2.1 above.

### 1.2.4   Example: $SU(3)$ Higgs-Yukawa-QCD

This example (which is further discussed in section 5) is obtained by adding fundamental Higgs fields and Yukawa couplings to $SU(3)_c$ QCD with $N_f = 3$ Dirac flavors,[41] which amounts to three left-right pairs of Weyl fermion quarks,

$$(\psi_\alpha)^a_i \ , \qquad (\chi_\alpha)^i_a \ . \tag{1.13}$$

Here $a, i = 1, 2, 3$ are (anti-) fundamental $SU(3)_c$ color and $SU(3)_f$ flavor indices, respectively. The left-handed quarks $(\psi_\alpha)^a_i$ transform in the bifundamental $(\mathbf{3}, \mathbf{3})$ of $SU(3)_c \times SU(3)_f$, while the right-handed quarks $(\chi_\alpha)^i_a$ transform in the conjugate $(\overline{\mathbf{3}}, \overline{\mathbf{3}})$, as indicated by the up-down placement of the indices.

To this theory we add an elementary scalar Higgs field, which (like the right-handed quarks) transforms in the complex $(\overline{\mathbf{3}}, \overline{\mathbf{3}})$ of $SU(3)_c \times SU(3)_f$,

$$h^i_a \ . \tag{1.14}$$

In addition to the kinetic terms for all fields, we add the following terms to the Lagrangian:

---

[40] As we elaborate in section 4, the SPT jump may occur entirely within the weakly-coupled Higgs regime, or it may occur within the strong-coupling region, depending on how we dial the parameters. We are grateful to Nati Seiberg for discussions about this issue.

[41] The details of this model are motivated by the behavior of conventional three-flavor QCD (without elementary Higgs fields) at large chemical potential $\mu_B$ for $U(1)_B$ baryon number (see section 1.3 below).

- A positive, $SU(3)_c \times SU(3)_f$ preserving Dirac mass for the quarks,[42]

$$\mathscr{L}_{\text{QCD}} \supset -m_q \left( \psi_i^a \chi_a^i + \overline{\psi}_a^i \overline{\chi}_i^a \right) , \qquad m_q > 0 . \tag{1.15}$$

Note that any real quark mass $m_q \in \mathbb{R}$ preserves both time-reversal $\mathsf{T}$ and parity $\mathsf{P}$, and hence by the $\mathsf{CPT}$ theorem also charge-conjugation $\mathsf{C}$.[43] This mass term also preserves a vector-like $U(1)_B$ baryon number symmetry, under which the quarks have charges

$$B(\psi_i^a) = -B(\chi_a^i) = \frac{1}{3} . \tag{1.16}$$

- Yukawa couplings that preserve both $SU(3)_c \times SU(3)_f$, as well as $\mathsf{T}$, $\mathsf{P}$, and $\mathsf{C}$,[44]

$$\mathscr{L}_{\text{Yukawa}} = y\varepsilon_{abc}\varepsilon^{ijk}\overline{h}_i^a \left( \psi_j^b \psi_k^c + \overline{\chi}_j^b \overline{\chi}_k^c \right) + (\text{h.c.}) , \qquad y > 0 . \tag{1.17}$$

We can also preserve the $U(1)_B$ baryon number symmetry if we assign

$$B(h_a^i) = \frac{2}{3} . \tag{1.18}$$

Via tree-level Higgs exchange, the Yukawa couplings (1.17) give rise to an attractive quark-quark interaction in the parity-even Lorentz scalar and color/flavor anti-symmetric channel – exactly the same channel that is mediated by one-gluon exchange between the quarks. This will play an important role in section 1.3, where we discuss QCD at high baryon density.

- We also add a suitable scalar potential for $h_a^i$ that is invariant under all the gauge and global symmetries of the model. (We spell out the details in section 5.) Below we will only quote the behavior of the model as a function of the Higgs mass-squared $M_h^2 \in \mathbb{R}$.

We proceed to explore the phase diagram (sketched in the top panel of figure 2) of the $SU(3)_c$ Higgs-Yukawa-QCD model as a function of $M_h^2 \in \mathbb{R}$ :

(C) When $M_h^2 \gg \Lambda_{\text{QCD}}^2$ we can integrate out the Higgs field, leaving $SU(3)_c$ QCD with $N_f = 3$ flavors and a positive quark mass $m_q > 0$, which is in the trivial confining SPT

---

[42] Since both $N_c = N_f = 3$ are odd, a negative quark mass $m_q < 0$ (though itself $\mathsf{T}$-invariant) would lead to spontaneous $\mathsf{T}$-breaking in the IR (see for instance [78, 74]), which is not observed in nature.

[43] The action of these symmetries in QCD are spelled out in section 3.

[44] Here the reality of $y$ is enforced by $\mathsf{T}$ and $\mathsf{P}$, and we can flip the sign of $h_a^i$ to achieve $y > 0$. Thus, unlike the $SU(2)_c$ Higgs-Yukawa-QCD example discussed above, the sign of $y$ is not physically significant here.

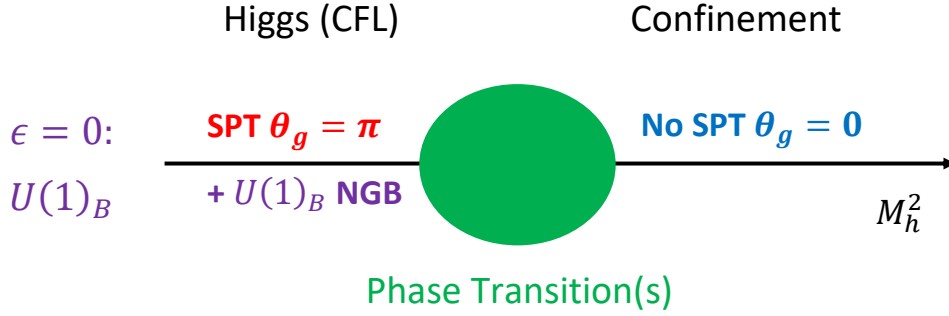

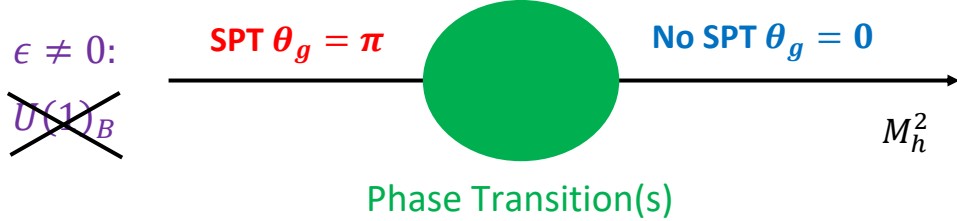

Figure 2: Top Panel: Phases of $SU(3)_c$ Higgs-Yukawa-QCD with three quark flavors, as a function of the Higgs mass squared $M_h^2 \in \mathbb{R}$. Bottom Panel: Turning on the $U(1)_B$-breaking deformation $\Delta\mathcal{L} = \varepsilon \det(h_a^i) + (\text{h.c.})$ lifts the massless NGB in the Higgs phase.

phase.

(H) When $M_h^2 \ll -\Lambda_{\mathrm{QCD}}^2$ we can reliably analyze the resulting weakly-coupled Higgs phase. The scalar potential is chosen to engineer a color-flavor locking (CFL) Higgs vev,

$$h_a^i = v\delta_a^i \ , \qquad v \in \mathbb{C} \ , \tag{1.19}$$

where we take $|v|$ sufficiently large.[45] The $SU(3)_c$ gauge symmetry is completely Higgsed; the $SU(3)_f$ flavor symmetry is preserved via mixing with the gauge symmetry.

A new feature is that the $U(1)_B$ baryon number symmetry is spontaneously broken to its $\mathbb{Z}_2$ fermion number subgroup by the vev of a gauge- and flavor-invariant operator of baryon number $B = 2$,

$$\langle \det(h_a^i) \rangle = v^3 \ . \tag{1.20}$$

Almost all fields acquire a mass (from Higgsing, the scalar potential, or the Yukawa

---

[45] As before, we assume $|v| \gg \Lambda_{\mathrm{QCD}}$ and $y|v| \gtrsim m_q$.

couplings). The only exception is the massless Nambu-Goldstone boson (NGB) for the spontaneously broken $U(1)_B$ symmetry.

In principle, gapless modes can absorb – or eat – an SPT (which only involves background fields) via a field redefinition of the gapless dynamical fields, thereby rendering the SPT meaningless. Whether or not this happens depends on the detailed properties of the gapless modes. For instance, gapless NGBs for broken symmetries are present in a variety of physical systems that are also believed support a non-trivial SPT described by $\theta_g = \pi$, e.g. topological lattice superconductors (which have gapless phonons) and the B-phase of superfluid ${}^3$He (with NGBs associated with broken rotations).[46] In these situations, the unbroken T-symmetry protecting the SPT is suitably decoupled from the spontaneously broken symmetries giving rise to the NGBs.[47] However, as we review in section 2.5, it is in fact possible for a $U(1)$ NGB to eat the $\theta_g = \pi$ SPT, if there is a mixed 't Hooft anomaly involving the $U(1)$ symmetry and gravity background fields.

We will argue that the Higgs phase of our $SU(3)_c$ Higgs-Yukawa-QCD model furnishes a gapless SPT where the $U(1)_B$ NGB safely coexists with a $\theta_g = \pi$ SPT protected by T-symmetry. As in the examples mentioned above, this is due to the fact that the $U(1)_B$ and T symmetries are sufficiently decoupled from one another. We will establish this below, via a deformation argument that explicitly gaps out the NGB; a complementary argument based on anomalies appears in section 2.5.

Taking momentarily for granted the fact (established below) that the NGB does not interfere with the SPT, let us analyze the fermion mass matrix in the Higgs phase to show that we indeed have $\theta_g = \pi$ there. To this end, note that the CFL Higgs vev (1.19) identifies fundamental $SU(3)_c$ and $SU(3)_f$ indices, so that both the left- and right-handed quarks decompose as

$$\psi_i^a \to \psi_i^j \ , \chi_a^i \to \chi_j^i = \mathbf{3} \otimes \overline{\mathbf{3}} = \mathbf{1} \oplus \mathbf{8} \ , \tag{1.21}$$

---

[46] We are grateful to Max Metlitski and Ashvin Vishwanath for emphasizing these examples to us.

[47] There is considerable interest in gapless SPTs without NGBs, see for instance [79–87]. More recently, this includes examples that are intrinsically gapless [88]; in particular they are not tensor products of a gapless phase with an ordinary SPT phase. A simple example of a gapless SPT that is not a tensor product of this kind is a free massless Dirac fermion in 2+1d, whose $U(1)$ flavor and T symmetries have a mixed parity anomaly [89–91]. This leads to a fractional quantum Hall conductance [92]. Such fractional quantum Hall conductance can also be realized in gapped systems and thus not "intrinsic" in the definition of [88]. Another example is two Dirac fermions in 2+1d transform as doublet of $SU(2)$ flavor symmetry, which also has a mixed parity anomaly with a fractional quantum Hall conductance for $SU(2)$ symmetry. Such non-Abelian symmetry fractional quantum Hall conductance cannot be realized by TQFTs with unique vacuum on sphere, and it is a new example of intrinsically gapless SPTs not discussed in [88]. Our examples only involve gapless NGBs stacked with an ordinary SPT, and the former can be gapped without affecting the latter.

under the unbroken $SU(3)_f$ symmetry. As we show in section 5, the singlet and octet masses are given by

$$M_{\mathbf{1}} = m_q \pm 4y|v| \,, \qquad M_{\mathbf{8}} = m_q \pm 2y|v| \,. \tag{1.22}$$

Note that the Majorana mass contribution due to the Yukawa couplings is twice as large for the $\mathbf{1}$ singlet fermions, relative to the $\mathbf{8}$ octet fermions. If $|v|$ is large enough (as we assume), then one singlet and one octet – an odd number – of Weyl fermions have flipped the sign of their masses relative to the confined phase (C) above, leading to $\theta_g = \pi$. There is also an SPT for the unbroken $SU(3)_f$ flavor symmetry that we discuss in section 5. An important point, elaborated there, is that the gravitational $\theta_g = \pi$ SPT is robust to turning on $SU(3)_f$-breaking Dirac masses (as long as these masses remain real and positive); by contrast, the fate of the flavor SPT is more subtle.

In summary, the confining phase is gapped and trivial, while the deep Higgs phase spontaneously breaks $U(1)_B$ (leading to an associated NGB) and also harbors a $\theta_g = \pi$ SPT.

Note that the pattern of $U(1)_B$ symmetry breaking is sufficient to establish that the two phases are separated by a transition of conventional Landau type, with the SPT just coming along for the ride, but this is misleading: the physics of $U(1)_B$ breaking may be entirely distinct from that which leads to the SPT jump. This is most obvious in the semi-classical limit, where $U(1)_B$ breaking and Higgsing both occur as soon as $h_a^i$ gets a vev, i.e. when $M_h^2 = 0$. By contrast, the SPT jump from $\theta_g = 0$ to $\theta_g = \pi$ occurs when the singlet mass $M_{\mathbf{1}}$ in (1.22) vanishes, which requires some non-zero, strictly negative $M_h^2 < 0$.

Quantum mechanically, these estimates are modified due to strong-coupling effects. For instance, the $U(1)_B$ breaking transition (and the Higgs-confinement crossover) are expected to lie somewhere in the strong-coupling region $|M_h^2| < \Lambda_{\mathrm{QCD}}^2$. By contrast, the scale of the SPT jump depends on parameters: taking $m_q \gg \Lambda_{\mathrm{QCD}}$ and $y \sim 1$ implies that $M_{\mathbf{1}} = 0$ deep in the weakly-coupled Higgs regime, where $|v| \gg \Lambda_{\mathrm{QCD}}$. In this case we can definitively say that there are two separate phase transitions. If we instead take $m_q \lesssim \Lambda_{\mathrm{QCD}}$ (and again $y \sim 1$) then the SPT jump also merges with the strong-coupling region, in which case it is not clear whether there will be two distinct phase transitions, or perhaps only one.

Let us sharpen the notion that $U(1)_B$ breaking and SPT jump are distinct phenomena. To this end, we modify the $SU(3)_c$ Higgs-Yukawa-QCD theory above by adding the following dimension-three operator to the Lagrangian,

$$\Delta \mathscr{L} = \varepsilon \det(h_a^i) + (\text{h.c.}) \,, \tag{1.23}$$

where $\varepsilon$ has mass-dimension one. When $\varepsilon \in \mathbb{R}$ this operator preserves all symmetries, including T and P, except $U(1)_B$ baryon number, which is explicitly broken to $\mathbb{Z}_2$ fermion number. Let us examine the effect of this operator on the confining and Higgs phases:

($\text{C}_\varepsilon$) In the confining phase $h_a^i$ is very massive and we can use its equation of motion to replace $\Delta\mathscr{L}$ by a highly irrelevant six-quark operator of mass dimension nine, which does not modify the trivial SPT in the confining phase.

($\text{H}_\varepsilon$) In the Higgs phase the explicit $U(1)_B$ breaking by the operator (1.23) gives a mass to the $U(1)_B$ NGB, leading to a single gapped vacuum, while the $\theta_g = \pi$ SPT remains unaffected.

In the theory with explicit $U(1)_B$ breaking via $\varepsilon \neq 0$ we thus find that the confining ($\text{C}_\varepsilon$) and Higgs ($\text{H}_\varepsilon$) phases are both gapped and featureless – and thus superficially indistinguishable – but the non-trivial SPT in the Higgs phase forces the existence of a Higgs-Confinement phase transition (see the lower panel in figure 2).

The preceding discussion also makes it clear the the non-trivial SPT jump between the confining and Higgs phases remains meaningful even if we take the explicit $U(1)_B$ breaking interaction $\varepsilon \to 0$. This shows that the massless NBG that emerges in this limit does not trivialize the SPT.

## 1.3 Application to QCD at Finite Baryon Density

As our final example, we consider conventional three-flavor QCD (without elementary Higgs fields), with real, positive quark masses. (See section 6 for a full account.) We often simplify the discussion by assuming a common positive quark mass $m_q > 0$, though some of our results are more general and also apply for physical quark masses. QCD with three Dirac flavors preserves C, P, and T, as well as $U(1)_B$ baryon number and (in the case of a common quark mass) a vector-like $SU(3)_f$ flavor symmetry.[48]

We study this theory at finite baryon number density – more precisely, at finite $U(1)_B$ chemical potential $\mu_B$. This breaks C and Lorentz invariance, but preserves P, T, as well as the flavor symmetries. It is therefore meaningful to analyze SPT phases associated with these symmetries, such as the $\theta_g, \theta_f \in \{0, \pi\}$ SPTs introduced above. Importantly, this is

---

[48] Famously, QCD admits a $\theta$-angle for its $SU(3)$ gauge group. (Note that the value of $\theta$ modulo $2\pi$ is meaningful, since we have fixed the quark masses to be positive.) However, experiments constrain $|\theta| \lesssim 10^{-10}$ (see chapter 9 of the PDG [93]). Thus taking $\theta \simeq 0$, so that T-symmetry is neither explicitly nor spontaneously broken, is an excellent approximation.

true even though turning on $\mu_B$ breaks Lorentz invariance, as is familiar from experience in condensed matter physics (see sections 2.2.3 and 2.3.3 for are more detailed discussion).

Upon Wick rotation to Euclidean signature, a real chemical potential $\mu_B$ turns into a purely imaginary background Wilson line for the $U(1)_B$ flavor symmetry, so that the Euclidean path integral measure is no longer positive definite. On the one hand this leads to a sign problem for Monte Carlo simulations, making finite-density lattice studies very challenging. On the other hand it means that non-trivial SPT phases are (at least in principle) possible. Indeed, one of our main results (summarized in section 1.3.1 below) is that three-flavor QCD at sufficiently large $\mu_B \gg \Lambda_{QCD}, m_q$ is a non-trivial SPT phase.

While the phases of QCD as a function of $\mu_B$ have been thoroughly explored (see the reviews [94–97]), our understanding remains far from complete. Two regimes that are under good theoretical control are the asymptotic regimes of very small and very large $\mu_B$:

- When $\mu_B \ll M_{\text{baryon}} \sim \Lambda_{\text{QCD}}$, the theory is continuously connected to the standard confined QCD vacuum at $\mu_B \to 0$, i.e. it is a gapped and trivial SPT.

- When $\mu_B \gg \Lambda_{\text{QCD}}, m_q$, i.e. at very high densities, asymptotic freedom implies that the theory is in a weakly-coupled, color-superconducting Higgs phase that also spontaneously breaks $U(1)_B$, leading to a single massless NGB. This is reviewed in section 1.3.1 below. There we also explain why this high-density phase also harbors a non-trivial SPT with $\theta_g = \pi$.

We conclude this introduction in section 1.3.2 by sketching some implications of the high-density SPT for the QCD phase diagram and for neutron stars.

### 1.3.1 High-Density QCD as a Gapless SPT

At very high densities, i.e. when $\mu_B \gg \Lambda_{\text{QCD}}, m_q$, QCD is in a weakly-coupled Higgs phase, which we briefly review (following [95]), with more details in section 6.1. In this regime, the Fermi surface of the quarks (with Fermi energy $\mu_B$) is destabilized by attractive single-gluon exchange, which pairs quarks in the parity-even spin-0 channel that is anti-symmetric in both color and flavor indices. This leads to a color-flavor locking (CFL) vev for the following composite Higgs field,

$$\mathcal{H}_a^i \equiv \varepsilon_{abc}\varepsilon^{ijk}\left(\psi_j^b\psi_k^c + \overline{\chi}_j^b\overline{\chi}_k^c\right) \ , \qquad \langle\mathcal{H}_a^i\rangle = v\delta_a^i \qquad |v| \sim \mu_B \ . \tag{1.24}$$

This has exactly the same quantum numbers as the fundamental Higgs field $h_a^i$ in the $SU(3)_c$ Higgs-Yukawa-QCD model (at $\mu_B = 0$) considered in section 1.2.4 above, whose vev (1.19)

also takes the CFL form in (1.24).

Indeed, we have carefully engineered the $SU(3)_c$ Higgs-Yukawa-QCD model so that its Higgs phase shares many features (summarized below (1.19)) of the large-$\mu_B$ Higgs phase of conventional QCD. In both models the $SU(3)_c$ gauge symmetry is completely Higgsed at the scale $|v| \sim \mu_B$, and the $SU(3)_f$ flavor symmetry is preserved by mixing with the gauge symmetry. By analogy with (1.20), the $U(1)_B$ symmetry is spontaneously broken by a vev for the the following gauge- and $SU(3)_f$-invariant dibaryon operator,

$$\det\left(\mathcal{H}_a^i\right) \, , \tag{1.25}$$

which leads to a single massless NGB. Both $\mathsf{P}$ and $\mathsf{T}$ are unbroken (up to conjugation by $U(1)_B$), so that we can contemplate SPT phases protected by these symmetries.

Finally, both Higgs phases have the feature that the fermions are massive, leaving only the massless $U(1)_B$ NGB at long distances. In the Higgs-Yukawa-QCD model this is due to the explicit Yukawa couplings (1.17), which lead to the fermion masses (1.22). The fermion gaps generated in the large-$\mu_B$ regime of ordinary QCD take a very similar form, except that they are generated through non-perturbative BCS pairing. Ignoring the quark mass (which is legitimate at large-$\mu_B$), the exponential scaling of the pairing gaps in the $\mathbf{1}$ and $\mathbf{8}$ of the unbroken $SU(3)_f$ symmetry takes the following form [98],

$$\Delta_{\mathbf{1}} = 2\Delta_{\mathbf{8}} \sim \mu_B e^{-\frac{K}{g(\mu_B)}} \, , \qquad K = \frac{3\pi^2}{\sqrt{2}} \, , \tag{1.26}$$

which closely resembles (1.22) upon setting $m_q = 0$ there.

Given that the two Higgs phases described above – that of $SU(3)_c$ Higgs-Yukawa-QCD in the vacuum and that of conventional QCD at large $\mu_B$ – appear to be qualitatively identical, it is also natural to suspect that they realize the same SPT phase. This is indeed the case, as we argue in section 6.2, so that high-density QCD inherits the gravitational SPT of Higgs-Yukawa-QCD discussed around (1.22),

$$\theta_g = \pi \, . \tag{1.27}$$

As was the case there, this SPT is robust to breaking $SU(3)_f$ via distinct, positive Dirac masses for the quarks, so that it is also present in high-density QCD with physical quark masses.

As we explain in section 6.2.1, the fact that the Higgs phases of the two models realize the same SPT is due to the fact that the SPT is essentially determined by the fermions,

which enjoy pairing in the same attractive channel, leading to qualitatively identical gaps. A more complete deformation argument that explicitly relates the two phases without changing the SPT appears in section 6.2.2. Importantly, the fermion gaps never close along the entire deformation.

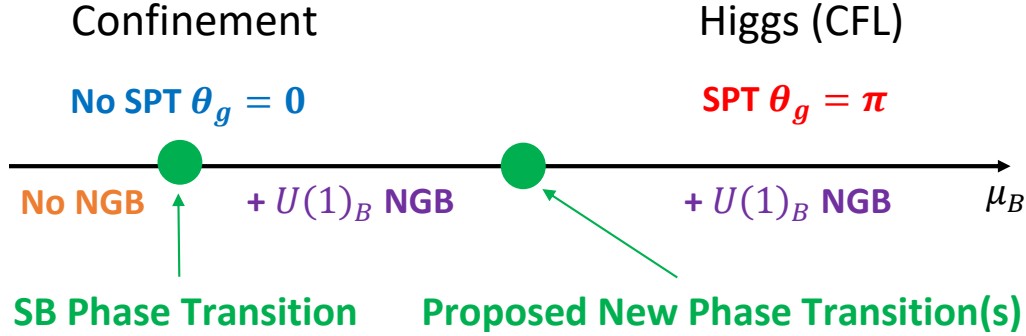

Figure 3: Phase diagram for three-flavor QCD with $SU(3)_f$ symmetry at $T = 0$ and $U(1)_B$ chemical potential $\mu_B$. As we increase $\mu_B$, the theory experiences a symmetry-breaking (SB) phase transition that spontaneously breaks $U(1)_B$, yielding a massless NGB (see [19]). We propose at least one additional phase transition within the $U(1)_B$-breaking phase where the SPT jumps from $\theta_g = 0$ to its value $\theta_g = \pi$ in the CFL Higgs phase at large $\mu_B$.

### 1.3.2   Implications for the QCD Phase Diagram and Neutron Stars

In section 6.3, which we briefly summarize here, we consider possible consequences of the fact that high-density QCD with three flavors is a non-trivial SPT with $\theta_g = \pi$.

The first application (discussed in section 6.3.1) is to the QCD phase diagram. We have already seen above that SPT jumps can enforce unexpected Higgs-confinement transitions in situations where they cannot be inferred by patterns of symmetry breaking. The presence of the $\theta_g = \pi$ SPT in QCD at large $\mu_B$ may similarly trigger new transitions in the QCD phase diagram. Here we consider the $T = 0$ phase diagram as a function of $\mu_B$ (though we expect our conclusions to also have implications at finite temperature), and we continue to focus on the $SU(3)_f$ symmetric case with a common quark mass $m_q$.

In this context, we make contact with the "quark-hadron continuity" proposal of [19] (see also [99, 100]). These authors proposed a simple phase diagram as a function of $\mu_B$: a gapped confining phase at small $\mu_B$ and a $U(1)_B$-breaking phase at large $\mu_B$ with an associated gapless NGB. These two phases are separated by a single symmetry-breaking transition, which is believed to occur at typical hadronic densities $\mu_{SF} \sim \Lambda_{QCD}$, i.e in the

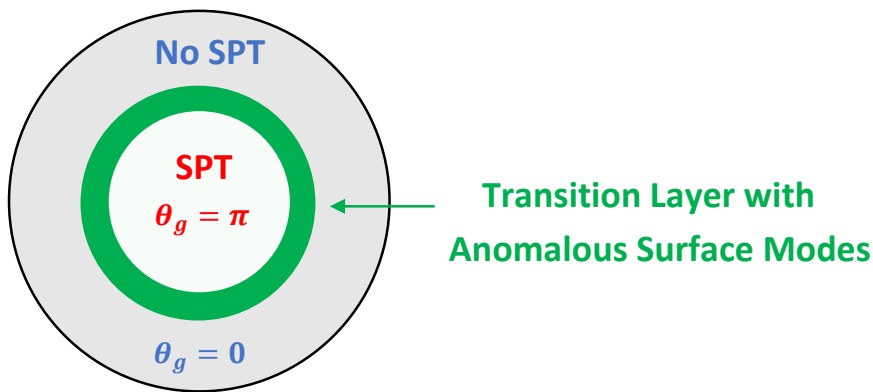

Figure 4: Neutron star with a dense quark-matter core supporting a $\theta_g = \pi$ SPT, separated from an outer region with $\theta_g = 0$ by a transition layer (shaded in green) with anomalous surface modes. In this diagram we have suppressed the gapless $U(1)_B$ NGB.

confined regime of QCD. (This is indicated by the left green dot, at lower $\mu_B$, in figure 3.) The authors of [19] proposed that the $U(1)_B$-breaking phase smoothly extends to arbitrarily high $\mu_B$, where QCD is a weakly-coupled CFL Higgs phase describe in terms of quarks (reviewed in section 1.3.1 above). This was termed quark-hadron continuity in [19]; it is a version of Higgs-confinement continuity that applies in the $U(1)_B$-breaking superfluid phase.

As we explain in section 6.3.1, it seems likely that this proposal must be modified in light of the fact that high-density QCD is an SPT with $\theta_g = \pi$, whose presence motivates a second phase transition within the $U(1)_B$ breaking phase. (This is indicated by the right green dot, at higher $\mu_B$, in figure 3.) This transition is not driven by symmetry breaking, but rather by the jump of the $\theta_g = \pi$ high-density SPT to its low-density value $\theta_g = 0$. The possibility of such a non-Landau transition within the superfluid phase was also contemplated in [101, 102], though their arguments (which focus on the $U(1)_B$ superfluid vortices) seem unrelated to our SPT considerations (which focus on the unbroken zero-form symmetries and the fermions).

Finally, in section 6.3.2, we briefly comment on the possibility that some neutron stars may contain quark-matter cores that are sufficiently dense to access the high-density CFL phase of three-flavor QCD. In this scenario the gravitational SPT jumps from $\theta_g = \pi$ in the core of the star to $\theta_g = 0$ in the outer regions (see figure 4). The transition layer through which this jump occurs (shaded green in figure 4) then necessarily harbors anomalous surface modes (e.g. massless fermions, though there are other possibilities), whose thermal Hall conductance is fixed by the SPT jump $\Delta\theta_g = \pi$. It is an interesting open question whether such anomalous surface layers inside dense stars have observable consequences.

# 2 Aspects of Symmetry Protected Topological (SPT) Phases

## 2.1 What is an SPT Phase?

In this section we review the notion of Symmetry Protected Topological (SPT) phases, elaborating on section 1.2.1 (which contains many references). A typical SPT is a gapped theory that is essentially trivial: it has a unique vacuum, and no dynamical degrees of freedom – massless or topological – at long distances. In condensed matter parlance it has neither conventional nor topological order.[49] This implies that all global symmetries – continuous or discrete, ordinary or generalized – are unbroken. In the Landau paradigm, such a phase is considered to be trivial.

What distinguishes an SPT from a genuinely trivial gapped phase is that the SPT has a topologically non-trivial action for the background fields that we can turn on in the theory, which reflect its global symmetries:

- We will be discussing relativistic QFTs in 3+1 dimensions, which can be studied on arbitrary spacetime manifolds $M_4$ with Riemannian metric $g_{\mu\nu}$, of either Lorentzian or Euclidean signature. When we discuss topological terms in the effective action, the Euclidean perspective is typically more convenient.

- In this paper we will restrict to spacetime manifolds that are orientable and carry a spin structure. The theories we discuss can also be analyzed on manifolds that do not admit such structures,[50] which leads to stronger results that we will present in [70].

- Many of our examples have an internal flavor symmetry group $G_f$. The associated background fields $A$ are ordinary connections on a principal $G_f$ bundle over the spacetime manifold $M_4$.

The partition function $Z[g, A]$, obtained by path integrating over all dynamical fields, is a functional of the background fields.[51] Because the theory is gapped, it is a local functional,

---

[49] In particular, it has a one-dimensional Hilbert space on any spatial manifold.

[50] For unorientable manifolds this requires a choice of orientation-reversing symmetry such as time-reversal or parity; for manifolds that do not admit a spin structure, it may require a twist by a suitable internal symmetry, such as baryon number. SPTs for the flavor and time-reversal symmetries of QCD on unorientable manifolds were classified in [103].

[51] It also depends on the topology of $M_4$, as well as the orientation and the spin structure.

which can be written in terms of a local effective action $S_{\text{eff}}[g, A]$ for the background fields. In Euclidean signature,

$$Z[g, A] = e^{-S_{\text{eff}}[g,A]} \ . \tag{2.1}$$

The terms in $S_{\text{eff}}[g, A]$ are typically of two types:

(i) Non-universal terms, with continuously adjustable coefficients. They are conventional local counterterms in the background fields, and (if desired) they can be continuously deformed to zero.

(ii) Topological terms in the background fields, with quantized coefficients. These are typically purely imaginary in Euclidean signature, i.e. they only affect the phase of the partition function.[52] They can still be thought of as well-defined local counterterms, albeit rigid ones that cannot be deformed continuously.

The distinguishing feature of these terms is that they implement a generalization of standard anomaly inflow [50]: they are well-defined on closed spacetime manifolds $M_4$, but cease to be so in the presence of a boundary $M_3 = \partial M_4$. Rather, they enforce the presence of particular low-energy (massless or topological) degrees of freedom on $M_3$, whose contribution to the partition function is itself anomalous, in such a way as to precisely cancel the ambiguity of the bulk terms in the presence of $M_3$. The combined bulk-boundary system is then well defined and anomaly free.

We will refer to all such quantized anomaly-inflow terms as SPTs, though this use is somewhat broader then in the condensed matter literature.[53] The preceding discussion makes it clear that SPTs can be usefully approached from two complementary points of view: via the anomaly-inflow terms for background fields in the bulk, or in terms of the anomalous boundary degrees of freedom. In this paper we mostly take the former, bulk point of view.

A familiar example of an SPT, or equivalently of anomaly inflow, is a 2+1 dimensional Chern-Simons term for a $U(1)$ background gauge field $A$,

$$S = \frac{ik}{4\pi} \int_{M_3} A \wedge dA \ . \tag{2.2}$$

---

[52] If one chooses to only keep the imaginary topological terms in $S_{\text{eff}}$, the partition function is a pure phase, whose complex conjugate coincides with its inverse. The resulting theories are known as invertible, and their relation to SPTs is discussed in [53].

[53] The term SPT is typically reserved for situations that require a particular global symmetry. Some invertible anomaly-inflow actions do not require any such symmetry, see for instance [104, 105, 51, 106–108].

Invariance under large gauge transformations on closed spin manifolds $M_3$ implies the quantization condition $k \in \mathbb{Z}$. If $M_3$ has a boundary $M_2 = \partial M_3$, then a $U(1)$ gauge transformation changes the partition function as follows,

$$Z[A + d\lambda] = Z[A] \exp\left(\frac{ik}{4\pi} \int_{M_2} \lambda \, dA\right) . \tag{2.3}$$

This implies the existence of massless edge modes on $M_2$, which couple to $A$ (they are therefore charged under the $U(1)$ symmetry), and whose contribution to the partition function is anomalous in such a way as to cancel the anomaly in (2.3). The anomaly in question is a conventional local 't Hooft anomaly associated with the two-point function of the $U(1)$ current.[54]

For our purposes in this paper, the most important thing about the Chern-Simons SPT in (2.2) is the quantization of its coefficient $k \in \mathbb{Z}$. If we imagine dialing the continuous coupling constants of the theory while preserving the $U(1)$ symmetry, and without closing the bulk gap,[55] then $k$ does not change. There are only two ways in which $k$ can change:

a) There is a first-order phase transition to a different (possibly trivial) SPT, so that $k$ jumps by an integer.

b) Gapless degrees of freedom appear, signaling a second order phase transition. (The value of $k$ in a massless theory need not be an integer, and is in general not even rational [58, 59].)

Thus two SPTs with different values of $k$ are necessarily separated by a phase transition.

The preceding statements apply to all SPTs: two gapped phases harboring distinct SPTs are necessarily separated by a quantum phase transition (which could be either first or second order), occurring at zero temperature. As was already mentioned, this phase transition cannot be characterized within the Landau paradigm, since no symmetry breaking is assumed to occur across the transition.

We will now proceed to describe the 3+1 dimensional SPT phases that are relevant for the analysis in this paper. They are characterized by $\theta$-terms for background gauge and gravity fields. The associated boundary anomalies enforced by these SPT phases are global parity anomalies in 2+1 dimensions [109, 110].

---

[54] The corresponding 't Hooft anomaly in 3+1 dimensions, which is associated with a three-point function of the $U(1)$ current, is associated with a Chern-Simons term $\sim \int AdAdA$ in 4+1 dimensions.

[55] Recall that a boundary, if present, necessarily hosts gapless degrees of freedom to cancel the bulk anomaly.

## 2.2 SPT for $U(1)$ Symmetry with $\theta_f = \pi$ Protected by Time-Reversal

### 2.2.1 Basic Properties

Given a 3+1 dimensional theory with a global $U(1)$ flavor symmetry, consider a $\theta$-term for the $U(1)$ background gauge field $A$, with field strength $F = dA$. In Euclidean signature,

$$S = \frac{i\theta_f}{8\pi^2} \int_{M_4} F \wedge F \ . \tag{2.4}$$

It follows from the Atiyah-Singer index theorem that $\theta_f$ has standard periodicity, $\theta_f \sim \theta_f + 2\pi$ on a spin manifolds $M_4$ (see for instance appendix A of [111]).

In addition to the $U(1)$ symmetry, the theory may also possess the following additional discrete symmetries:

- Charge conjugation $\mathsf{C}$, acting as

$$\mathsf{C} : A \to -A \ . \tag{2.5}$$

- Parity $\mathsf{P}$, acting as

$$\mathsf{P} : A_0(t, \vec{x}) \to A_0(t, -\vec{x}) \ , \qquad \vec{A}(t, \vec{x}) \to -\vec{A}(t, -\vec{x}) \ . \tag{2.6}$$

- Time Reversal $\mathsf{T}$, which is anti-unitary and acts via

$$\mathsf{T} : A_0(t, \vec{x}) \to A_0(-t, \vec{x}) \ , \qquad \vec{A}(t, \vec{x}) \to -\vec{A}(-t, \vec{x}) \ . \tag{2.7}$$

In relativistic theories, $\mathsf{CPT}$ is always an exact, unbroken symmetry. When this is the case, it suffices to focus on two out of the three symmetries, which we typically take to be $\mathsf{C}$ and $\mathsf{T}$. Note that $\mathsf{P}$ and $\mathsf{T}$ preserve the $U(1)$ charge, while $\mathsf{C}$ negates it.

The discrete symmetries above act on $\theta_f$ as follows,

$$\mathsf{C} : \theta_f \to \theta_f \ , \qquad \mathsf{T} : \theta_f \to -\theta_f \ . \tag{2.8}$$

Since $\theta_f$ is a $2\pi$-periodic angle, there are two values that are compatible with $\mathsf{T}$-symmetry (and hence also with $\mathsf{CT}$),

$$\theta_f = 0 \ , \qquad \theta_f = \pi \ . \tag{2.9}$$

Thus either T or CT symmetry quantizes the allowed values of $\theta_f$, and this leads to SPTs protected by either T or CT, together with $U(1)$.

The SPT with $\theta_f = 0$ is trivial, while the one corresponding to $\theta_f = \pi$ is non-trivial. This can be seen from two complementary points of view:

- The partition function of the bulk SPT on a closed spacetime manifold $M_4$ is given by

$$Z(\theta_f = 0, \pi) = e^{i\theta_f I_f} , \qquad I_f = \frac{1}{8\pi^2} \int_{M_4} F \wedge F \in \mathbb{Z} . \qquad (2.10)$$

Here $I_f$ is the $U(1)$ instanton number. Thus $Z(\theta_f)$ is always real (as required on general grounds) but while the $\theta_f = 0$ SPT always produces a positive partition function, the partition function of the SPT at $\theta_f = \pi$ can produce a non-trivial sign,

$$Z(\theta_f = \pi) = (-1)^{I_f} . \qquad (2.11)$$

- The $\theta_f = \pi$ SPT represents the anomaly-inflow action for the parity anomaly [109,110] in 2+1 dimensions: on a manifold $M_4$ with boundary $M_3 = \partial M_4$, the $\theta_f = \pi$ SPT can be integrated by parts to produce a Chern-Simons term (2.2) with level $k = \frac{1}{2}$ on $M_3$.[56] Since this term flips sign under either T or CT, the partition function of the boundary degrees of freedom on $M_3$ must compensate by transforming as follows,

$$\mathsf{T} \text{ or } \mathsf{CT} : Z_{M_3} \to Z_{M_3} \exp\left( \frac{i}{4\pi} \int_{M_3} A \wedge dA \right) . \qquad (2.12)$$

This mixed anomaly between an orientation-reversing symmetry (which can be either T or CT) and $U(1)$ in 2+1 dimensions is known as the parity anomaly. The anomaly is $\mathbb{Z}_2$-valued, in accord with the fact that the SPT partition function (2.11) is a sign.

The anomalous boundary is the defining feature of any SPT, but in some cases the SPT can also be detected in other ways. For instance, we can detect the non-trivial SPT with $\theta_f = \pi$ in (2.4) using the Witten effect [112]: a probe Dirac monopole (equivalently, an 't Hooft line defect) for the background $U(1)$ gauge field acquires $U(1)$ charge $\frac{\theta_f}{2\pi} = \frac{1}{2}$.

---

[56] This in turn gives rise to a half-integer Hall conductance in units of $e^2/h$, as discussed in section 2.2.3 below.

### 2.2.2  Example: Massive Dirac Fermion

Throughout this paper, we will forgo explicit 4-component Dirac fermions. Rather, each Dirac fermion is represented by a pair of 2-component Weyl fermions,

$$\psi_\alpha \, , \qquad \chi_\alpha \, . \tag{2.13}$$

Here $\alpha = 1, 2$ is a left-handed Weyl spinor index.[57] Hermitian conjugation exchanges left-handed undotted Weyl spinors with dotted right-handed Weyl spinors. Thus the hermitian conjugate fields are

$$\overline{\psi}_{\dot\alpha} = (\psi_\alpha)^\dagger \, , \qquad \overline{\chi}_{\dot\alpha} = (\chi_\alpha)^\dagger \, . \tag{2.14}$$

The conventional 4-component Dirac fermion is then given by

$$\Psi_{\text{Dirac}} = \left( \psi_\alpha, \, \overline{\chi}^{\dot\alpha} \right) \, . \tag{2.15}$$

The free massive Dirac fermion is described by the following Lagrangian,

$$\mathscr{L} = -i\overline{\psi}\overline{\sigma}^\mu\partial_\mu\psi - i\overline{\chi}\overline{\sigma}^\mu\partial_\mu\chi - m\psi\chi - \overline{m}\overline{\psi}\overline{\chi} \, . \tag{2.16}$$

In general the mass $m$ can be complex.

The theory (2.16) enjoys various symmetries, which will play an important role below:

- There is a vector-like $U(1)$ flavor symmetry under which the fields have charges

$$Q(\psi) = -Q(\chi) = 1 \, . \tag{2.17}$$

- Charge-conjugation $\mathsf{C}$ negates the $U(1)$ charge $Q$ and exchanges

$$\mathsf{C} : \psi_\alpha \leftrightarrow \chi_\alpha \, . \tag{2.18}$$

- Anti-unitary time-reversal symmetry $\mathsf{T}$ acts via

$$\mathsf{T} : \psi_\alpha(t, \vec{x}) \to \psi^\alpha(-t, \vec{x}) \, , \qquad \chi_\alpha(t, \vec{x}) \to \chi^\alpha(-t, \vec{x}) \, . \tag{2.19}$$

Note that $\mathsf{T}^2 = (-1)^F$. In order for this to be a symmetry of (2.16), the mass must be

---

[57] We use the conventions of Wess and Bagger [65] for 2-component Weyl spinors. See also [74].

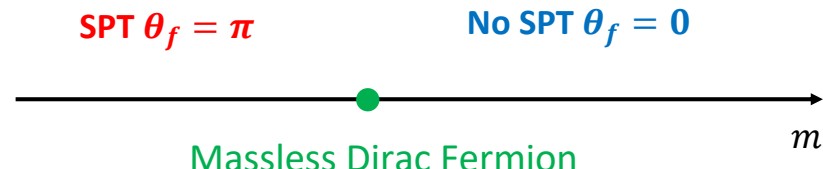

Figure 5: Phase diagram of a free Dirac fermion as a function of the T-preserving real Dirac mass $m$. The $m \neq 0$ phases are gapped and trivial but furnish different SPTs protected by the T and $U(1)$ symmetries. They are separated by a second order phase transition at $m = 0$, where a massless Dirac fermion appears.

real, which we will assume henceforth,

$$\text{T symmetry} \quad \Longleftrightarrow \quad m \in \mathbb{R} \ . \tag{2.20}$$

Note that T preserves the $U(1)$ charge $Q$. Thus the symmetry group is $U(1) \rtimes$ T; this is the symmetry characterizing a topological insulator in condensed matter physics.

- Parity acts via

$$\mathsf{P} : \psi_\alpha(t, \vec{x}) \to i\overline{\chi}^{\dot{\alpha}}(t, -\vec{x}) \ , \qquad \chi_\alpha(t, \vec{x}) \to i\overline{\psi}^{\dot{\alpha}}(t, -\vec{x}) \ . \tag{2.21}$$

This symmetry also requires a real Dirac mass $m \in \mathbb{R}$, as was the case for T, consistent with the CPT theorem.[58] Note that P preserves the $U(1)$ charge $Q$.

Some of the examples in this paper are Lorentz-invariant QFTs and therefore obey the CPT theorem. In those examples we can focus on two out of the three symmetries, e.g. on C and T, with P implied by the theorem. However, in section 6 we turn on a finite chemical potential $\mu$ for the $U(1)$ charge $Q$, which amounts to subtracting $\mu Q$ from the Hamiltonian. This explicitly breaks C, as well as Lorentz invariance, but preserves T and P. In this situation the CPT theorem no longer holds, so that T and P are genuinely independent symmetries.

We will now explore the phase diagram as a function of the real mass $m$: (see figure 5)

---

[58] The action of CPT on the fermions is given by

$$\mathsf{CPT} : \psi_\alpha(x) \to -i\overline{\psi}_{\dot{\alpha}}(-x) \ , \qquad \chi_\alpha(x) \to -i\overline{\chi}_{\dot{\alpha}}(-x) \ . \tag{2.22}$$

Note that $(\mathsf{CPT})^2 = (-1)^F$.

- When $m > 0$ the theory is gapped, and we can regulate the fermion path integral in such a way that the partition function $Z > 0$ is real and positive, e.g. by choosing the Pauli-Villars regulator fermion to have a mass with the same sign as $m$. Comparing with (2.11) we see that this corresponds to the trivial SPT. (See [49] for a detailed discussion.) More generally, we are free to modify the path integral by an explicit SPT counterterm (2.4) with $\theta_f = \pi$ in order to ensure that $m > 0$ corresponds to the trivial SPT. Thus the statement that $m > 0$ is the trivial SPT is scheme dependent and tied to a particular choice of counterterms that we make once and for all.

- When $m = 0$ the Dirac fermion is massless, and the theory undergoes a second order phase transition.

- When $m < 0$ the Dirac fermion is massive again. In the absence of background fields, this massive phase is physically indistinguishable from the massive phase at $m > 0$. This can be made explicit by a $U(1)_{\text{axial}}$ transformations of the fields,

$$U(1)_{\text{axial}} : \psi_\alpha \to e^{i\varphi/2}\psi_\alpha , \qquad \chi_\alpha \to e^{i\varphi/2}\chi_\alpha . \tag{2.23}$$

This transformation has the effect of rotating the mass $m$ in the complex plane,

$$U(1)_{\text{axial}} : m \to e^{i\varphi}m . \tag{2.24}$$

Thus a rotation with $\varphi = \pi$ exchanges positive and negative $m$.

Thanks to the standard axial-vector triangle anomaly, the $U(1)_{\text{axial}}$ symmetry is anomalous if we turn on a background field $A$ for the vector-like $U(1)$ flavor symmetry.[59] This leads to the following transformation rule for the partition function,[60]

$$Z(e^{i\varphi}m) = Z(m)\exp\left(\frac{i\varphi}{8\pi^2}\int_{M_4} F \wedge F\right) . \tag{2.25}$$

Specializing to $\varphi = \pi$, we find that flipping $m \to -m$ induces the $\theta_f = \pi$ SPT with partition function (2.11),

$$Z(-m) = Z(m)\exp\left(\frac{i\pi}{8\pi^2}\int_{M_4} F \wedge F\right) = Z(m)(-1)^{I_f} , \tag{2.26}$$

---

[59] For now we ignore gravitational backgrounds; we will include them below.

[60] This formula can be derived using the standard descent formalism, which relates triangle anomalies to index theorems; see for instance section 2 of [11] for a recent pedagogical discussion with references.

with $I_f \in \mathbb{Z}$ the $U(1)$ instanton number.

Since we have chosen counterterms so that the $m > 0$ SPT is trivial ($\theta_f = 0$), we conclude that the $m < 0$ phase necessarily harbors the non-trivial $\theta_f = \pi$ SPT.

In short, we find that the $m > 0$ and $m < 0$ phases are both gapped and trivial, but differ by an SPT. By the general logic reviewed in section 2.1, these two phases must be separated by a phase transition. Here the transition is second order and occurs at the $m = 0$ point where the Dirac fermion is massless, but the order of the phase transition is not fixed by the SPT, as we will now sketch.

It is straightforward to embed the SPT-enforced transition of the free Dirac fermion above into an interacting model by promoting the bare mass $m$ to a dynamical real scalar field $\sigma$, which is invariant under C, P, and T. If we add a generic real potential $V(\sigma)$ that contains both even and odd powers of $\sigma$, we can easily engineer a first-order jump from large positive to large negative vev for $\sigma$ as we dial parameters. No symmetries are broken across this phase transition, but the Dirac mass flips sign, leading to a discontinuous jump in the SPT.

The SPT phase discussed above is protected by time-reversal (or parity) symmetry, together with the $U(1)$ flavor symmetry. If we break time-reversal, we can continuously dial the flavor theta-angle $\theta_f$ to any real value, while preserving the $U(1)$ flavor symmetry. In particular, $\theta_f = \pi$ and $\theta_f = 0$ are smoothly connected; only T or P pins the theta-angle at $\theta_f = 0, \pi$. In the free massive fermion example, breaking T allows the fermion mass $m$ to be complex. We can therefore smoothly connect $m > 0$ and $m < 0$ without closing the gap, by passing through the complex plane. This phenomenon described above is closely related to the space-of-couplings anomalies or higher Berry phases analyzed in [113–119].

### 2.2.3 Detecting the $\theta_f = \pi$ SPT via 2+1d Quantum Hall Conductance

Here we mention two (closely related) alternative ways to determine the $\theta_f = \pi$ SPT discussed above, which involves codimension-one defects or boundaries. These alternative diagnostics have the virtue that they do not require Lorentz symmetry and can therefore be used in non-relativistic settings – including finite-density QCD (see section 6).

Consider the codimension-one defect that generates the T-symmetry, oriented along a 2+1d submanifold of spacetime. Since this defect implements the action of T, it maps the $\theta_f = \pi$ SPT on one side to $\theta_f = -\pi$, leading to an overall jump by $\Delta\theta_f = 2\pi$ across the wall. This has the effect of decorating the T-defect by a $U(1)$ Chern-Simons term with

level $k = 1$,[61] which in turn implies a non-zero quantum Hall conductance $\sigma_{xy} = 1$, in units of $e^2/h$ (with $e$ the electron charge, and $h$ Planck's constant), see for instance [39, 125].

Similarly, we can consider a symmetry-preserving boundary across which the SPT jumps from $\theta_f = \pi$ to $\theta_f = 0$. In this case the Chern-Simons term induced on the boundary is effectively fractional, $k = \frac{1}{2}$, leading to a correspondingly fractional $\sigma_{xy} = \frac{1}{2}$.

Since the $2\pi$-periodicity of $\theta_f$ in the bulk is broken in the presence of boundaries, we should more generally characterize the non-trivial SPT by $\theta_f = (2n + 1)\pi$ with $n \in \mathbb{Z}$. The integer $n$ reflects a suitable allowed counterterm on the boundary or wall. This leads to induced Chern-Simons levels (equivalently, quantum Hall conductivity) $k = 1 + 2n$ in the T-wall setup, and $k = \frac{1}{2} + n$ in the boundary setup.

## 2.3 SPT with Gravitational $\theta_g = \pi$ Protected by Time-Reversal

### 2.3.1 Basic Properties

In Euclidean signature, the gravitational theta-term is given by

$$S = \frac{i\theta_g}{384\pi^2} \int_{M_4} \text{Tr}(R \wedge R) = \frac{i\theta_g \sigma}{16} = i\theta_g I_g \; . \tag{2.27}$$

Here $\sigma$ is the signature of the four-manifold $M_4$, which is related to the $\widehat{A}(R)$-genus appearing in the Atiyah-Singer index theorem via

$$I_g = \frac{\sigma}{16} = \frac{1}{2} \int_{M_4} \widehat{A}(R) \; . \tag{2.28}$$

Here $I_g$ is the gravitational instanton number. On an oriented spin manifold, the index theorem implies that $\int_{M_4} \widehat{A}(R) \in 2\mathbb{Z}$ is an even integer (see for instance appendix A of [111]). It follows that such manifolds have $\sigma \in 16\mathbb{Z}$, so that $I_g \in \mathbb{Z}$ and the gravitational theta-angle in (2.27) has standard periodicity $\theta_g \sim \theta_g + 2\pi$.

As was the case for the $U(1)$ $\theta$-angle, invariance under an orientation-reversing symmetry, such as T or CT, quantizes

$$\theta_g = 0 \; , \qquad \theta_g = \pi \; , \tag{2.29}$$

leading to purely gravitational SPTs protected by T or CT only. The partition function of

---

[61] See [120, 52, 121–124] for a discussion of decorated domain walls.

the non-trivial $\theta_g = \pi$ SPT is given by

$$Z(\theta_g = \pi) = (-1)^{I_g} \ , \qquad I_g = \frac{\sigma}{16} \in \mathbb{Z} \ . \tag{2.30}$$

On a manifold with boundary this SPT can be integrated by parts to produce what is effectively a level-$\frac{1}{2}$ gravitational Chern-Simons term,[62] whose flip under T or CT transforms the partition function by a properly quantized gravitational Chern-Simons counterterm. This is the purely gravitational version of the parity anomaly.

### 2.3.2 Example: Massive Weyl Fermion

Consider a single massive Weyl Fermion $\psi_\alpha$ with Lagrangian

$$\mathcal{L} = -i\overline{\psi}\overline{\sigma}^\mu \partial_\mu \psi - \frac{1}{2}m(\psi\psi + \overline{\psi\psi}) \ , \qquad m \in \mathbb{R} \ . \tag{2.31}$$

Reality of the mass implies invariance under the time-reversal symmetry (see e.g. [126])

$$\mathsf{T} : \psi_\alpha(t, \vec{x}) \to \psi^\alpha(-t, \vec{x}) \ . \tag{2.32}$$

Note that a single Weyl fermion does not have charge-conjugation symmetry.[63] As before, we choose counterterms so that $m > 0$ corresponds to the trivial SPT.

In order to determine the SPT for $m < 0$ we again use a $U(1)_{\text{axial}}$ transformation, as in (2.23). The mixed anomaly between $U(1)_{\text{axial}}$ and gravity then implies that the partition function obeys

$$Z(e^{i\varphi}m) = Z(m) \exp\left(\frac{i\varphi}{2} \int_{M_4} \widehat{A}(R)\right) = Z(m)e^{\frac{i\varphi\sigma}{16}} \ . \tag{2.33}$$

Comparing with (2.27) we recognize this as a gravitational theta-angle $\theta_g = \varphi$. Setting this angle equal to $\pi$, we find that

$$Z(-m) = Z(m)(-1)^{I_g} \ , \qquad I_g = \frac{\sigma}{16} \in \mathbb{Z} \ . \tag{2.34}$$

Thus the $m < 0$ phase of the massive Weyl fermion is precisely the non-trivial gravitational SPT with $\theta_g = \pi$.

---

[62] This in turn gives rise to a thermal Hall conductance $\kappa_{xy} = 1/4 + n/2$ with $n \in \mathbb{Z}$, in units of $\pi^2 k_B^2 T/3h$, where $T$ is temperature, $h$ is Planck's constant, and $k_B$ is Boltzmann's constant. This is further discussed in section 2.3.3 below.

[63] The CPT theorem also guarantees invariance under $\mathsf{CP} : \psi_\alpha(t, \vec{x}) \to i\overline{\psi}^{\dot{\alpha}}(t, -\vec{x})$.

In summary, the massive Weyl fermion interpolates between the trivial SPT at $m > 0$, through the massless point, to the non-trivial $\theta_g = \pi$ SPT at $m < 0$.

As we did for the Dirac fermion, it is straightforward to embed this transition into an interacting model by promoting the bare mass $m$ to a dynamical real scalar field. We can then study the (generally first order) transition that occurs when the vev of the scalar jumps from large positive to large negative values.

Let us generalize the preceding discussion to multiple Weyl fermions $\psi_\alpha^I$ with time-reversal acting as in (2.32) for every value of $I$, that is $\mathsf{T} : \psi_\alpha^I(t, \vec{x}) \to \psi^{\alpha I}(-t, \vec{x})$. The most general mass term compatible with this symmetry is

$$\mathscr{L}_{\text{mass}} = -\frac{1}{2} m_{IJ}(\psi^I \psi^J + \overline{\psi}^I \overline{\psi}^J) , \qquad m_{IJ} = m_{JI} = m_{IJ}^* . \tag{2.35}$$

The real symmetric matrix $m_{IJ}$ can be diagonalized, with eigenvalues $m_I \in \mathbb{R}$, using a real orthogonal matrix. Since such an orthogonal transformation of the fields does not have a mixed anomaly with gravity, the value of the gravitational $\theta_g$-angle is entirely determined by the number (mod 2) of negative eigenvalues $m_I$, or equivalently by the sign of the determinant of the mass matrix,

$$\theta_g = \begin{cases} 0 & \text{if } \det m_{IJ} > 0 \\ \pi & \text{if } \det m_{IJ} < 0 \end{cases} \tag{2.36}$$

Applying this to the example of the free Dirac fermion in (2.16), we see that the mass matrix

$$m_{IJ} = \begin{pmatrix} 0 & m \\ m & 0 \end{pmatrix} , \tag{2.37}$$

has determinant $\det m_{IJ} = -m^2 \leq 0$. Thus the determinant never changes sign, and hence both gapped phases of the massive Dirac fermion have $\theta_g = 0$.

As discussed at the end of section 2.2.2, the $\theta_f = \pi$ SPT is protected by both $U(1)$ and time-reversal symmetry. If $\mathsf{T}$ is broken, the SPT can be trivialized. Analogous statements hold for the $\theta_g = \pi$ SPT considered here, except that this SPT is only protected by $\mathsf{T}$.

### 2.3.3 Detecting the $\theta_g = \pi$ SPT via 2+1d Thermal Hall Conductance

In analogy with section 2.2.3, we now recall how to determine the gravitational $\theta_g = \pi$ SPT using the thermal Hall conductance on codimension-one defects or boundaries. As mentioned previously, these diagnostics are particularly useful in settings without Lorentz symmetry, where it is not possible to consider partition functions on Euclidean spacetime

manifolds.

First, we consider a T-wall, which maps the SPT characterized by $\theta_g = (2n+1)\pi$ on one side of the wall to $\theta_g = -(2n+1)\pi$ on the other side. As before, the integer $n \in \mathbb{Z}$ parametrizes possible counterterm ambiguities. The net result is a jump by $\Delta\theta_g = 2\pi(2n+1)$ across the wall, which in turn implies that it harbors a gravitational Chern-Simons term whose quantized coefficient can be characterized by saying that it gives rise to a framing anomaly with chiral central charge

$$c = \frac{1}{2} + n \ . \tag{2.38}$$

This in turn induces a non-zero thermal Hall conductance on the wall,

$$\kappa_{xy} = c = \frac{1}{2} + n \ , \qquad n \in \mathbb{Z} \ . \tag{2.39}$$

This is measured in units of $\pi^2 k_B^2 T/(3h)$, where $k_B$ is the Boltzmann constant, $T$ is temperature, and $h$ is the Planck constant. Recall that the thermal hall effect involves non-dissipative heat transport in a direction transverse to the applied temperature gradient. See for instance [127–129, 125, 130, 131].

Alternatively, we can consider a symmetry-preserving boundary, where the jump in $\theta_g$ is half of what it was for the T-wall above. This leads to the following thermal Hall conductivity (equivalently, chiral central charge),

$$\kappa_{xy} = c = \frac{1}{4} + \frac{n}{2} \ , \qquad n \in \mathbb{Z} \ . \tag{2.40}$$

Examples of anomalous T-symmetric boundary states that saturate this anomaly include odd numbers of massless 2+1d Majorana fermions (as for instance reviewed in [49]), or certain gapped states with non-Abelian anyons (see e.g. [66, 132–135]). Note that Abelian anyons (which can be described by Abelian Chern-Simons gauge theories, possibly stacked with a fermionic SPT) at most give rise to half-integral $c \in \frac{1}{2}\mathbb{Z}$ and are therefore not suitable T-symmetric boundary states for the $\theta_g = \pi$ SPT in the bulk.

The previous discussion applies to boundaries that neither explicitly nor spontaneously break T. There are also boundary states that spontaneously break T, related by the action of the T-wall discussed above. This implies that the thermal Hall conductance in the two boundary vacua related by the spontaneously broken T-symmetry differs by $\frac{1}{2}+n$, with $n \in \mathbb{Z}$. In particular, at least one boundary vacuum must have nonzero thermal Hall conductivity.

## 2.4 Generalization to Weyl Fermions with Non-Abelian Flavor Symmetry

For future reference, consider a collection of massive Weyl fermions $\psi_\alpha^I$ in a real (possibly reducible) representation $r$ of a non-Abelian flavor symmetry group $G_f$. Here the index $I$ runs from $I = 1, \ldots, \dim r$, with $\dim r$ the real dimension of the representation $r$. We choose a common mass $m \in \mathbb{R}$ for all fermions, so that the mass term $m\psi\psi$ in (2.31) is replaced by $m\delta_{IJ}\psi^I\psi^J$. We now compute the jump in the SPT as we change the sign of $m$.

The generalization of (2.25) and (2.33) to this case takes the form

$$
\begin{aligned}
Z(e^{i\varphi}m) &= Z(m)\exp\left(\frac{i\varphi}{2}\int_{M_4}\mathrm{Tr}_r\left(\widehat{A}(R)e^{F/2\pi}\right)\right) \\
&= Z(m)\exp\left(\frac{i(\dim r)\varphi\sigma}{16} + \frac{i\varphi}{16\pi^2}\int_{M_4}\mathrm{Tr}_r F\wedge F\right) .
\end{aligned}
\tag{2.41}
$$

Here $F$ is the hermitian non-Abelian background field strength associated with the $G_f$ symmetry. The Dynkin index $I(r)$ of the representation $r$ is defined so that

$$
\mathrm{Tr}_r = 2I(r)\,\mathrm{Tr}_\square \ ,
\tag{2.42}
$$

where $\square$ denotes the fundamental representation of $G_f$. The conventionally normalized instanton number for the $G_f$ background field $F$ is given by

$$
I(G_f) = \frac{1}{8\pi^2}\int_{M_4}\mathrm{Tr}_\square F\wedge F .
\tag{2.43}
$$

Note that this is an integer if $G_f$ is simply connected; otherwise it may be fractional.

Specializing to $\varphi = \pi$ in (2.41), and using (2.42), (2.43), we conclude that

$$
Z(-m) = Z(m)e^{i\left((\dim r)\pi I_g + I(r)\pi I(G_f)\right)} ,
\tag{2.44}
$$

with $I_g = \sigma/16$ the gravitational instanton number. Thus flipping the mass $m \to -m$ in this model generates a gravitational theta-angle

$$
\theta_g = (\dim r)\pi \ ,
\tag{2.45}
$$

as well as an SPT protected by the non-Abelian $G_f$ symmetry and time-reversal,

$$S = i\theta_f I(G_f) = \frac{i\theta_f}{8\pi^2} \int_{M_4} \mathrm{Tr}_\square F \wedge F \ , \qquad \theta_f = I(r)\pi \ . \qquad (2.46)$$

A formula for the Dynkin index $I(r)$ for a general $SU(N)$ representation $r$ is reviewed in appendix A.

## 2.5 Gapless SPTs and Nambu-Goldstone Bosons

In this section, we have so far assumed that the SPT phases under discussion are gapped, which is the context in which they are best understood. Recently there has been considerable interest in gapless SPT phases, see for instance [79–88], and in particular the extent to which they share the characteristic rigidity of their gapped counterparts. Since we will encounter some examples of gapless SPTs in our analysis below, we collect various useful observations about them here.

### 2.5.1 Eating SPTs via 't Hooft Anomalies

An important question is whether a particular SPT phase, whose effective action only depends on background fields, can be removed by performing a field redefinition of the dynamical fields. In this case the SPT is said to be "eaten" by the dynamical degrees of freedom. One way in which the SPT can be eaten is that the symmetry protecting it is spontaneously broken, see for instance [87] for examples of this type.

Here we will only discuss SPTs for unbroken symmetries, but even those can be eaten by gapless (or long-range entangled topological) degrees of freedom. Examples of this phenomenon have been observed in [86,87,136–139], and all of them involve an 't Hooft anomaly (either microscopic or emergent) for the low energy degrees of freedom. This is natural, because such an anomaly leads to a phase ambiguity in the partition function that depends on the background fields and that can potentially absorb the SPT.

Let us sketch a partial converse of this statement: an SPT can only be eaten if its background fields participate in a mixed anomaly with some symmetry. The converse is partial, because we will assume that the SPT admits a topological boundary condition. This is not true for all SPTs, but it holds for the $\theta_g = \pi$ and $\theta_f = \pi$ SPTs that are important throughout this paper (with a suitable restriction to the Cartan torus in the flavor case) .

Symmetries are transformations of the dynamical fields that leave the theory invariant. If our theory can eat an SPT phase, then stacking the theory with that SPT phase constitutes

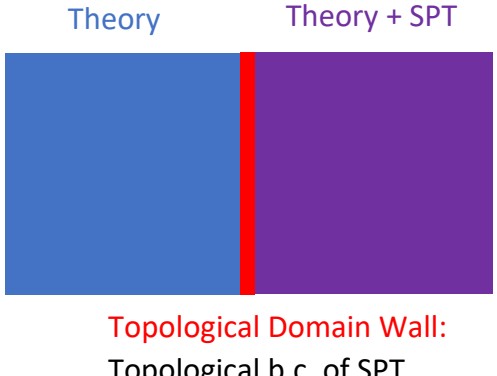

Theory          Theory + SPT

Topological Domain Wall:
Topological b.c. of SPT

Figure 6: Domain wall defect that implements eating an SPT phase in a theory that is invariant under stacking with that SPT phase. If the SPT admits a topological boundary condition, then the defect is topological and generates a zero-form symmetry that has a mixed anomaly with the background fields that participate in the SPT.

a symmetry operation. Moreover, this symmetry shifts the partition function by a phase (supplied by the SPT) and therefore it has a mixed anomaly with the background fields that participate in the SPT. The generator of this anomalous zero-form symmetry is given by a codimension-one defect that eats the SPT phase, i.e. the two sides of the wall differ by stacking our theory with the SPT. This is shown in figure 6. Since we assumed that the SPT we are stacking with admits a topological boundary condition, this SPT-eating defect is indeed topological, as expected for a symmetry generator.

### 2.5.2    Partition Functions and Gapless SPTs

One way to diagnose gapped SPT phases is via their partition function in the presence of background fields on euclidean spacetime manifolds. Here we will make some related observations in the context of gapless SPTs.[64] We distinguish three cases:

1.) Let us consider some gapless degrees of freedom whose partition function (in a suitable scheme) is strictly positive in the presence of all background fields we would like to consider,

$$Z_{\text{gapless}} > 0 \ . \tag{2.47}$$

In this case the SPT phase of the gapless degrees of freedom is well defined (in particular

---

[64] We are grateful to Ryan Thorngren for discussions about the content of this subsection, and for explaining some of the intuitive arguments underlying [87].

they cannot eat any SPT) and trivial. Stacking this gapless phase with any SPT phase produces a partition function

$$Z_{\text{gapless}} Z_{\text{SPT}} \ , \tag{2.48}$$

whose phase is entirely determined by the SPT. In other words, the SPT is not changed by stacking with gapless modes that satisfy (2.47).

2.) If the partition function of the gapless degrees of freedom is always non-vanishing (but, unlike (2.47) above, cannot be made positive in any scheme),

$$Z_{\text{gapless}} \neq 0 \ , \tag{2.49}$$

then stacking with an SPT phase leads to a different partition function,

$$Z_{\text{gapless}} Z_{\text{SPT}} \neq Z_{\text{gapless}} \ . \tag{2.50}$$

In particular, the SPT we stack with cannot be eaten by the gapless modes, but now they contribute to the phase of the partition function, and hence to the overall SPT phase. Examples of this kind in 1+1d are discussed in [87]. An example in 2+1d is a free, massless Dirac fermion, which effectively has a half-integer Chern-Simons term for its $U(1)$ symmetry that cannot be cancelled by any integer-quantized Chern-Simons SPT.

3.) Finally, the partition function of the gapless phase can vanish (at least for some choice of background fields),

$$Z_{\text{gapless}} = 0 \ . \tag{2.51}$$

Precisely this happens in the presence of a suitable anomaly. As we have explained above, this makes it possible for the gapless modes to eat SPTs, roughly because

$$Z_{\text{gapless}} Z_{\text{SPT}} = 0 = Z_{\text{gapless}} \ , \tag{2.52}$$

so that stacking with the SPT is a symmetry.

### 2.5.3 Gapless SPT Phase with a $U(1)$ Nambu-Goldstone Boson

In this paper we will encounter gapless phases that contain a NGB for a spontaneously broken flavor symmetry $U(1)_B$ with background field $B$. In addition, there can be SPTs for a distinct, unbroken $U(1)$ flavor symmetry with background field $A$, as well as gravity,

both protected by an unbroken time-reversal symmetry. Let us take the $U(1)_B$ Goldstone boson $\phi$ to be a dimensionless scalar of periodicity $\phi \sim \phi + 2\pi$. The shift symmetry of $\phi$, with current $f_\phi d\phi$ coupling to $B$, is the broken $U(1)_B$ symmetry.[65] In addition, $\phi$ has an unbroken, generally emergent, $U(1)_C^{(2)}$ two-form winding symmetry, whose current $\sim \star d\phi$ couples to a three-form gauge field $C^{(3)}$ with field strength $H^{(4)} = dC^{(3)}$. All together the low-energy Lagrangian for $\phi$ is,

$$S = \frac{f_\phi^2}{2} \int_{M_4} (d\phi - B) \wedge \star (d\phi - B) + \frac{i}{2\pi} \int_{M_4} \phi H^{(4)} \; . \tag{2.53}$$

Note that the broken $U(1)_B$ symmetry and the unbroken $U(1)_C^{(2)}$ winding symmetry have a mixed 't Hooft anomaly. In the presentation (2.53), the theory is manifestly invariant under $C^{(3)}$ background gauge transformations,[66] but a $U(1)_B$ background gauge transformation $B \to B + d\lambda_B$ shifts

$$S \to S + \frac{i}{2\pi} \int_{M_4} \lambda_B H^{(4)} \; . \tag{2.54}$$

Comparing with (2.4) we see that we can absorb a $\theta_f = \pi$ SPT by choosing

$$\lambda_B = \pi \; , \qquad \frac{1}{2\pi} H^{(4)} = \frac{1}{8\pi^2} F \wedge F \; . \tag{2.55}$$

Note that both $\frac{1}{2\pi} H^{(4)}$ and $\frac{1}{8\pi^2} F \wedge F$ have integral periods. However, substituting back into (2.54) we find a non-vanishing mixed anomaly between the broken $U(1)_B$ symmetry and the unbroken $U(1)$, in accord with the general discussion in section 2.5.1.[67] Completely analogously, comparing with (2.27), we find that we can absorb $\theta_g = \pi$ SPT by choosing

$$\lambda_B = \pi \; , \qquad \frac{1}{2\pi} H^{(4)} = \frac{1}{384\pi^2} \mathrm{Tr}(R \wedge R) \; , \tag{2.56}$$

but this comes at the cost of a mixed $U(1)_B$ anomaly with gravity.

Conversely, if the spontaneously broken $U(1)_B$ symmetry has no mixed 't Hooft anomalies with the unbroken $U(1)$ symmetry or gravity, then the $\theta_f = \pi$ and $\theta_g = \pi$ SPTs cannot be absorbed by the massless NGB, and hence they continue to be meaningful. This is precisely

---

[65] Here $f_\phi$ is the decay constant of $\phi$.

[66] If we add to the action a counterterm $\sim \int_{M_4} B \wedge C^{(3)}$ then a $C^{(3)}$ gauge transformation leads to a shift $\sim dB$. Clearly such a shift cannot eat an SPT involving only the $U(1)$ background field $A$ or gravity.

[67] In this example, the mixed 't Hooft anomaly between the $U(1)$ flavor symmetry and the broken $U(1)_B$ symmetry entirely comes from the mixed 't Hooft anomaly between the $U(1)_C^{(2)}$ two-form symmetry and the $U(1)_B$ symmetry.

the situation that we will encounter below in several examples. In these examples we will be able to check that the SPTs remain unchanged if we break the $U(1)_B$ symmetry explicitly, thereby giving a mass to the Goldstone boson. Since the SPTs are thus connected to the conventional gapped setting, this provides another argument that they are robust. The fact that we get a well-defined SPT when lifting the NGB is directly related to the absence of a mixed 't Hooft anomaly involving $U(1)_B$.

# 3 Review of Massive QCD

In this section we review some useful facts about QCD with $SU(N)_c$ gauge group and $N_f$ flavors of massive Dirac quarks in the fundamental representation of $SU(N)_c$. The quark masses explicitly break the axial symmetries of massless QCD; we focus on the unbroken vector-like symmetries.

## 3.1 Fields and Lagrangian

Rather than working with 4-component Dirac fermions, we work with pairs of 2-component Weyl fermions,

$$(\psi_\alpha)_i^a \ , \qquad (\chi_\alpha)_a^i \ , \qquad a = 1, \ldots, N \ , \qquad i = 1, \ldots, N_f \ . \tag{3.1}$$

Here $\alpha = 1, 2$ is a left-handed Weyl spinor index,[68] which we will generally suppress. The raised and lowered $a$-indices are fundamental and anti-fundamental $SU(N)_c$ color indices, respectively, so that the pair $(\psi_i^a, \chi_a^i)$ transforms as $(\mathbf{N}, \overline{\mathbf{N}})$ of the $SU(N)_c$ gauge group. Likewise, the lowered and raised $i$-indices are fundamental and anti-fundamental $SU(N_f)$ flavor indices, and $(\psi_i^a, \chi_a^i)$ transforms as $(\mathbf{N_f}, \overline{\mathbf{N_f}})$ under $SU(N_f)$.

Hermitian conjugation exchanges left-handed undotted Weyl spinors with dotted right-handed Weyl spinors; it also exchanges fundamental and anti-fundamental representations of $SU(N)_c$ and $SU(N_f)$. Thus the hermitian conjugate quark fields are

$$(\overline{\psi}_{\dot\alpha})_a^i = ((\psi_\alpha)_i^a)^\dagger \ , \qquad (\overline{\chi}_{\dot\alpha})_i^a = ((\chi_\alpha)_a^i)^\dagger \ . \tag{3.2}$$

---

[68] We use the conventions of Wess and Bagger [65] for 2-component Weyl spinors. See [74] for a presentation of QCD in these conventions.

The conventional 4-component Dirac quarks are then given by

$$(\Psi_D)_i^a = \left((\psi_\alpha)_i^a \, , \, (\overline{\chi}^{\dot\alpha})_i^a\right) \, . \tag{3.3}$$

In this paper we will mostly discuss the case of $N_f$ degenerate quarks, with a common real Dirac mass $m_q \in \mathbb{R}$, and vanishing theta-angle, $\theta_c = 0$, for the dynamical $SU(N)_c$ gauge field.[69] The Lagrangian in Lorentzian signature is then given by

$$\mathscr{L} = -\frac{1}{4g^2} f_{\mu\nu}^A f^{A\mu\nu} - i\overline{\psi}_a^i \overline{\sigma}^\mu D_\mu \psi_i^a - i\overline{\chi}_i^a \overline{\sigma}^\mu D_\mu \chi_a^i - m_q \left(\psi_i^a \chi_a^i + \overline{\psi}_a^i \overline{\chi}_i^a\right) \, , \qquad m_q \in \mathbb{R} \, . \tag{3.4}$$

Here $f_{\mu\nu}^A$ is the $SU(N)_c$ field strength (with $A = 1, \ldots, N^2 - 1$ an $SU(N)_c$ adjoint index) and $D_\mu$ is the $SU(N)_c$ covariant derivative in the appropriate representation.

## 3.2 Continuous Symmetries

The continuous gauge and global symmetries that act faithfully on the quark fields are[70]

$$\frac{SU(N)_c \times U(N_f)_f \times \mathrm{Spin}(4)_{\mathrm{Lorentz}}}{\mathcal{I}} \, . \tag{3.5}$$

Here $SU(N)_c$ is the gauge group and $U(N_f)_f$ is the vector-like flavor symmetry preserved by the quark mass.[71] Note that

$$U(N_f) = \frac{U(1)_Q \times SU(N_f)}{\mathbb{Z}_{N_f}} \, , \tag{3.6}$$

where $U(1)_Q$ is a quark-number symmetry with charge assignments

$$Q(\psi_i^a) = -Q(\chi_a^i) = 1 \, . \tag{3.7}$$

The quotient $\mathcal{I}$ in (3.5) implements various identifications ensuring that the symmetry action is faithful. Thus it determines the precise form of the global symmetry group:

- The center of gauge group $\mathbb{Z}_N \subset SU(N)_c$ is identified with the central $\mathbb{Z}_N$ subgroup

---

[69] An anomalous $U(1)_{\mathrm{axial}}$ transformation can be used to eliminate $\theta_c$ in favor of a (generally) complex mass $m_q \in \mathbb{C}$. Below we will additionally impose time-reversal symmetry, which requires real $m_q \in \mathbb{R}$.

[70] See for instance section 4.1 of [114] for a related discussion.

[71] Whenever we wish to highlight that something pertains to color or flavor we will use the subscripts $c$ and $f$ respectively, though we will omit them if confusion is unlikely to arise.

of the $U(N_f)$ flavor symmetry. Ignoring spacetime symmetries, the internal flavor symmetry is therefore always given by

$$G_f = \frac{U(N_f)}{\mathbb{Z}_N} \ . \tag{3.8}$$

The central subgroup of $G_f$ is the properly normalized baryon number symmetry $U(1)_B$, under which gauge-invariant baryon operators (constructed out of $N$ suitably color-antisymmetrized quarks or anti-quarks) have integer charge, $B \in \mathbb{Z}$. Note that the $\mathbb{Z}_N$ quotient in (3.8) correctly relates quark- and baryon-numbers,

$$B = \frac{1}{N}Q \ . \tag{3.9}$$

- Fermion parity $(-1)^F$ generates a central $\mathbb{Z}_2^F \subset \mathrm{Spin}(4)_{\mathrm{Lorentz}}$. If the number $N$ of colors is even, then $(-1)^F$ is identified with the central element $(-1) \in SU(N)_c$ of the gauge group, i.e. fermion number is gauged. In this case the full connected global symmetry is given by

$$G_f \times SO(4)_{\mathrm{Lorentz}} = \frac{U(N_f)}{\mathbb{Z}_N} \times SO(4)_{\mathrm{Lorentz}} \qquad (N \text{ even}) \ . \tag{3.10}$$

Such a theory is bosonic: even though it is formulated using fermionic quark fields, all gauge invariant local operators are bosons and the theory can be studied on non-spin manifolds.[72]

- If the number $N$ of colors is odd, then the theory is fermionic, i.e. there are gauge-invariant local operators that carry fermion number $(-1)^F$ and (in the absence of twists, see below) the theory can only be formulated on manifolds with a spin structure. Note however that $(-1)^F$ is identified with the central element $(-1) \in G_f$ of the flavor symmetry group, so that the full global symmetry is

$$\frac{G_f \times \mathrm{Spin}(4)_{\mathrm{Lorentz}}}{\mathbb{Z}_2^F} = \frac{U(N_f)/\mathbb{Z}_N \times \mathrm{Spin}(4)_{\mathrm{Lorentz}}}{\mathbb{Z}_2^F} \ . \tag{3.11}$$

---

[72] This utilizes the symmetry group $(SU(N)_c \times \mathrm{Spin}(4)_{\mathrm{Lorentz}})/\mathbb{Z}_2^F$ that arises when $N$ is even. On oriented Riemannian manifolds $M_4$ without a spin structure, the second Stiefel-Whitney class $w_2(M_4) \in H^2(M_4, \mathbb{Z}_2)$ of the tangent bundle is non-trivial, while the gauge fields are connections on an $SU(N)_c/\mathbb{Z}_2$ bundle. The $\mathbb{Z}_2^F$ quotient in the symmetry group implies that the class that measures the obstruction to lifting the $SU(N)_c/\mathbb{Z}_2$ gauge bundle to an $SU(N)_c$ bundle is precisely given by $w_2(M_4)$.

In particular this implies that $(-1)^F$ is identified with $(-1) \in U(1)_B \subset G_f$, so that fundamental baryons and anti-baryons with $B = \pm 1$ are necessarily fermions. This is consistent with the fact that they contain an odd number $N$ of constituent quarks. This identification further implies that the theory can be placed on non-spin manifolds, as long as the $U(1)_B$ background gauge field is taken to be a $\text{spin}_c$ connection.[73]

## 3.3  Discrete Symmetries

The QCD Lagrangian (3.4) is invariant under discrete C, P, and T symmetries. The way these symmetries act on the quark fields is identical to that described in section 2.2.2 for free Dirac fermions, and we repeat it here:

$$
\begin{aligned}
&\mathsf{C} : (\psi_\alpha)^a_i \leftrightarrow (\chi_\alpha)^i_a \ , \\
&\mathsf{T} : (\psi_\alpha)^a_i(t, \vec{x}) \to (\psi^\alpha)^a_i(-t, \vec{x}) \ , \qquad (\chi_\alpha)^i_a(t, \vec{x}) \to (\chi^\alpha)^i_a(-t, \vec{x}) \ , \\
&\mathsf{P} : (\psi_\alpha)^a_i(t, \vec{x}) \to i(\overline{\chi}^{\dot\alpha})^a_i(t, -\vec{x}) \ , \qquad (\chi_\alpha)^i_a(t, \vec{x}) \to i(\overline{\psi}^{\dot\alpha})^i_a(t, -\vec{x}) \ .
\end{aligned}
\tag{3.12}
$$

The time-reversal and parity symmetries satisfy

$$
\mathsf{T}^2 = \mathsf{P}^2 = (-1)^F \ .
\tag{3.13}
$$

We also observe that both T and P preserve the $U(1)_B$ baryon number charge $B$. Thus QCD at finite baryon chemical potential $\mu_B$ preserves T and P, but breaks C and thus also CPT. Note that the CPT theorem does not hold, because Lorentz invariance is broken in the presence of a chemical potential.

The actions of the discrete symmetries on vector-like gauge fields $A$, which could be either the dynamical $SU(N)_c$ color gauge fields, or the $U(N_f)$ flavor background gauge fields, are generalizations of the action on $U(1)$ gauge fields in section 2.2.1. If we represent $A$ by Hermitian matrices, then

$$
\begin{aligned}
&\mathsf{C} : A \to -A^* \ , \\
&\mathsf{T} : A_0(t, \vec{x}) \to A_0^*(-t, \vec{x}) \ , \qquad \vec{A}(t, \vec{x}) \to -\vec{A}^*(-t, \vec{x}) \ , \\
&\mathsf{P} : A_0(t, \vec{x}) \to A_0(t, -\vec{x}) \ , \qquad \vec{A}(t, \vec{x}) \to -\vec{A}(t, -\vec{x}) \ .
\end{aligned}
\tag{3.14}
$$

---

[73] In general this leads to additional information not accessible on spin manifolds, see e.g. [111, 140–143].

## 3.4 Vafa-Witten Positivity

When the quark mass $m_q > 0$ is positive, the QCD path integral measure can be regulated in a way that makes the path-integral measure positive definite [71–73].[74] We refer to this as Vafa-Witten positivity. It has several important consequences:

- The vector-like $G_f$ flavor symmetry is not spontaneously broken [72].

- Parity P, and hence also CT, are not spontaneously broken [73].

- Vafa-Witten positivity continuous to hold in the presence of gravitational background fields, as well as background gauge fields for the vector-like flavor symmetry $G_f$. QCD with positive quark mass $m_q > 0$ is gapped, and positivity in the presence of background fields implies that it is also a trivial SPT phase. See [74] for a related detailed discussion.

Note that ensuring Vafa-Witten positivity in the presence of background fields requires a particular choice of SPT counterterms.[75] A more invariant statement is that QCD is in the same SPT phase for any $m_q > 0$. By choosing the counterterms in a particular way we can take it to be the trivial SPT, but this is not strictly necessary.

Below we will deform QCD by adding Yukawa couplings or turning on a chemical potential $\mu_B$ for the $U(1)_B$ symmetry. Both of these deformations invalidate Vafa-Witten positivity, and hence open the door to potentially non-trivial SPT phases.

# 4 $SU(2)$ Higgs-Yukawa-QCD

In this section we elaborate on the example introduced in section 1.2.3, which is motivated by the classic analysis of Higgs-confinement continuity in [24].

## 4.1 Defining the Theory

The model is a deformation of QCD with $N = 2$ colors and $N_f = 1$ massive Dirac quark flavor in the doublet representation of the $SU(2)_c$ gauge group, i.e. two $SU(2)_c$ fundamental Weyl quarks,

$$(\psi^a)_\alpha \ , \qquad (\chi_a)_\alpha \ . \tag{4.1}$$

---

[74] Following these references, we work in the presence of a UV cutoff $\Lambda_{\mathrm{UV}}$ and take $m_q$ to be the bare (rather than a renormalized or physical) quark mass.

[75] This is in addition to the requirement that we set the theta-angle for the dynamical gauge fields to zero, $\theta_c = 0$, as we have done throughout.

Here $\alpha = 1, 2$ is a left-handed Weyl spinor index, and $a = 1, 2$ is a fundamental $SU(2)_c$ color index. According to the discussion in section 3, the continuous symmetry of this theory is

$$U(1)_B \times SO(4)_{\text{Lorentz}} \ . \tag{4.2}$$

Additionally there are the discrete $\mathsf{C}$, $\mathsf{T}$, $\mathsf{P}$ symmetries. Note that $U(1)_B$ baryon number is normalized so that

$$B(\psi) = -B(\chi) = \frac{1}{2} \ . \tag{4.3}$$

This implies that all gauge-invariant local operators have $B \in \mathbb{Z}$. Moreover, the baryons of this theory, e.g. $\varepsilon_{ab}\psi^a\psi^b$ with $B = 1$, are bosons. In fact the entire theory is bosonic, because fermion number $(-1)^F$ is an element of the $SU(2)_c$ gauge group, and hence it can in principle be studied on spacetime manifolds without a spin structure (though we will not do so here), in accord with the general discussion in Section 3.2 for even $N$.

The presentation above (which applies for general $N$) obscures the full global symmetry of the $N = 2$ theory we are considering here. Since bot $\psi^a$ and $\chi_a$ transform in the pseudo-real doublet representation of $SU(2)_c$, we can use the $SU(2)_c$ invariant symbols $\varepsilon^{ab}, \varepsilon_{ab}$ to raise and lower the gauge indices,[76] putting the left- and right-handed quarks on equal footing:

$$\psi^a_{i=1} = \psi^a \ , \qquad \psi^a_{i=2} = \chi^a \ . \tag{4.4}$$

The fermions $\psi^a_i$ transform as $(\mathbf{2}, \mathbf{2})$ under the following symmetry group,

$$\frac{SU(2)_c \times SU(2)_f}{\mathbb{Z}_2} = SO(4)_{c,f} \ . \tag{4.5}$$

Here $SU(2)_f$ is a flavor symmetry under which $i = 1, 2$ is a doublet index. The $\mathbb{Z}_2$ quotient indicates that the central element of $(-1) \in SU(2)_f$ is gauged, so that the actual global symmetry of the model is

$$G_f \times SO(4)_{\text{Lorentz}} \ , \qquad G_f = \frac{SU(2)_f}{\mathbb{Z}_2} = SO(3)_f \ . \tag{4.6}$$

The $U(1)_B$ baryon number symmetry in (4.2) and (4.3) is the Cartan subgroup of $G_f$, and the charge-conjugation symmetry $\mathsf{C}$ exchanging $\psi = \psi_{i=1}$ and $\chi = \psi_{i=2}$ is the associated Weyl reflection. Note that the $SU(2)_c$ gauge theory has no further notion of charge-conjugation

---

[76] We use the same conventions for $SU(2)$ index raising and lowering as for 2-component Weyl spinors.

symmetry, so that it is sufficient to enforce $\mathsf{T}$ (or equivalently $\mathsf{CP}$) symmetry,

$$\mathsf{T} : (\psi_i^a)_\alpha(t, \vec{x}) \to (\psi_i^a)^\alpha(-t, \vec{x}) . \tag{4.7}$$

As before this implies that $\mathsf{T}^2 = (-1)^F$ so that the fermions are Kramers doublets.

Let us add a Dirac mass term for $\psi^a, \chi_a$, which is manifestly $SU(2)_f$ invariant when expressed in terms of $\psi_i^a$,

$$\mathscr{L}_{\text{mass}} = -\frac{m_q}{2} \varepsilon_{ab} \varepsilon^{ij} \psi_i^a \psi_j^b . \tag{4.8}$$

We also impose time reversal $\mathsf{T}$, which requires $m_q \in \mathbb{R}$. According to the discussion at the end of section 3.4, we can choose counterterms and regulate the theory in such a way that the $m_q > 0$ phase is a trivially gapped SPT phase.

A explained in section 1, theories with a gauged $(-1)^F$ symmetry (such as a conventional electronic superconductor) can never exhibit Higgs-confinement continuity, because Lorentz-scalar Higgs fields can at most break the gauge symmetry down to the $\mathbb{Z}_2^F$ subgroup generated by $(-1)^F$. Following [24], we therefore add a complex Lorentz-scalar Higgs field in the $SU(2)_c$ fundamental representation, so that $(-1)^F$ is no longer gauged and we can completely break the gauge group,

$$h^a , \qquad \overline{h}_a = (h^a)^\dagger . \tag{4.9}$$

We can reorganize the complex 2-component Higgs doublet into a real 4-component field

$$h_i^a , \qquad (h_i^a)^\dagger = h_a^i . \tag{4.10}$$

Here we choose $i = 1, 2$ to be a doublet index under the $SU(2)_f$ flavor symmetry acting on the fermions.[77] Then $h_i^a$ transforms in the real, fundamental vector representation of the gauge and global $SO(4)_{c,f}$ group in (4.5). This is similar to the Higgs field in the standard model, with $SU(2)_c$ analogous to the weak gauge group and $SU(2)_f$ playing the role of the custodial symmetry. We also take $h_i^a$ to transform as a scalar under time-reversal,

$$\mathsf{T} : h_i^a(t, \vec{x}) \to h_i^a(-t, \vec{x}) , \tag{4.11}$$

in accord with the relation $\mathsf{T}^2 = (-1)^F$.

---

[77] A priori, the $SU(2)$ flavor symmetry acting on the Higgs field is decoupled from the $SU(2)_f$ symmetry acting on the fermions, but in section 4.2 below we add Yukawa couplings that identify them.

Finally, we add a T- and $SO(4)$-invariant quartic potential for the Higgs field,

$$V(h) = M_h^2 |h|^2 + \lambda |h|^4 , \qquad \lambda > 0 , \qquad M_h^2 \in \mathbb{R} . \tag{4.12}$$

Here $|h|^2 = h_i^a h_a^i$ is the invariant length of the Higgs field.

## 4.2 SPT-Enforced Higgs-Confinement Transitions from Yukawa Couplings

A natural way to avoid the Higgs-confinement continuity in [24] is to evade the Vafa-Witten theorem by adding Yukawa couplings to the theory. As we will show, this can lead to non-trivial SPT phases, separated by phase transitions that violate Higgs-confinement continuity.

The field content of our model does not allow for renormalizable Yukawa couplings, but it does have dimension-5 Yukawa couplings that preserve all gauge and global symmetries,[78]

$$\mathscr{L}_{\text{Yukawa}} = \frac{1}{2} \left( y_1 h_a^i h_b^j + y_2 h_a^j h_b^i \right) \psi_i^a \psi_j^b + (\text{h.c.}) , \qquad y_1, y_2 \in \mathbb{R} . \tag{4.13}$$

Here the Yukawa couplings $y_1, y_2$ have dimension of inverse mass, e.g. we could take $y_1, y_2 \sim \frac{\epsilon}{\Lambda}$, with $\epsilon$ a small dimensionless coupling. Time-reversal symmetry T requires $y_1, y_2 \in \mathbb{R}$ to be real. As we will see the signs of $y_1, y_2$ are important; indeed there is no symmetry that allows us to flip these signs.

The analysis of the confining phase proceeds exactly as before: $h_a^i$ has no vev there, and the Yukawas (4.13) are irrelevant operators with a small coefficient. We do not expect them to significantly affect the confining dynamics of the $SU(2)_c$ gauge theory, which as before leads to a gapped and trivial SPT phase.

By contrast, in the Higgs phase $h_i^a$ acquires the vev

$$h_i^a = v \delta_i^a , \qquad v > 0 . \tag{4.14}$$

The Yukawa couplings now contribute to the fermion mass matrix $m_{IJ}$, where $I, J = 1, 2, 3, 4$

---

[78] These couplings are reminiscent of the dimension-5 Weinberg operators in the standard model.

are $SO(4)_{c,f}$ vector indices,

$$
\begin{aligned}
-\frac{1}{2}m_{IJ}\psi^I\psi^J &= \frac{1}{2}\left(-m_q\varepsilon_{ab}\varepsilon^{ij} + y_1 v\delta_a^i\delta_b^j + y_2 v\delta_a^j\delta_b^i\right)\psi_i^a\psi_j^b \\
&= \frac{1}{2}\left((m_q + y_1 v)\delta_a^i\delta_b^j + (-m_q + y_2 v)\delta_a^j\delta_b^i\right)\psi_i^a\psi_j^b .
\end{aligned}
\tag{4.15}
$$

In order to switch from the $\psi_i^a$ to the $\psi^I$ basis, we use the 4-dimensional Euclidean sigma-matrices,

$$
(\sigma_E^I)^a{}_i = ((\vec{\tau})^a{}_i, i\delta^a{}_i) , \qquad I = 1,2,3,4 ,
\tag{4.16}
$$

where the extra factor of $i$ for the 4th component is due to Wick rotation. Here $(\vec{\tau})^a{}_i$ stands for the standard Pauli matrices $(\tau^I)^a{}_i$, with $I = 1,2,3$. We then make the change of variables[79]

$$
\psi^a{}_i = \frac{1}{\sqrt{2}}(\sigma_E^I)^a{}_i\psi^I .
\tag{4.17}
$$

Note that we have started to assign horizontal slots to color and flavor indices, since they can now be identified, and this can lead to ambiguities. Substituting into (4.15) we find

$$
\begin{aligned}
-\frac{1}{2}m_{IJ}\psi^I\psi^J &= -\frac{1}{2}(m_q + 2y_1 v + y_2 v)\psi^4\psi^4 + \frac{1}{2}(-m_q + y_2 v)\frac{1}{2}(\tau^I)^a{}_i(\tau^J)^b{}_j\delta_a^j\delta_b^i\psi^I\psi^J \\
&= -\frac{1}{2}(m_q + 2y_1 v + y_2 v)\psi^4\psi^4 + \frac{1}{2}(-m_q + y_2 v)\sum_{I=1}^{3}\psi^I\psi^I .
\end{aligned}
\tag{4.18}
$$

Thus the mass eigenvalues are

$$
M_{\mathbf{1}} = m_q + (2y_1 + y_2)v , \qquad M_{\mathbf{3}} = m_q - y_2 v ,
\tag{4.19}
$$

with degeneracy $\mathbf{1}$ and $\mathbf{3}$ respectively, which also indicates their transformation properties under $SU(2)_f'$. As a consistency check, turning off the Yukawa couplings leads to the common Dirac quark mass $m_q$.

Thus, as we dial $M_h^2$ to larger negative values, thereby increasing the Higgs vev $v$, we find – depending on the signs of $y_1, y_2$ – that either the singlet fermion mass $M_{\mathbf{1}}$ or the triplet fermion mass $M_{\mathbf{3}}$ (or both) can flip sign. Deep in the Higgs phase we are at weak coupling and our semiclassical analysis is reliable:

- If only $M_{\mathbf{1}}$ flips sign, the gravitational SPT jumps to $\theta_g = \pi$, but the flavor SPT remains

---

[79] It can be checked that the matrix $(\sigma_E^I)^a{}_i$ affecting the change of variables is an $SO(4)$ matrix, i.e. it is orthogonal of unit determinant.

trivial.

- If only $M_\mathbf{3}$ flips sign, we get a jump to $\theta_g = 3\pi$, which is equivalent to $\pi$ modulo $2\pi$. Because the triplet transforms in the adjoint of $SU(2)'_f$, it also follows from (2.46) that the flavor SPT jumps to

$$\theta_f = 2\pi \; . \tag{4.20}$$

Note that this is not the trivial SPT, because the flavor symmetry is $SO(3)'_f = SU(2)'_2/\mathbb{Z}_2$, so that the instanton number on spin manifolds can be half-integer,[80]

$$I(SO(3)'_f) \in \frac{1}{2}\mathbb{Z} \; , \tag{4.21}$$

Thus $\theta_f \sim \theta_f + 4\pi$ and $\theta_f = 2\pi$ is the non-trivial, $\mathsf{T}$-invariant SPT at the midpoint.

- If both $M_\mathbf{1}$ and $M_\mathbf{3}$ flip sign, then $\theta_g = 0$ and $\theta_f = 2\pi$, so we only have the flavor SPT.

If the signs of the Yukawa couplings are such that there is a non-trivial jump in the SPT, the passage from confinement at large $M_h^2 > 0$ to Higgsing at large $M_h^2 < 0$ will not be continuous: there must be a phase transition. In this sense the Yukawa couplings destroy Higgs-confinement continuity, but this only works because of time-reversal symmetry. As already mentioned in section 1.2.1, explicitly breaking $\mathsf{T}$ allows the Yukawa couplings $y_1, y_2 \in \mathbb{C}$ to be complex, so that we can interpolate between the SPTs and avoid the massless fermion points by dialing through complex values of $M_\mathbf{1}, M_\mathbf{3} \in \mathbb{C}$.

# 5 $SU(3)$ Higgs-Yukawa-QCD

In this section we consider QCD with $SU(3)_c$ gauge group and $N_f = 3$ quark flavors. We mostly focus on the degenerate case with equal mass $m_q$ for all quarks and $SU(3)_f$ flavor symmetry, but in section 5.4 we explicitly comment on the fate of our results when $SU(3)_f$ is broken by more general quark masses.

We add an elementary Higgs field $h_a^i$ in the $(\overline{\mathbf{3}}, \overline{\mathbf{3}})$ representation of $SU(3)_c \times SU(3)_f$, as well as Yukawa couplings to the quarks, whose particular properties – notably invariance under parity $\mathsf{P}$ and time-reversal $\mathsf{T}$ – are motivated by applications to QCD at finite $U(1)_B$ baryon-number density (see section 6 below). The goal of this section is twofold:

---

[80] Recall that the introduction of the Higgs field turned the previously bosonic $SU(2)_c$ QCD theory into a theory that requires a spin structure.

1.) To explore the confining and Higgs phases of the model. The former is gapped and trivial, while the latter spontaneously breaks $U(1)_B$. Additionally, the Higgs phase harbors non-trivial SPT phases for $SU(3)_f$ and gravity (protected by P and T) that we determine. We also explain why the SPT phases safely co-exist with the $U(1)_B$ NGB.

2.) To present another example in which naive Higgs-confinement continuity is thwarted by an SPT jump between the two phases. To this end we add an explicit $U(1)_B$-breaking perturbation to gap out the NGB in the Higgs phase, rendering it naively indistinguishable from the confining phase. Nevertheless, the non-trivial SPTs in the Higgs phase enforce at least one Higgs-confinement phase transition.

## 5.1   QCD with $N = 3$ Colors and $N_f = 3$ Flavors

We will consider QCD with three degenerate, massive quark flavors, i.e. $SU(N)_c$ gauge theory with $N = 3$ colors and $N_f = 3$ Dirac quark flavors. As in section 3, we represent the Dirac quarks by pairs of 2-component Weyl fermions,

$$(\psi_\alpha)_i^a , \qquad (\chi_\alpha)_a^i , \qquad a = 1, \ldots, N = 3 , \qquad i = 1, \ldots, N_f = 3 . \tag{5.1}$$

The QCD Lagrangian for general $N$ and $N_f$ appears in (3.4). We repeat here the Dirac masses for the quarks,

$$\mathscr{L}_{\text{QCD}} \supset -m_q \left( \psi_i^a \chi_a^i + \overline{\psi}_a^i \overline{\chi}_i^a \right) , \qquad m_q \in \mathbb{R} . \tag{5.2}$$

The reality of $m_q$ is enforced by time-reversal symmetry T and also by parity P, whose action on the quarks is spelled out in (3.12).

The vector-like flavor symmetry preserved by the common quark mass is (see section 3.2)

$$G_f = \frac{U(3)_f}{\mathbb{Z}_3} = U(1)_B \times PSU(3)_f . \tag{5.3}$$

Here $U(3)_f = (U(1)_Q \times SU(3)_f)/\mathbb{Z}_3$, where $Q(\psi_i^a) = -Q(\chi_a^i) = 1$ is quark number, and the $\mathbb{Z}_3$ quotients lead to

$$\frac{U(1)_Q}{\mathbb{Z}_3} = U(1)_B , \qquad \frac{SU(3)_f}{\mathbb{Z}_3} = PSU(3)_f . \tag{5.4}$$

The statement that the standard Gell-Mann $SU(3)_f$ flavor symmetry of QCD is actually $PSU(3)_f$ reflects the fact that all gauge-invariant local operators that can be constructed

out of quark fields do not transform under the shared $\mathbb{Z}_3$ center of the $SU(3)_c$ color group and the $SU(3)_f$ flavor symmetry.

Note that $PSU(3)$ symmetry allows more general background fields than $SU(3)$. In particular, the $PSU(3)$ instanton number can be fractional,[81] even on spin manifolds, where

$$I(PSU(3)_f) \in \frac{1}{3}\mathbb{Z} \ . \tag{5.5}$$

It follows that the flavor theta-angle $\theta_f$ has periodicity $6\pi$, rather than $2\pi$,

$$\theta_f I(PSU(3)_f) \ , \qquad \theta_f \sim \theta_f + 6\pi \ , \tag{5.6}$$

which in turn implies that the T- and P-invariant SPTs are

$$\theta_f I(PSU(3)_f) \ , \qquad \theta_f = 0, 3\pi \ (\text{mod } 6\pi) \ . \tag{5.7}$$

A pleasant simplification is that T and P symmetry enables us to unambiguously detect these SPTs by restricting to genuine $SU(3)_f$ background gauge fields, for which $I(SU(3)_f) \in \mathbb{Z}$ and the periodicity of $\theta_f$ is reduced from $6\pi$ to $2\pi$. There is thus a bijection,

$$\theta_f I(PSU(3)_f) \ , \quad \theta_f = 0, 3\pi \ (\text{mod } 6\pi) \qquad \longleftrightarrow \qquad \theta_f I(SU(3)_f) \ , \quad \theta_f = 0, \pi \ (\text{mod } 2\pi) \ , \tag{5.8}$$

which identifies the trivial $\theta_f = 0$ SPTs on both sides, as well as the two non-trivial SPTs: $\theta_f = 3\pi \ (\text{mod } 6\pi) \longleftrightarrow \theta_f = \pi \ (\text{mod } 2\pi)$. To simplify the presentation, we will mostly consider $SU(3)_f$ background gauge fields for the remainder of the paper, so that $\theta_f \sim \theta_f + 2\pi$ has standard periodicity.

Let us summarize the behavior of three-flavor QCD as a function of the real Dirac quark mass $m_q \in \mathbb{R}$ (see for instance [78] for a detailed discussion with references).

- $m_q > 0$: the theory is fully gapped. Moreover, it enjoys Vafa-Witten positivity (see section 3.4). This means that it can be regulated in such a way that the Euclidean path integral measure is positive. This involves setting all theta angles ($\theta_c$ for the dynamical $SU(3)_c$ gauge fields and $\theta_f, \theta_g$ for flavor and gravity background fields) to zero. The Euclidean partition function on any four-manifolds $M_4$ is then also positive, so that the theory is a trivial SPT for all positive quark masses $m_q > 0$.

---

[81] A $PSU(3)$ background gauge field with fractional instanton number can be thought of as an $SU(3)$ configuration with non-trivial 't Hooft flux.

- $m_q < 0$: If the quark mass is sufficiently large and negative, $m_q \ll -\Lambda_{\text{QCD}}$ (where $\Lambda_{\text{QCD}}$ is the $SU(3)_c$ strong-coupling scale), then we can safely integrate out the quarks and flow to pure $SU(3)_c$ gauge theory with $\theta_c = \pi$ for the dynamical gauge fields. This theory is expected to spontaneously break $\mathsf{T}$ and $\mathsf{P}$, resulting in two degenerate, gapped, confining vacua (see for instance [144] and references therein). Consequently, SPTs protected by $\mathsf{T}$ and $\mathsf{P}$, such as the $\theta_f, \theta_g = \pi$ SPTs we have been considering, are not meaningful in this phase (see section 2).

## 5.2 Higgs Fields, Yukawa Couplings, and Color-Flavor Locking

We now add to $N = N_f = 3$ QCD an elementary Higgs field $h_a^i$ in the $(\overline{\mathbf{3}}, \mathbf{3})$ representation of $SU(3)_c \times SU(3)_f$. Here $h_a^i$ has precisely the same quantum numbers as the following diquark operator,

$$h_a^i \sim \varepsilon_{abc}\varepsilon^{ijk}\left(\psi_j^b \psi_k^c + \overline{\chi}_j^b \overline{\chi}_k^c\right) , \tag{5.9}$$

e.g. it has $U(1)_B$ baryon number

$$B(h_a^i) = \frac{2}{3} . \tag{5.10}$$

Comparing with (3.12), we also deduce that $h_a^i$ is a scalar under $\mathsf{P}$ and $\mathsf{T}$,

$$\mathsf{P} : h_a^i(t, \vec{x}) \rightarrow h_a^i(t, -\vec{x}) , \qquad \mathsf{T} : h_a^i(t, \vec{x}) \rightarrow h_a^i(-t, \vec{x}) . \tag{5.11}$$

Note that the anti-unitary symmetry $\mathsf{T}$ complex conjugates vevs, e.g. $\mathsf{T} : \langle h_a^i \rangle \rightarrow \langle h_a^i \rangle^*$.

Consider a Higgs-Yukawa deformation of QCD defined by the following Lagrangian,

$$\mathscr{L}_{\text{Higgs-Yukawa-QCD}} = \mathscr{L}_{\text{QCD}} + \mathscr{L}_{\text{Higgs}} + \mathscr{L}_{\text{Yukawa}} . \tag{5.12}$$

Here $\mathscr{L}_{\text{QCD}}$ is the QCD Lagrangian in (3.4), while

$$\mathscr{L}_{\text{Higgs}} = -D_\mu \overline{h}_i^a D^\mu h_a^i - V(h) , \tag{5.13}$$

with $V(h)$ a suitable Higgs potential to be discussed below. The Yukawa couplings are

$$\mathscr{L}_{\text{Yukawa}} = y\varepsilon_{abc}\varepsilon^{ijk}\overline{h}_i^a \left(\psi_j^b \psi_k^c + \overline{\chi}_j^b \overline{\chi}_k^c\right) + (\text{h.c.}) , \qquad y > 0 , \tag{5.14}$$

so that the entire Lagrangian is invariant under the gauge and flavor symmetries. Note that the Yukawa couplings preserve $\mathsf{P}$ and $\mathsf{T}$ (and hence also $\mathsf{C}$) as long as $y \in \mathbb{R}$. In fact we can say more: prior to adding the Yukawa coupling, the $U(1)_B$ symmetry acting on the

fermions is not tied to the symmetry that rotates the Higgs field $h_a^i$ by an overall phase. We are thus free to use this symmetry to rotate $y$ by a phase, which is therefore of no physical consequence. Using this freedom, we take the Yukawa coupling constant to be positive,[82]

$$y > 0 \ . \tag{5.15}$$

Let us finally discuss the Higgs potential $V(h)$ in (5.13). We consider all terms compatible with gauge and global symmetries, up to quartic order,

$$V(h) = M_h^2 \left( \bar{h}_i^a h_a^i \right) + \lambda_S \, \bar{h}_i^a h_a^j \, \bar{h}_j^b h_b^i + \lambda_D \left( \bar{h}_i^a h_a^i \right)^2 \ . \tag{5.16}$$

Here the Higgs mass-squared $M_h^2 \in \mathbb{R}$ is arbitrary real parameter, while $\lambda_S, \lambda_D \in \mathbb{R}$ are single- and double-trace quartic couplings, respectively. The same class of scalar potentials were analyzed in [145, 146], where it was shown that imposing suitable bounds on $\lambda_S, \lambda_D$ leads to a potential that is bounded from below and triggers a color-flavor locking (CFL) Higgs vev if we dial sufficiently deeply into the weakly-coupled Higgs phase by taking $M_h^2 \ll -\Lambda_{\text{QCD}}^2$,[83]

$$\langle h_a^i \rangle = v \delta_a^i \ , \qquad v \in \mathbb{C} \ , \qquad |v|^2 = -\frac{M_h^2}{2(\lambda_S + 3\lambda_D)} \ . \tag{5.18}$$

By dialing the couplings $\lambda_{S,D}$, or higher-order terms in the potential, we could engineer other vevs for $h_a^i$.[84] The reason we focus on the CFL vev is that it characterizes the symmetry-breaking pattern that arises in high-density QCD, to be discussed in section 6.

Let us describe this symmetry-breaking pattern, which is triggered by the CFL vev (5.18), in more detail:

- The $SU(3)_c$ gauge symmetry is completely Higgsed at the scale $|v|$; the gluons acquire

---

[82] The fact that the sign (and more generally the phase) of $y$ has no physical consequences distinguishes this model from the $SU(2)_c$ gauge theory example analyzed in section 4. There the presence of certain SPTs was directly tied to the signs of the Yukawas $y_1, y_2$ in (4.13).

[83] Specifically, we must impose (see appendix A of [145] and section 4.2 of [146])

$$\lambda_S > 0 \ , \qquad \lambda_S + \lambda_D > 0 \ , \qquad \lambda_S + 3\lambda_D > 0 \ . \tag{5.17}$$

In principle $\lambda_D$ can have either sign; once the sign is fixed, either the second or the third inequality is redundant.

[84] For instance, we could take $\langle h_a^i \rangle$ to be of rank two,

$$\langle h_a^i \rangle = \text{diag}(v_1, v_2, 0) \ , \qquad v_1, v_2 \neq 0 \ , \tag{5.19}$$

which completely Higgses the $SU(3)_c$ gauge group without spontaneously breaking the $U(1)_B$ symmetry (since $\det(h_a^i) = 0$). It does however spontaneously break part of the $SU(3)_f$ flavor symmetry.

a mass $\sim g(|v|)|v|$. The quarks are also massive due to the Yukawa couplings; see section 5.3 below for a discussion of the quark masses and their relation to SPTs.

- A diagonal combination of the $SU(3)_f$ flavor symmetry and the $SU(3)_c$ gauge symmetry is unbroken and furnishes an unbroken $PSU(3)'_f$ symmetry, i.e.

$$\frac{SU(3)_c \times SU(3)_f}{\mathbb{Z}_3} \quad \to \quad PSU(3)'_f = \frac{SU(3)'_f}{\mathbb{Z}_3} \ . \tag{5.20}$$

- Since $h^i_a$ carries baryon number $B = \frac{2}{3}$, the CFL vev in (6.2) spontaneously breaks $U(1)_B$. The gauge-invariant order parameter for $U(1)_B$ breaking is the following $SU(3)_f$ singlet of baryon number $B = 2$,

$$\det(h^i_a) \ . \tag{5.21}$$

Thus the CFL vev leads to the following breaking pattern:

$$\langle \det(h^i_a) \rangle = v^3 \qquad \Longrightarrow \qquad U(1)_B \ \to \ \mathbb{Z}^F_2 \ . \tag{5.22}$$

Here the unbroken $\mathbb{Z}^F_2 \subset U(1)_B$ subgroup is generated by fermion parity $(-1)^F$, which cannot be spontaneously broken. As usual, the spontaneously broken $U(1)_B$ symmetry leads to a massless NGB. It is the only dynamical long-distance degree of freedom in the CFL Higgs phase.

- Since $\mathsf{P}$ is unitary and $h^i_a$ is a scalar (rather than a pseudo-scalar, see (5.11)), it follows that $\mathsf{P}$ remains unbroken in the presence of the complex CFL vev $v \in \mathbb{C}$ in (5.18).

By contrast, $\mathsf{T}$ is anti-unitary and commutes with $B$, but not with exponentiated $U(1)_B$ phase rotations. Since $\mathsf{T}$ leaves the operator $h^i_a$ invariant (see (5.11)) and complex conjugates its vev, it is thus ostensibly broken unless $v \in \mathbb{R}$ is real.

However, we can define an unbroken time-reversal symmetry $\mathsf{T}'$ by conjugating $\mathsf{T}$ with a suitable element of the spontaneously broken $U(1)_B$ symmetry (which effectively rotates back to real $v$).

The fact that the CFL Higgs phase of our Higgs-Yukawa-QCD model enjoys unbroken orientation-reversing symmetries such as $\mathsf{P}$ and $\mathsf{T}'$ will be important below, when we analyze the SPT phases protected by these symmetries.

## 5.3  SPTs and the Phase Diagram of $SU(3)$ Higgs-Yukawa-QCD

We now explore the phase diagram (sketched in the top panel of figure 2) as a function of the Higgs mass-squared $M_h^2 \in \mathbb{R}$:

(C) When $M_h^2 \gg \Lambda_{\text{QCD}}^2$ is positive and large relative to the $SU(3)_c$ strong-coupling scale $\Lambda_{\text{QCD}}$, we can integrate out the Higgs field, leaving $SU(3)_c$ QCD with $N_f = 3$ flavors and a positive quark mass $m_q > 0$, which we take to be in the trivial confining SPT phase.

(H) When $M_h^2 \ll -\Lambda_{\text{QCD}}^2$ is large and negative, so that we are at weak gauge coupling, the Higgs field acquires a complex color-flavor locking (CFL) vev (5.18), whose consequences for the bosonic fields – all of which are gapped except for the $U(1)_B$ NGB – were already listed below that equation. The fermions, which will all turn out to be massive, are analyzed below, where we show that they induce both a gravitational and a flavor SPT with $\theta_f = \theta_g = \pi$.

In summary, the confining phase is a gapped and trivial SPT, while the Higgs phase spontaneously breaks $U(1)_B$, resulting in a massless NGB. It is therefore clearly separated from the confining phase by at least one symmetry-breaking transition – a fact that follows from standard Landau-type arguments.

A more refined statement is that the Higgs phase is a gapless SPT with $\theta_g = \theta_f = \pi$, but this SPT is not necessary to diagnose a transition. As we will explain below, this SPT implies the existence of other (non-symmetry-breaking) phases transitions – at least for some values of the parameters in our model.

Before deducing these SPTs by analyzing the fermion masses in the Higgs phase, we must dispense with an important point that we anticipated in section 2.5: since there is a massless $U(1)_B$ NGB, we must ensure that the $\theta_f$ and $\theta_g$ SPTs cannot be eaten by it, i.e. removed by a suitable field redefinition of the NGB field. As explained in section 2.5, this happens if and only if the $U(1)_B$ symmetry has a mixed anomaly with either $SU(3)_f'$ or gravity. However, these mixed anomalies vanish (after all, they only involve the vector-like symmetries of massive QCD), we conclude that any potential $\theta_f$ and $\theta_g$ SPTs for flavor or gravity background fields are meaningful. We will therefore proceed with our analysis of SPTs in the gapless Higgs phase. However, as an additional sanity check, we will verify explicitly (in section 5.4) that these SPTs are not affected if we add explicit $U(1)_B$-violating interactions that give the NGB a mass.

We now proceed to analyze the fermion mass terms in the CFL Higgs phase of our model,

including in particular the Yukawa couplings in (5.14),

$$-\frac{1}{2}m_{IJ}\psi^I\psi^J = m_q\psi_i^a\chi_a^i - y\varepsilon_{abc}\varepsilon^{ijk}\overline{h}_i^a\psi_j^b\psi_k^c - y\varepsilon^{abc}\varepsilon_{ijk}h_a^i\chi_b^j\chi_c^k \ . \tag{5.23}$$

Here $I, J = 1, \ldots 18$. Thanks to the CFL vev (5.18) proportional to $\delta_a^i$, we can identify flavor and color indices. Thus both fermions,

$$\psi_j^i \ , \qquad \chi_j^i \ , \tag{5.24}$$

transform in the $\mathbf{3}\otimes\overline{\mathbf{3}} = \mathbf{1}\oplus\mathbf{8}$ of the unbroken $SU(3)_f'$ flavor symmetry, with $\mathbf{8}$ the traceless adjoint representation and $\mathbf{1}$ the singlet trace.

Let us compute the mass for the singlet trace fields,

$$\psi_j^i \rightarrow \frac{1}{\sqrt{3}}\delta_j^i\psi_{\mathbf{1}}, \qquad \chi_j^i \rightarrow \frac{1}{\sqrt{3}}\delta_j^i\chi_{\mathbf{1}} \ . \tag{5.25}$$

Here the factor $\sqrt{3}$ ensures that $\psi_{\mathbf{1}}, \chi_{\mathbf{1}}$ have canonical kinetic terms. Substituting into the mass terms, and taking the Higgs vev $v$ in (5.18) to be positive without loss of generality, $v > 0$, we find

$$m_q\psi_{\mathbf{1}}\chi_{\mathbf{1}} - 2yv\psi_{\mathbf{1}}\psi_{\mathbf{1}} - 2yv\chi_{\mathbf{1}}\chi_{\mathbf{1}} = \frac{1}{2}(\psi_{\mathbf{1}}, \chi_{\mathbf{1}})\begin{pmatrix} -4yv & m_q \\ m_q & -4yv \end{pmatrix}\begin{pmatrix} \psi_{\mathbf{1}} \\ \chi_{\mathbf{1}} \end{pmatrix} \ . \tag{5.26}$$

The determinant is $(4yv)^2 - m_q^2$, with eigenvalues $4yv \pm m_q$. Thus the masses for singlet fermions are[85]

$$M_{\mathbf{1}} = m_q \pm 4yv \ , \qquad v > 0 \ . \tag{5.28}$$

Now let us discuss the traceless adjoints. We can write

$$\varepsilon_{abc}\varepsilon^{ijk}\delta_i^a = \delta_b^j\delta_c^k - \delta_b^k\delta_c^i \ . \tag{5.29}$$

---

[85] Note that the eigenvectors are $\frac{1}{\sqrt{2}}(\psi_{\mathbf{1}} \pm \chi_{\mathbf{1}})$. The change of basis matrix

$$U = \frac{1}{\sqrt{2}}\begin{pmatrix} 1 & 1 \\ -1 & 1 \end{pmatrix} \ . \tag{5.27}$$

is real orthogonal.

Then using the tracelessness we find the mass term

$$m_q \psi_j^i \chi_i^j + yv\psi_j^i \psi_i^j + yv\chi_i^j \chi_j^i = (\psi \ \chi) \begin{pmatrix} 2yv & m_q \\ m_q & 2yv \end{pmatrix} \begin{pmatrix} \psi \\ \chi \end{pmatrix} . \tag{5.30}$$

The determinant is $(2yv)^2 - m_q^2$ with eigenvalues $2yv \pm m_q$. Thus the masses for the traceless adjoint fermions in the **8** of the unbroken $SU(3)_f'$ symmetry are

$$M_{\mathbf{8}} = m_q \pm 2yv , \qquad v > 0 . \tag{5.31}$$

Note that the singlet Majorana mass $4yv$ due to the Yukawa couplings is twice as large as that for the octet fermions.

We can now finally discuss the structures of the SPTs as a function of $m_q, yv > 0$. We first do so semiclassically, before commenting on possible quantum effects below:

- For small vevs, in the range
  $$0 < yv < \frac{m_q}{4} , \tag{5.32}$$
  the SPT phase is trivial.

- At $yv = m_q/4$, the mass of one singlet Weyl fermion flips sign, so that we have
  $$\theta_g = \pi , \qquad \frac{m_q}{4} < yv < \frac{m_q}{2} . \tag{5.33}$$

- At $yv = m_q/2$, the mass of one Weyl fermion in the adjoint **8** of $SU(3)_f'$ flips sign. Thus $\theta_g$ does not jump, but we get $\theta_f = N\pi = 3\pi$ from the adjoint,
  $$\theta_g = \pi , \qquad \theta_f = 3\pi , \qquad \frac{m_q}{2} < yv . \tag{5.34}$$

The extent to which strong-coupling effects, which kick in at scales below $\Lambda_{\mathrm{QCD}}$, can modify the semi-classical picture above depends on the quark mass $m_q$:

1.) If $m_q \gg \Lambda_{\mathrm{QCD}}$, then the vev $v$ needed to reach the first SPT transition $yv \sim m_q$ (assuming $y \lesssim 1$) is already in the weakly coupled Higgs regime $v \gg \Lambda_{\mathrm{QCD}}$. Then all the the SPT jumps associated with the vanishing fermion masses discussed above occur essentially at weak coupling. Thus we expect the three distinct SPT phases described above, all of which occur within the $U(1)_B$-breaking Higgs phase, to persist.

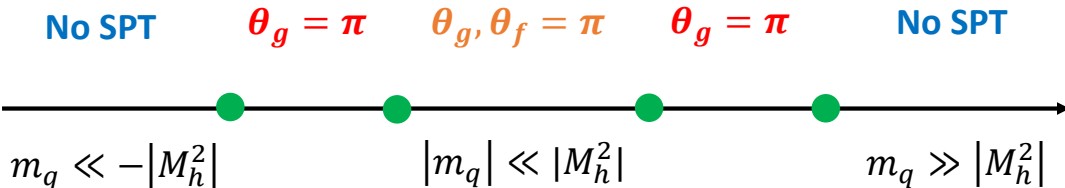

Higgs (CFL): Fixed $M_h^2 \ll 0$

**No SPT**    $\boldsymbol{\theta_g = \pi}$    $\boldsymbol{\theta_g, \theta_f = \pi}$    $\boldsymbol{\theta_g = \pi}$    **No SPT**

$m_q \ll -|M_h^2|$    $|m_q| \ll |M_h^2|$    $m_q \gg |M_h^2|$

Figure 7: Phase diagram of $SU(3)$ Higgs-Yukawa-QCD with three flavors in the weakly-coupled CFL Higgs phase at $M_h^2 \ll -\Lambda_{\mathrm{QCD}}^2$. As we dial the quark mass $m_q \in \mathbb{R}$, we trigger different SPT phase. At small quark masses, equivalently at large negative $M_h^2$, the SPT phase is characterized by $\theta_g = \theta_f = \pi$. We have omitted the massless NGB for the spontaneously broken $U(1)_B$ symmetry, which is present throughout the phase diagram.

2.) If $m_q \lesssim \Lambda_{\mathrm{QCD}}$, then the semiclassical transitions described above all happen deep within the strongly coupled regime of the $SU(3)_c$ gauge theory. In this case we cannot claim to reliably describe the jumps themselves. However we can completely reliably describe the asymptotic SPT phase at large vev $v \gg \Lambda_{\mathrm{QCD}}$, which is characterized by

$$\theta_g = \pi \ , \qquad \theta_f = 3\pi = \pi \ (\mathrm{mod} \ 2\pi) \ . \tag{5.35}$$

Even though we are primarily interested in the theory with $m_q > 0$, it is an interesting fact that the SPTs that arise deep within the Higgs phase only depend on $|m_q|$, i.e. they are symmetric under $m_q \to -m_q$. This is not the case in the confining phase, where $\mathsf{T}$ is unbroken for $m_q > 0$, but spontaneously broken when $m_q$ is sufficiently negative. The phase diagram for different $m_q$, at fixed and large negative Higgs mass-squared $M_h^2 \ll -\Lambda_{\mathrm{QCD}}^2$, is sketched in figure 7. Note that the asymptotic phase at large negative $M_h^2$ is always an SPT with $\theta_f = \theta_g = \pi$, as already emphasized above.

## 5.4    Breaking Flavor Symmetries

The purpose of this section is explain what happens to the Higgs-Yukawa-QCD model analyzed above if we break some of the flavor symmetry. We consider two kinds of breaking:

*Breaking the $SU(3)_f$ Flavor Symmetry.* We can do this by allowing independent quark masses $m_{u,d,s}$ for the three flavors, as long as these are real and positive,

$$m_u, m_d, m_s > 0 . \tag{5.36}$$

This preserves C, P, and T, but it explicitly breaks the $SU(3)_f$ flavor symmetry to its Cartan subgroup, $U(1)_1 \times U(1)_2$. The gravitational SPT is completely robust against this flavor-breaking perturbation, and so is our conclusion that the confining and CFL Higgs phases are characterized by $\theta_g = 0$ and $\theta_g = \pi$, respectively. To study the fate of the $SU(3)_f$ flavor SPT, we first reduce the $SU(3)_f$ background field strength $F$ to the conventional normalized Abelian field strengths $F_1, F_2$ of the $U(1)_1 \times U(1)_2$ Cartan subgroup,

$$F = \mathrm{diag}\left(F_1, F_2, -(F_1 + F_2)\right) . \tag{5.37}$$

This in turn implies that the $SU(3)_f$ flavor SPT reduces as follows,

$$\frac{\theta_f}{8\pi^2} \int_{M_4} \mathrm{Tr}_\square F \wedge F = \frac{2\theta_f}{8\pi^2} \int_{M_4} F_1 \wedge F_1 + \frac{\theta_f}{4\pi^2} \int_{M_4} F_1 \wedge F_2 + \frac{2\theta_f}{8\pi^2} \int_{M_4} F_2 \wedge F_2 . \tag{5.38}$$

Thus the $\theta_f = \pi$ flavor SPT in the Higgs phase leads to a $2\pi$ theta-angle for $U(1)_1$ or $U(1)_2$, which is trivial on spin manifolds. However, the off-diagonal theta-term

$$\frac{(\theta_f = \pi)}{4\pi^2} \int_{M_4} F_1 \wedge F_2 , \tag{5.39}$$

can be detected on spin manifolds, e.g. on $S_1^2 \times S_2^2$ by threading one unit of $F_1$-flux through $S_1^2$, and one unit of $F_2$-flux through $S_2^2$.

*Breaking the $U(1)_B$ Baryon Number Symmetry.* This is interesting, because it allows us to lift the gapless NGB associated with $U(1)_B$ breaking in the Higgs phase. To this end, we deform the theory by adding to it the following dimension three operator,

$$\Delta \mathscr{L} = \varepsilon \det(h_a^i) + (\text{h.c.}) , \tag{5.40}$$

where $\varepsilon$ has mass-dimension one. When $\varepsilon \in \mathbb{R}$ this operator preserves all symmetries, except $U(1)_B$, which is explicitly broken to its $\mathbb{Z}_2$ fermion number subgroup. The effects of this operator on the confining and Higgs phases at large positive and negative $M_h^2$ (see the lower panel in figure 2) are as follows:

($C_\varepsilon$) In the confining phase $h_a^i$ is heavy and can be integrated out. This leaves a highly

irrelevant six-quark operator of mass dimension nine, which does not modify the trivial SPT in the confining phase.

($H_\varepsilon$) In the Higgs phase the operator (5.40) gives the $U(1)_B$ NGB a mass-squared $\sim \varepsilon$, leading to a single gapped vacuum supporting the $\theta_g = \pi$ SPT, which is unaffected.

The lessons from this are twofold: first, we learn that the $\theta_g = \pi$ SPT in the Higgs phase is decoupled from the $U(1)_B$ NGB, and therefore also meaningful in the $\varepsilon \to 0$ limit in which the NGB becomes massless. Second, the theory with $\varepsilon \neq 0$ furnishes another example of a model with completely gapped and featureless confining ($C_\varepsilon$) and Higgs ($H_\varepsilon$) phases, which are nevertheless separated by a Higgs-confinement transition enforced by an SPT jump.

# 6  Three-Flavor QCD at Finite Baryon Density

In this section we consider conventional QCD, i.e. $SU(3)_c$ gauge theory, with $N_f = 3$ light Dirac quarks ($u$, $d$, $s$). Despite the various sources of C, P, and T violation in the standard model of particle physics, the QCD subsector of the theory is (to an impressive degree) invariant under all of these symmetries – they are neither explicitly nor spontaneously broken. This translates into the statement that the physical quark masses $m_u, m_d, m_s > 0$ are all real and positive, at zero $\theta$-angle for the dynamical $SU(3)_c$ gauge fields. While some of our analysis below applies to general quark masses of this kind, we will simplify parts of the discussion by assuming a common positive quark mass,

$$m_q = m_u = m_d = m_s > 0 \ . \tag{6.1}$$

We will study this theory at finite chemical potential $\mu_B$ for $U(1)_B$ baryon number, which is the grand canonical counterpart of finite baryon density. In Lorentzian signature, $\mu_B = A_0$ is the time component of the $U(1)_B$ background gauge field $A_\mu$. Consequently, activating $\mu_B$ breaks some of the symmetries of QCD:

- Charge-conjugation C maps $\mu_B \to -\mu_B$ and is explicitly broken; so are Lorentz boosts.

- Parity P and time-reversal T are preserved,[86] as are any flavor symmetries preserved by the quark masses, such as $U(1)_B$, and in the case of a common quark mass $m_q > 0$, a vector-like $SU(3)_f$ (Gell-Mann) flavor symmetry. Spacetime translations and spatial rotations are also good symmetries.

---

[86] This is compatible with the broken C symmetry: since Lorentz symmetry is explicitly broken, the CPT theorem no longer holds.

The fact that the orientation-reversing symmetries P and T are preserved means that we can ask whether the phases one encounters at finite $\mu_B$ are SPTs protected by these symmetries, such as the $\theta_g, \theta_f \in \{0, \pi\}$ gravitational and flavor theta-angles using throughout the paper. As explained in section 1, the fact that Lorentz invariance is explicitly broken does not in principle invalidate these SPTs (as is familiar from many examples in condensed matter physics, where Lorentz symmetry is always broken). It does however complicate the analysis, because we can no longer rely on a host of relativistic tricks, e.g. detecting SPTs by analyzing the theory on suitably non-trivial Euclidean spacetime four-manifolds $M_4$.

Another feature of turning on a real chemical potential $A_0 = \mu_B$ in Lorentzian signature is that this turns into a purely imaginary background Wilson line $A_4^E = i\mu_B$ for the Wick rotated $U(1)_B$ background gauge field $A_\mu^E$ in Euclidean signature. Consequently the Euclidean path integral measure is no longer positive definite.[87] This has two important consequences:

1.) There is a so-called sign problem for Monte Carlo simulations, making lattice studies of finite-density QCD very challenging, as reviewed in [149, 150].

2.) The positivity assumptions of the Vafa-Witten theorem (see section 3.4) no longer hold, which raises the possibility of non-trivial SPT phases. Indeed we will argue below that three-flavor QCD at sufficiently large $\mu_B \gg \Lambda_{QCD}, m_u, m_d, m_s$ is a non-trivial SPT.

The QCD phase diagram as a function of $\mu_B$ is very rich. Despite much effort, it is only partially understood (in large part due to the sign problem); see the reviews [94–97] for a summary of what is known. However, the asymptotic regimes of very small and very large $\mu_B$ are under good theoretical control:

- When $\mu_B \ll M_{\text{baryon}} \sim \Lambda_{\text{QCD}}$, the theory is effectively at zero density and continuously connected to the standard confined QCD vacuum at $\mu_B \to 0$, i.e. it is a gapped and trivial SPT.

- When $\mu_B \gg \Lambda_{\text{QCD}}, m_u, m_d, m_s$, i.e. at very high densities, asymptotic freedom implies that the theory is in a weakly-coupled, color-superconducting Higgs phase that also spontaneously breaks $U(1)_B$, leading to a single massless NGB. This is reviewed in [95]; we will sketch the essentials in section 6.1 below.

  We then proceed to argue (in section 6.2) that high-density QCD realizes exactly the same SPT phase as the weakly-coupled Higgs phase of $SU(3)_c$ Higgs-Yukawa-QCD (at

---

[87] Although we only study the case of real chemical potential, it is also interesting to consider the case of purely imaginary chemical potential [147], which amounts to real $A_4^E$ and does not suffer from a sign problem (see also [148]).

zero density) analyzed in section 5 above. This implies that high-density QCD also harbors non-trivial gravitational and flavor SPTs described by $\theta_g = \theta_f = \pi$, in addition to the massless $U(1)_B$ NGB.

In section 6.3 we consider some implications of the fact that high-density QCD is a non-trivial SPT phase. We first examine possible consequences for the QCD phase diagram, focusing on the simplified case of a common quark mass $m_q > 0$ with $SU(3)_f$ flavor symmetry. This case was analyzed in [19], under the assumption that $m_q \lesssim \Lambda_{\text{QCD}}$; these authors proposed that the confining low-density phase and the $U(1)_B$ breaking high-density phase are separated by a single symmetry-breaking transition of Landau type at $\mu_B \sim \Lambda_{\text{QCD}}$. We explain why the non-trivial $\theta_g = \pi$ SPT at high densities suggests a second transition within the $U(1)_B$ breaking phase, in tension with the Higgs-confinement (or "quark-hadron") continuity proposal of [19].

Finally, we briefly mention possible implications of the SPT for neutron stars, which furnish a realization of cold, dense QCD matter in nature.

## 6.1 Color-Flavor-Locking (CFL) at High Densities

Here we briefly review (following [95]) the behavior of three-flavor QCD at very large baryon chemical potential, $\mu_B \gg \Lambda_{\text{QCD}}, m_u, m_d, m_s$. In this regime the differences between the quark masses can be largely ignored, and we will assume a common non-zero quark mass $m_q > 0$, which preserves an $SU(3)_f$ flavor symmetry.[88]

In the absence of interactions, the quarks form a degenerate Fermi gas with Fermi energy $\mu_B$. The light modes near the Fermi surface have large momenta $\sim \mu_B$, which implies that the effective gauge coupling $g(\mu_B) \ll 1$ is very small thanks to asymptotic freedom. To study the BCS pairing instabilities of the Fermi surface, it suffices to consider one-gluon exchange between pairs of quarks. The most relevant attractive interaction is in the parity-even spin-0 (i.e. scalar rather than pseudo-scalar) channel, and it is anti-symmetric in both color and flavor indices.[89] This leads to a two-quark condensate with the following quantum

---

[88] The story changes slightly if the quarks are exactly massless: in this case the high-density CFL phase has additional NGBs due to chiral symmetry breaking [151].

[89] Pairing in the spin-0 channel is favored because it is enhanced by the spherically symmetric Fermi surface. Since one-gluon exchange is attractive in the color anti-symmetric channel (and repulsive in the color-symmetric channel), restricting to spin-0 also implies anti-symmetry in flavor. Finally, the fact that the scalar channel is favored over the parity-odd pseudo-scalar one is not visible in perturbation theory and requires considerations involving QCD instantons. See section II.A of [95] for more details.

numbers, or equivalently an expectation value for the following composite Higgs field [151],

$$\mathcal{H}_a^i \equiv \varepsilon_{abc}\varepsilon^{ijk}\left(\psi_j^b\psi_k^c + \overline{\chi}_j^b\overline{\chi}_k^c\right) \ , \qquad \langle\mathcal{H}_a^i\rangle = v\delta_a^i \qquad |v| \sim \mu_B \ . \tag{6.2}$$

Not coincidentally, the quantum numbers of the composite Higgs field $\mathcal{H}_a^i$ above are exactly the same as those of the elementary Higgs field $h_a^i$ in the $SU(3)_c$ Higgs-Yukawa-QCD model (at zero density), introduced in (5.9). Indeed, we have intentionally engineered the properties of the elementary field $h_a^i$ in the latter model to replicate the physics of the composite field $\mathcal{H}_a^i$ in high-density QCD. In addition to having identical quantum numbers, the scalar potential for $h_a^i$ in (5.13) was chosen to trigger the Color-Flavor-Locking (CFL) vev in (5.18) in the Higgs phase, which exactly matches the CFL vev of $\mathcal{H}_a^i$ in (6.2).

Since both models lead to weakly-coupled Higgs condensates with identical quantum numbers, many qualitative aspects of the physics are very similar:[90]

- The $SU(3)_c$ gauge symmetry is completely Higgsed at the scale $|v| \sim \mu_B$, leading to vector boson masses $\sim g(|v|)|v| \sim g(\mu_B)\mu_B$ (see section IV.F in [95]).

- The $SU(3)_f$ flavor symmetry is preserved by mixing with the gauge symmetry.

- The $U(1)_B$ symmetry is spontaneously broken, leading to a single massless NGB. The gauge-invariant order parameter is the following six-quark (or hexaquark) operator, which has baryon number two and is otherwise flavor neutral,

$$\det\left(\mathcal{H}_a^i\right) \ , \qquad B\left(\det\left(\mathcal{H}_a^i\right)\right) = 2 \ . \tag{6.3}$$

  In the ordinary QCD vacuum at $\mu_B = 0$ the operator $\det(\mathcal{H}_a^i)$ has been conjectured [152] to create a particle known as the $H$-dibaryon – an exotic bound state of two baryons. The $H$-dibaryon will make an appearance in section 6.3.1 below.

- Parity $\mathsf{P}$ is unbroken, and $\mathsf{T}$ is unbroken after a suitable conjugation by $U(1)_B$. As in Higgs-Yukawa-QCD, this means that we can contemplate SPT phases protected by these symmetries.

- The $U(1)_B$ NGB is the only gapless mode. In particular, the fermions acquire non-perturbative pairing gaps. As in Higgs-Yukawa-QCD, the fermions decompose into a singlet $\mathbf{1}$ and an octet $\mathbf{8}$ of the unbroken $SU(3)_f$ flavor symmetry, and the respective

---

[90] Of course there are also differences, e.g. finite-density QCD is not Lorentz invariant and the sound speed in the high-density $U(1)_B$ superfluid phase is not the speed of light.

gaps differ by a factor of two. The exponential scaling of these gaps was found in [98],

$$\Delta_{\mathbf{1}} = 2\Delta_{\mathbf{8}} \sim \mu_B e^{-\frac{K}{g(\mu_B)}} \ , \qquad K = \frac{3\pi^2}{\sqrt{2}} \ . \tag{6.4}$$

Note that the gap exponent scales as $1/g$, rather than the $1/g^2$ scaling familiar from standard BCS theory. This enhancement of the gap is due to the fact that the pairing interaction is long range, because it is mediated by gluons that are initially massless, before acquiring masses due to screening; by contrast, a short-range pairing interaction would lead to $1/g^2$. Comparing with section 5, where the fermion gap due to Higgsing is $\sim y|v|$, we see that the role of the dimensionless Yukawa coupling $y$ is played by $e^{-K/g(\mu_B)}$.[91]

## 6.2   Determining the SPT Phase of High-Density QCD

We have seen above that the weakly-coupled Higgs phase of $SU(3)_c$ Higgs-Yukawa-QCD (in the vacuum) and the weakly-coupled CFL phase of high-density QCD appear to be qualitatively identical: they realize the same symmetry-breaking pattern, leading to a phase that is fully gapped except for the massless NGB of the spontaneously broken $U(1)_B$ symmetry.

We will now argue that the two phases are also identical as far as SPTs are concerned, i.e. both support SPTs that (in relativistic terms) can be summarized by theta-angles for background gravity and the $SU(3)_f$ flavor symmetry,

$$\theta_g = \theta_f = \pi \ . \tag{6.5}$$

### 6.2.1   Argument Based on Quark Pairing

If we subscribe to the idea that the SPT in both phases is entirely due to the (ultimately almost free) fermions, then it seems plausible that the SPTs should also be the same. This is because the fermions essentially have the same fate in the two models – albeit in a somewhat different order:

(i) In QCD at large $\mu_B$, the Fermi surface is initially gapless, before gluon exchange leads to pairing and the fermion gaps (6.4).

---

[91] As mentioned at the beginning of section 6.1, we focus on the regime where the chemical potential $\mu_B$ is much larger than the quark masses – sufficiently large so that $\Delta_{\mathbf{1}}, \Delta_{\mathbf{8}} \gg m_u, m_d, m_s$.

(ii) Higgs-Yukawa-QCD at $\mu_B = 0$, but deep within its Higgs phase, pairs fermions in exactly the same channel (via the Yukawa couplings), leading to gaps that have exactly the same structure (including the relative factor of 2 in (6.4)), though the overall magnitude of the gap is set by $y|v|$, rather then by BCS pairing. Once the fermions acquire these gaps, it can be checked explicitly that turning on a chemical potential $\mu_B$ deforms but never closes said gaps.[92]

A consistency check of this picture is that the SPT phase can (at least in principle) be determined at $\mu_B = 0$, as we do in the Higgs-Yukawa-QCD model. This is so because charge-conjugation $\mathsf{C}$, which flips $\mu_B \to -\mu_B$ and is unbroken at $\mu_B = 0$, does not have a mixed anomaly with the symmetries that protect the SPT. If present, such an anomaly would render the SPT at $\mu_B = 0$ ambiguous.

### 6.2.2 Deforming to Zero-Density Higgs-Yukawa-QCD

We will now sketch an explicit deformation that connects the CFL phase of high-density QCD and the Higgs phases of zero-density Higgs-Yukawa-QCD, without changing the SPT. Crucially, the fermions will remain gapped throughout the deformation. By contrast, the deformation will at intermediate stages give rise to new massless bosons, but we argue that they do not affect the SPT phase.

Let us start with three-flavor QCD at very large $\mu_B \gg \Lambda_{\mathrm{QCD}}$, so we are in the weakly-coupled phase reviewed in section 6.1 above.[93] In particular, the composite Higgs field in (6.2) gets a CFL vev

$$\langle \mathcal{H}_a^i \rangle = v \delta_a^i \,, \qquad |v| \sim \mu_B \,, \tag{6.7}$$

leading to Higgsing at the scale $\mu_B$ and a gapless $U(1)_B$ NGB. In order to focus on the SPT, we will follow the discussion in section 5.4 and simplify our life by lifting the NGB via

---

[92] For a single free Dirac quark whose gap (equivalently, Majorana mass) $\Delta > 0$ is much larger than its Dirac mass $m_q$ (so that $m_q$ can be ignored), the dispersion relation at 3-momentum $\vec{k}$ and chemical potential $\mu_B$ for the vector-like $U(1)_B$ symmetry acting on the quark is

$$\omega_\pm(\vec{k}) = \sqrt{(|\vec{k}| \pm \mu_B)^2 + \Delta^2} \,, \tag{6.6}$$

see e.g. equation (30) in [95]. This shows that the fermions are never gapless – a conclusion that remains unchanged if we include $m_q$, as long as $\Delta \gtrsim m_q$.

[93] We also assume that the quark masses $m_u, m_d, m_s \lesssim \Lambda_{\mathrm{QCD}}$, as is the case in real QCD. Without this assumption some inequalities that appear in the subsequent argument have to be modified slightly.

a $U(1)_B$ breaking interaction involving the (highly irrelevant) $H$-dibaryon operator,

$$\Delta \mathscr{L} = \varepsilon \det \left( \mathcal{H}_a^i \right) + (\text{h.c.}) , \qquad \varepsilon > 0 . \tag{6.8}$$

This explicitly breaks $U(1)_B$, but preserves all other symmetries (in particular those that protect the SPTs we are interested in). Once we turn on this interaction, $\mu_B$ is demoted from $U(1)_B$ chemical potential to a Lorentz-breaking but otherwise acceptable coupling in the Lagrangian. (This is possible because we are working at finite $\mu_B$, rather than at finite baryon density, which would be destabilized by the interaction (6.8).) One possible worry is that this coupling, unlike a chemical potential, is renormalized, but the theory is very weakly coupled and we are only working to first order in $\varepsilon$. This is sufficient to give the NGB a mass-squared $\sim \varepsilon$. It also pins the phase of the complex vev $v$ in (6.8) to be real and positive,

$$v \sim \mu_B > 0 . \tag{6.9}$$

In order to make contact with Higgs-Yukawa-QCD, we now add to the theory above a fundamental scalar Higgs field

$$h_a^{i'} , \tag{6.10}$$

which is transforms as a $\overline{\mathbf{3}}$ of $SU(3)_c$, with anti-fundamental index $a = 1, 2, 3$, but is not directly coupled to the fermions. It therefore has its own flavor symmetry,

$$U(3)_f' = \frac{U(1)_B' \times SU(3)_f'}{\mathbb{Z}_3} , \tag{6.11}$$

which acts on the index $i' = 1, 2, 3$ and is unrelated to the $SU(3)_f$ flavor symmetry acting on the fermions.[94] If we fix unitary gauge and remove all the NGBs eaten by the $SU(3)_c$ gauge fields from $\mathcal{H}_a^i$, then all components of $h_a^i$ effectively become physical,

$$h_a^{i'} \rightarrow h_j^{i'} . \tag{6.12}$$

Equivalently, we could work with the gauge-invariant composite $\overline{\mathcal{H}}_j^a h_a^{i'}$.

We also add for $h_a^{i'}$ the scalar potential (5.16) of the $SU(3)_c$ Higgs-Yukawa-QCD theory in section 5. At very large positive mass $M_h^2 \gg \Lambda_{\text{QCD}}^2$ for the field $h_a^{i'}$ we can integrate it out and recover the finite-density QCD theory we started with.

---

[94] Before breaking $U(1)_B$, the symmetry group is $\left( U(3)_f \times U(3)_f' \right) / \mathbb{Z}_3$, where the $\mathbb{Z}_3$ quotient is due to identification of the diagonal $\mathbb{Z}_3$ flavor symmetry with the center of the $SU(3)_c$ gauge group.

The next step is to dial $M_h^2$ through zero to large (but not too large) negative values,

$$-\mu_B^2 \ll M_h^2 \ll -\Lambda_{\text{QCD}}^2 . \tag{6.13}$$

Since the gauge theory is pinned in the weakly-coupled Higgs phase by the large vev $v \sim \mu_B$ in (6.9), the scalar $h_j^{i'}$ is effectively decoupled from the rest of the theory as we dial $M_h^2$ from large positive values into the range (6.13). Moreover, its Euclidean partition function on any four-manifold $M_4$ (which is meaningful since the scalar preserves Lorentz invariance) is manifestly positive, even as we dial through the symmetry-breaking transition at $M_h = 0$, where new massless modes appear. Thus the presence of the scalar does not eat or change the SPT phase of the finite-density QCD theory that we are trying to determine. See section 2.5.2 for a discussion of this type of argument for analyzing gapless SPTs.

At the point $M_h = 0$ the scalar $h_j^{i'}$ undergoes a phase transition to a symmetry-breaking phase with a CFL vev of the form

$$h_j^{i'} = v' \delta_j^{i'} , \qquad v' \in \mathbb{C} . \tag{6.14}$$

This spontaneously breaks the $SU(3)_f \times SU(3)_f'$ symmetry to its diagonal subgroup, and it also breaks the $U(1)_B'$ symmetry that rotates $h_j^{i'}$ by an overall phase. This gives rise to massless NGBs, which do not affect the SPT thanks to the positivity of the $h_j^{i'}$ partition function discussed above.

Another consequence of the $h$-vev $v'$ in (6.14) is that the scale of Higgsing is shifted,

$$v \to \sqrt{v^2 + |v'|^2} , \qquad v \sim \mu_B , \tag{6.15}$$

but for now we have $|v'| \ll v$ thanks to (6.13), so this effect is small.

Next, we add the Yukawa coupling (5.14) present in the $SU(3)_c$ Higgs-Yukawa-QCD model to the combined theory,

$$\mathscr{L}_{\text{Yukawa}} = y\varepsilon_{abc}\varepsilon^{ijk}\overline{h}_i^a \left( \psi_j^b \psi_k^c + \overline{\chi}_j^b \overline{\chi}_k^c \right) + (\text{h.c.}) = y\overline{\mathcal{H}}_i^a h_a^i + (\text{h.c.}) , \qquad y > 0 . \tag{6.16}$$

This interaction identifies $i = i'$, explicitly breaking the $SU(3)_f \times SU(3)_f'$ symmetry to its diagonal subgroup; it also explicitly breaks the $U(1)_B'$ symmetry acting on $h_a^i$. Thus all symmetries that were previously spontaneously broken, and gave rise to gapless NGBs, are now explicitly broken, so that all NGBs are gapped out. The Yukawa coupling (6.16) also aligns the previously misaligned vevs of the composite and elementary Higgs fields in color

and flavor space,

$$\langle \mathcal{H}_a^i \rangle = v \delta_a^i \qquad (v \sim \mu_B > 0) \ , \qquad \langle h_a^i \rangle = v' \delta_a^i \qquad (v' > 0) \ . \tag{6.17}$$

A similar alignment also occurs for the total fermion gaps, since it is energetically favorable for these to be as large as possible. This implies that the fermion gaps (6.4) due to pairing at the Fermi surface and the gaps induced by the Yukawa couplings (6.16) add,

$$\Delta_\mathbf{1}^{\text{total}} = 2\Delta_\mathbf{8}^{\text{total}} \sim \mu_B e^{-\frac{1}{g_{\text{eff}}}} + yv \ , \qquad yv > 0 \ . \tag{6.18}$$

Here we have replaced the gauge coupling $g(\mu_B)$ in (6.4) by an effective coupling $g_{\text{eff}} > 0$ that captures the total pairing interaction at the Fermi surface, which receives contributes from both gluon exchange, and the Yukawa couplings together with Higgs boson exchange. As we have emphasized, both of these interactions lead to pairing in the same channel. In the regime $\mu_B \gg v'$ both the gluons and the Higgs are light at the Fermi surface, leading to a long-range pairing interaction. By contrast, if $v' \gg \mu_B$ then both bosonic mediators are heavy at the Fermi surface and can be integrated out, leading to contact interactions.

Finally, we can lower the chemical potential $\mu_B$, while keeping $M_h^2$ and hence $v'$ fixed. This lowers the scale of Higgsing from $\mu_B$ to $v'$ according to (6.15), and it also changes the fermion gaps according to (6.18), but the theory remains gapped throughout. At $\mu_B = 0$ we recover the Higgs phase of Higgs-Yukawa-QCD at zero density, with a small mass for the $U(1)_B$ NGB that is inherited from (6.8) via (6.16). This completes the argument.

## 6.3   Possible Implications of the SPT

In this section we discuss several possible physical consequences of the fact that three-color QCD at very high baryon density is a non-trivial SPT with $\theta_g = \theta_f = \pi$.

### 6.3.1   Unexpected Transitions in the QCD Phase Diagram

We have shown that SPT jumps can can enforce Higgs-confinement transitions in theories where they cannot be anticipated by symmetry-breaking considerations in the spirit of Landau. We have also shown that QCD at very large $U(1)_B$ chemical potential $\mu_B \gg \Lambda_{\text{QCD}}$ is a nontrivial SPT with $\theta_g = \theta_f = \pi$, as well as well as a $U(1)_B$ breaking superfluid. It is therefore natural to suspect that the SPT may trigger new, unexpected transitions in the

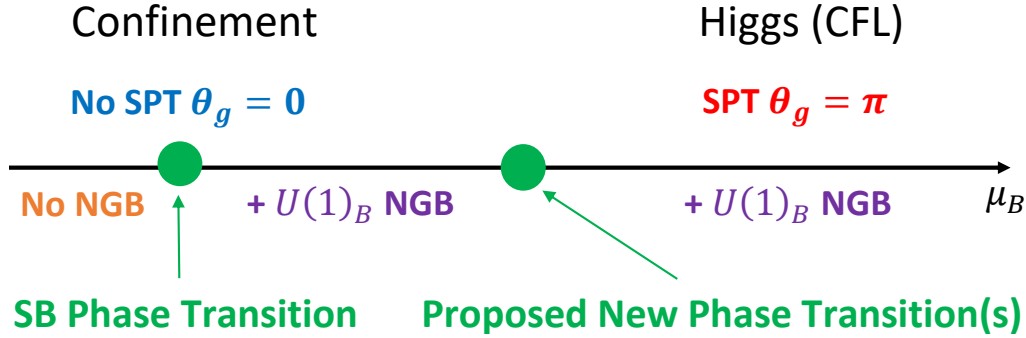

Figure 8: Proposed phase diagram for three-flavor QCD at finite baryon density. As we increase the chemical potential for $U(1)_B$, the theory experiences a symmetry-breaking (SB) phase transition that spontaneously breaks the $U(1)_B$ symmetry, resulting in Nambu-Goldstone bosons (NGB). We propose at least one additional phase transition due to the jump of the SPT phase from $\theta_g = 0$ at lower densities to $\theta_g = \pi$ in the CFL Higgs phase at high densities.

QCD phase diagram as a function of $\mu_B$.[95]

We will consider this question in the somewhat simpler $SU(3)_f$ flavor symmetric context with a single common quark mass $m_q \lesssim \Lambda_{\text{QCD}}$. This case was also considered in [19] (see also the closely related work [99,100]), where a simple phase diagram with a single transition was proposed as a function of $\mu_B$ (see figure 8):

(C) When $\mu_B < \mu_{\text{SF}} \sim \Lambda_{\text{QCD}}$ the theory is gapped and confining.

(SF) When $\mu_B \sim \mu_{\text{SF}}$, there is a phase transition to a superfluid (denoted by SF) regime with spontaneously broken $U(1)_B$ symmetry. Since this transition is believed to occur at typical hadronic densities, i.e. at $\mu_{\text{SF}} \sim \Lambda_{\text{QCD}}$, it should be possible to describe it using the confined hadronic degrees of freedom of conventional zero-density QCD. The authors of [19] proposed that this phase smoothly extends to arbitrarily high $\mu_B$, where QCD is known to be a weakly-coupled CFL Higgs phase that also breaks $U(1)_B$ (see section 6.1). This is a version of Higgs-confinement continuity that applies within the $U(1)_B$ breaking superfluid phase and was termed quark-hadron continuity in [19].

As we now explain, we consider this picture to be in tension with the SPT phases at low and high densities that we have established: a trivial SPT in the confining phase at $\mu_B < \mu_{\text{SF}}$,

and a non-trivial SPT with $\theta_g = \theta_f = \pi$ in the high-density CFL phase at $\mu_B \gg \Lambda_{\mathrm{QCD}}$. The only way these SPTs are compatible with the quark-hadron continuity proposal of [19] is that if the $U(1)_B$ superfluid transition at $\mu_B = \mu_{\mathrm{SF}}$ exactly coincides with the SPT jump from $\theta_g, \theta_f = 0$ to $\theta_g, \theta_f = \pi$. While we cannot rule this out rigorously (after all, the transition occurs deep within the strong-coupling region), we find it implausible – essentially because the two phenomena are not intrinsically intertwined and their coincidence would seem like a finely-tuned accident.[96] To wit:

[95] As before, we consider the theory at zero temperature, but in general new features on the $T = 0$ boundary of the QCD phase diagram also have implications at finite temperature.

(i) Applying the same Landau-style reasoning to the $SU(3)_c$ Higgs-Yukawa-QCD model in section 5 also leads to the proposal that there should only be two phases, separated by a single $U(1)_B$ breaking transition. However, there we explicitly established the existence of additional transitions, where the SPTs jump – at least for some values of the parameters.

(ii) Below we discuss a simple, minimal model for the $U(1)_B$ breaking superfluid transition at $\mu_B = \mu_{\mathrm{SF}} \sim \Lambda_{\mathrm{QCD}}$ in terms of the lightest confined, hadronic degrees of freedom. This simple model can accommodate the jump of the flavor SPT from $\theta_f = 0$ to $\pi$, but it is not sufficient to explain the SPT jump from $\theta_g = 0$ to $\pi$, suggesting a second SPT-enforced phase transition within the superfluid phase, i.e. at higher values of $\mu_B > \mu_{\mathrm{SF}}$ (see figure 8).

Before we discuss the model in point (ii) above, we pause to mention that the possibility of a non-Landau phase transition within the superfluid phase was already contemplated in [101] (with subsequent related work in [102]), where the superfluid vortices associated with the spontaneously broken $U(1)_B$ symmetry were analyzed. The properties of these vortices (which the authors of [101] deduced using a purely bosonic Landau-Ginzburg-type effective model) appear to be unrelated to our SPT considerations (which focus on the unbroken symmetries and are entirely driven by the fermions).

Returning to point (ii) above, we will now review the picture for the $U(1)_B$ breaking transition $\mu_B = \mu_{\mathrm{SF}}$ put forward in [19], before elaborating on it by analyzing the SPTs. This picture involves the $H$-dibaryon particle, already introduced around (6.3), which is a parity-even spin-0 scalar that is also an $SU(3)_f$ flavor singlet and carries baryon number $B = 2$. It

---

[96] It is well known that QCD with physical quark masses displays a number of finely tuned phenomena, e.g. the famously shallow binding of the deuteron. For this reason we cannot simply dismiss the apparent fine tuning mentioned above out of hand.

can therefore be created by the six-quark operator in (6.3), see also (6.2), which we repeat,

$$H = \det\left(\mathcal{H}_a^i\right) \ , \qquad \mathcal{H}_a^i \equiv \varepsilon_{abc}\varepsilon^{ijk}\left(\psi_j^b\psi_k^c + \overline{\chi}_j^b\overline{\chi}_k^c\right) \ . \tag{6.19}$$

The existence of such an exotic bound state in zero-density QCD was first proposed in [152], by appealing to a strong chromo-magnetic attraction among the six quarks in (6.19) within the context of the MIT bag model for hadrons. While the $H$-dibaryon has not been observed experimentally (see [153,154] for some recent searches), its existence is on more solid footing in the $SU(3)_f$ flavor-symmetric case with a single quark mass $m_q$ that we are considering here. Additional theoretical evidence includes lattice simulations (see e.g. the recent study [155] and references therein) and Skyrme-type constructions using three-flavor chiral Lagrangians (see e.g. [156,157] and references therein).

The proposal of [19] is that the $U(1)_B$ breaking superfluid transition at $\mu_B = \mu_{\mathrm{SF}}$ is driven by Bose condensation of the $H$-particle. For future reference, we note that this transition is plausibly first order, see [158,159]. We will now refine this picture by considering the behavior of the SPTs in the vicinity of the transition.

We employ a simple effective Lagrangian that summarizes the interactions of the $H$-particle and the lightest spin-$\frac{1}{2}$ baryons, which transform in the adjoint $\mathbf{8}$ of $SU(3)_f$. Note that all of these are color-neutral confined degrees of freedom, as is appropriate at hadronic densities. Gauge-invariant interpolating fields with $B = \pm 1$ that describe the $\mathbf{8}$ baryons are

$$(\mathcal{B}_\alpha)_i^j = \varepsilon_{abc}\varepsilon^{jkl}\varepsilon^{\beta\gamma}(\psi_\alpha)_i^a(\psi_\beta)_k^b(\psi_\gamma)_l^c \ , \qquad (\widetilde{\mathcal{B}}_\alpha)_j^i = \varepsilon^{abc}\varepsilon_{jkl}\varepsilon^{\beta\gamma}(\chi_\alpha)_a^i(\chi_\beta)_b^k(\chi_\gamma)_c^l \ . \tag{6.20}$$

Both of these are traceless in the flavor indices, $(\mathcal{B}_\alpha)_i^i = (\widetilde{\mathcal{B}}_\alpha)_i^i = 0$, as required for an $\mathbf{8}$ of $SU(3)_f$. For our purposes, it will suffice to consider a crude effective Lagrangian that treats $H, \mathcal{B}, \widetilde{\mathcal{B}}$ as elementary fields, with canonical kinetic terms. This is surely too crude to capture the strong interactions among these particles, but it may give useful clues about the SPTs. There are mass terms for all fields, as well as Yukawa couplings $Y_\mathbf{8}$ that describe the interaction of $H$ with two $\mathbf{8}$ baryons,

$$\begin{aligned}
\mathscr{L}_{\mathrm{masses + Yukawas}}^{H\mathcal{B}\widetilde{\mathcal{B}}} = &- M_H^2|H|^2 - M_{\mathbf{8}\,\mathrm{baryon}}\left((\mathcal{B}^\alpha)_i^j(\widetilde{\mathcal{B}}_\alpha)_j^i + \mathrm{h.c.}\right) \\
&- Y_\mathbf{8}\overline{H}\left((\mathcal{B}^\alpha)_i^j(\mathcal{B}_\alpha)_j^i + (\widetilde{\mathcal{B}}_{\dot\alpha}^\dagger)_i^j(\widetilde{\mathcal{B}}^{\dagger\dot\alpha})_j^i\right) + (\mathrm{h.c.}) \ ,
\end{aligned} \tag{6.21}$$

At $\mu_B = 0$ we are describing QCD in the standard Lorentz-invariant vacuum, where the

following inequalities hold

$$M_H^2 > 0 \ , \qquad M_{\mathbf{8} \text{ baryon}} > 0 \ , \qquad Y_{\mathbf{8}} > 0 \ . \tag{6.22}$$

The first inequality ensures that $\langle H \rangle = 0$, so that $U(1)_B$ is unbroken and $M_H$ is the mass of the $H$-particle. The remaining inequalities follow from $\mathsf{T}$ and $\mathsf{P}$ symmetry,[97] and the requirement that the $\mu_B = 0$ confining phase is a trivial SPT.

For simplicity, we assume that $M_H < 2M_{\mathbf{8} \text{ baryon}}$, so that the $H$ cannot decay into baryon pairs.[98] In this case the $H$-particle has the smallest baryon number per unit mass and is expected to condense at $\mu_B = \mu_{\text{SF}} \sim \frac{1}{2}M_H$, triggering the $U(1)_B$-breaking transition. The fate of the SPTs depends on the strength of this first-order transition. If the transition is sufficiently strong, then the vev $\langle H \rangle$ jumps to sufficiently large values to ensure that the Majorana mass $\sim Y_{\mathbf{8}}|\langle H \rangle|$ of the baryons can overwhelm their Dirac mass $M_{\mathbf{8} \text{ baryon}}$. If this happens, half of the 16 Weyl fermions comprising the $\mathbf{8}$ Dirac baryons experience a sign flip for their real mass eigenvalues. This does not affect the gravitational SPT, which remains

$$\theta_g = 0 \ . \tag{6.24}$$

However, comparing with (2.46) we see that the flavor SPT jumps to $\theta_f = I(r)\pi$, where $r = \mathbf{8}$ is the adjoint of $SU(3)_f$, with Dynkin index $I(r) = 3$ (see appendix A). We thus find that the superfluid phase described by $H$-condensation is also a non-trivial flavor SPT with

$$\theta_f = 3\pi \sim \pi \quad (\text{mod } 2\pi) \ . \tag{6.25}$$

If the $U(1)_B$-breaking transition is only weakly first order, then $Y_{\mathbf{8}}|\langle H \rangle| < M_{\mathbf{8} \text{ baryon}}$ and the $\mathbf{8}$ baryon masses remain positive at the transition. In this case the flavor SPT jump is delayed to higher values of $\mu_B$, forcing another transition. We will now argue that such a delayed SPT jump, accompanied by another transition at $\mu_B > \mu_{\text{SF}}$, is much more plausible for the gravitational SPT.

The discussion above shows that the coupling of the $\mathbf{8}$ baryons to the $H$-particle, together

---

[97] Note the actions of $\mathsf{C}$, $\mathsf{P}$, and $\mathsf{T}$ in (3.12) on the baryons,

$$\mathsf{C} : (\mathcal{B}_\alpha)_i^j \leftrightarrow (\widetilde{\mathcal{B}}_\alpha)_j^i \ , \qquad \mathsf{P} : (\mathcal{B}_\alpha)_i^j \to i(\widetilde{\mathcal{B}}^{\dagger\dot\alpha})_i^j \ , \qquad \mathsf{T} : (\mathcal{B}_\alpha)_i^j \to (\mathcal{B}^\alpha)_i^j \ . \tag{6.23}$$

[98] The picture is only slightly modified if $M_H > 2M_{\mathbf{8} \text{ baryon}}$. In this case the $U(1)_B$-breaking transition occurs at $\mu_B \sim M_{\mathbf{8} \text{ baryon}} < \frac{1}{2}M_H$ and is triggered by BCS pairing of the $\mathbf{8}$ baryons in the channel with the quantum numbers of the $H$.

with $H$-condensation at the $U(1)_B$-breaking transition, can (at least in principle) also account for the jump of the flavor SPT to its large-$\mu_B$ value $\theta_f = \pi$. However, it does not account for the gravitational SPT $\theta_g = \pi$ that is present in the large-$\mu_B$ CFL Higgs phase. As explained in section 6.2, this SPT is ultimately due to the fact that the total number of quarks in three-flavor QCD, i.e. $(N_f = 3) \times (N_c = 3) = 9$, is odd.

The apparent mismatch between the 9 elementary quarks in the Higgs regime and the **8** familiar spin-$\frac{1}{2}$ baryons in the confined regime was addressed in the context of the quark-hadron continuity proposal of [19], by appealing to the possibility of a $9^{\text{th}}$ confined spin-$\frac{1}{2}$ baryon that is a singlet **1** under the $SU(3)_f$ symmetry,

$$b_\alpha , \qquad \widetilde{b}_\alpha . \tag{6.26}$$

If such a baryon exists at all, it is expected to be considerably heavier than the **8** baryons,

$$M_{\mathbf{1}\text{ baryon}} \gg M_{\mathbf{8}\text{ baryon}} . \tag{6.27}$$

The reason is Fermi statistics: in a constituent quark description it is not possible to maintain anti-symmetry in color, flavor, and spin without exciting some quarks to higher orbitals, resulting in a heavier baryon.

By analogy with (6.21), we add Yukawa couplings of the **1** baryons to the $H$-particle,

$$\Delta \mathscr{L} = Y_{\mathbf{1}} \overline{H} \left( b^\alpha b_\alpha + b^\dagger_\alpha b^{\dagger\dot\alpha} \right) + (\text{h.c.}) , \qquad Y_{\mathbf{1}} \in \mathbb{R} . \tag{6.28}$$

A vev for $H$ then splits the singlet baryon into two Weyl fermions with real masses

$$M_{\mathbf{1}\text{ baryon}} \pm Y_{\mathbf{1}} |\langle H \rangle| . \tag{6.29}$$

If the vev $|\langle H \rangle|$ at the $U(1)_B$-breaking transition is so large as to make one of these masses negative, then the gravitational SPT jumps to its large-$\mu_B$ value

$$\theta_g = \pi . \tag{6.30}$$

This would lead to a picture consistent with the quark-hadron continuity proposal of [19], with the additional feature that the SPTs jump to their large-$\mu_B$ values precisely at the $U(1)_B$-breaking transition.

However, depending on the size of the (in principle large) mass difference (6.27) between **1** and **8** baryons (as well as other factors, e.g. the size of the Yukawa couplings $Y_{\mathbf{1},\mathbf{8}}$, which we

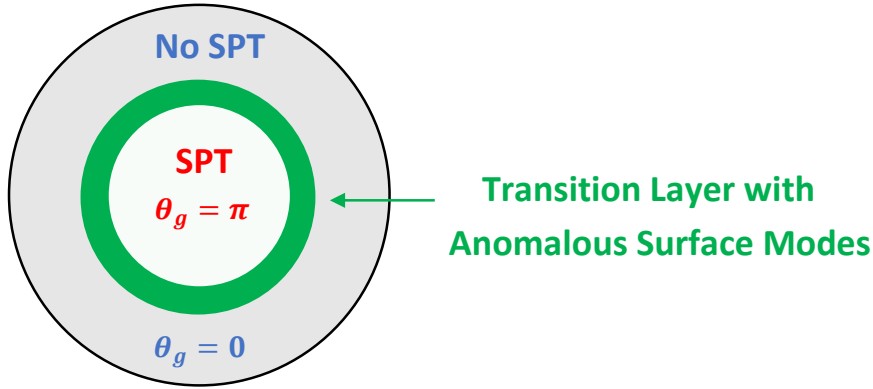

Figure 9: Neutron star with a dense quark-matter core supporting a $\theta_g = \pi$ SPT (red text), separated from an outer region with $\theta_g = 0$ (blue text) by a transition layer with anomalous surface modes (green region and text) characterized by a non-vanishing, quantized thermal Hall conductance $\kappa_{xy}$. In this diagram we have suppressed the gapless $U(1)_B$ NGBs.

assume to be comparable), it seems more plausible that the larger vev $\langle H \rangle \sim M_{\mathbf{1}\ \text{baryon}}/Y_{\mathbf{1}}$ that is needed to flip the sign of the singlet baryon mass is only attained at higher values of $\mu_B > \mu_{\text{SF}}$, leading to a new phase transition within the $U(1)_B$ superfluid phase (indicated in figure 8), where the gravitational SPT jumps from $\theta_g = 0$ to $\pi$.

### 6.3.2   New Phenomena Inside Neutron Stars

Neutron stars are cold, compact astrophysical objects whose description requires a detailed understanding of QCD at finite density and very low temperatures. Conversely, astrophysical observations of neutron stars can enhance our understanding of this challenging region of the QCD phase diagram. See for instance the recent reviews [160, 97].

It has long been an intriguing possibility that some neutron star cores may be sufficiently dense to contain inner regions of deconfined quark matter, surrounded by outer regions of confined hadronic matter, see e.g. [161–164] for very recent discussions informed by modern astrophysical observations, with references to the vast literature on this subject. As reviewed in [161], the quest to identify quark matter inside neutron stars in part hinges on our ability to identify sharp consequences of the putative quark-matter cores. We would like to briefly highlight one new such consequence, which follows from the fact that high-density QCD is a non-trivial SPT with $\theta_g = \pi$ (i.e. a topological superconductor), as was shown in section 6.2.

Crudely speaking, neutron stars explore the QCD phase diagram as a function of $\mu_B$, starting with large $\mu_B$ in the high-density core of the star and dropping to smaller $\mu_B$ in

the lower-density outer regions. Let us assume that there are neutron stars whose cores access the deconfined CFL Higgs phase at large $\mu_B$, which we now know to also support a $\theta_g = \pi$ SPT. As we march radially outward from the core, we will eventually encounter a phase transition at which the SPT jumps from $\theta_g = \pi$ in the interior to $\theta_g = 0$ in the outer region (see figure 9). The two phases are separated by a transition layer (the green region in figure 9), whose width and surface tension depend on the details of the phase transition.[99]

As we have argued in section 6.3.1, the SPT transition where $\theta_g$ jumps is plausibly distinct from the $U(1)_B$-breaking superfluid transition at hadronic densities; in figure 8, the SPT transition is the green dot on the right (at larger $\mu_B$), while the $U(1)_B$-breaking transition is the green dot on the left (at smaller $\mu_B$). If we assume that these transitions are first order, the resulting picture is similar to the one recently explored in [164], where the authors considered hybrid stars with a sequence of phases separated by multiple transition layers (or domain walls). Luckily, our comments below are very general, and hence robust. They only rely on the existence of a quark core with $\theta_g = \pi$, which is separated from an outer region with $\theta_g = 0$ by a transition layer. Essentially no further details are needed.

As explained in section 2, an SPT jump always implies anomalous edge modes at boundaries. A variant of this statement is that an SPT jump across a transition layer or domain wall always implies anomalous surface modes on the wall. In particular, in section 2.3.3 we recalled that a jump $\Delta\theta_g = \pi$ in the gravitational theta-angle leads to a non-trivial thermal Hall conductance on the domain wall; for T-symmetric walls it is

$$\kappa_{xy} = \frac{1}{4} + \frac{n}{2} , \qquad n \in \mathbb{Z} . \tag{6.31}$$

Here $\kappa_{xy}$ is measured in units of $\pi^2 k_B^2 T/3h$, where $k_B$ and $h$ are the Boltzmann and Planck constants, respectively, and $T$ is the temperature. Recall that the thermal Hall effect, whose magnitude is controlled by $\kappa_{xy}$, involves non-dissipative heat flow in the direction transverse to the temperature gradient.

The detailed nature of the anomalous surface modes is not uniquely fixed by $\kappa_{xy}$. As we reviewed in section 2.3.3, possible candidates include an odd number of massless 2+1d Majorana fermions or gapped, non-Abelian anyons (both with T-symmetry); another possibility is spontaneous T-breaking at the surface. It is clearly desirable to understand the extent to which these anomalous surface modes have observable implications for neutron stars. We leave this interesting question to future work.

---

[99] We continue to use a scheme in which the SPT in the outer confined regions (as well as outside the star) is trivial, but this is only for convenience. The SPT jump across the transition layer is scheme independent.

# Acknowledgements

We thank A. Cherman, C. Córdova, R. Jaffe, A. Kapustin, I. Klebanov, Z. Komargodski, M. Metlitski, R. Pisarski, K. Roumpedakis, D. Saltzberg, N. Seiberg, R. Thorngren, and A. Vishwanath for useful discussions. TD is supported by DOE Early Career Award DE-SC0020421. TD and P.-S.H. are supported by the Simons Collaboration on Global Categorical Symmetries. TD would like to thank the organizers and participants of the "Paths to Quantum Field Theory 2023" workshop at Durham University. The work of P.-S.H. was partially performed at the Aspen Center for Physics, which is supported by NSF grant PHY-2210452. P-S.H. also thanks the Nordic Institute for Theoretical Physics for hosting the workshop "Categorical aspects of symmetries," during which part of this work was performed.

# A    Dynkin Index for $SU(N)$ Representations

In this appendix we review the Dynkin index for general $SU(N)$ representations. For more details, see e.g. [165].

The Dynkin index $I(r)$ for an $SU(N)$ representation $r$ can be computed from

$$I(r) = \frac{\dim r}{2(N^2 - 1)} C_2(r) , \tag{A.1}$$

where $\dim r$ is the dimension of the representation $r$ and $C_2(r)$ is its quadratic Casimir. An explicit formula for the latter can be written down in terms of the Young tableau associated with the representation $r$, whose row lengths are $\{\ell_i\}_{i=1}^{N-1}$ with $\ell_1 \geq \ell_2 \geq \cdots \geq \ell_{N-1}$. Then

$$C_2(r) = N n_r - \frac{n_r^2}{N} + \sum_{i=1}^{N-1} \ell_i(\ell_i + 1 - 2i) , \tag{A.2}$$

where $n_r$ is the total number of boxes in the Young tableau.

Let us list some useful examples:

- The fundamental representation $\square$ has Young tableau $(\ell_i) = (1, 0, \cdots, 0)$, so that

$$I\left(\square\right) = \frac{N}{2(N^2 - 1)} \left(N - \frac{1}{N}\right) = \frac{1}{2} . \tag{A.3}$$

- The adjoint representation has Young tableau $(\ell_i) = (2, 1, \cdots, 1)$, which gives

$$I(\text{adjoint}) = N . \tag{A.4}$$

- The two-index antisymmetric representation has Young tableau $(\ell_i) = (1, 1, 0, 0 \cdots, 0)$,

$$I\left(\begin{array}{c}\square\\\square\end{array}\right) = \frac{N(N-1)/2}{2(N^2 - 1)} \left(2N - \frac{4}{N} - 2\right) = \frac{N-2}{2} . \tag{A.5}$$

- The two-index symmetric representation has Young tableau $\ell = (2, 0, \cdots, 0)$,

$$I\left(\square\square\right) = \frac{N(N+1)/2}{2(N^2 - 1)} \left(2N - \frac{4}{N} + 2\right) = \frac{N+2}{2} . \tag{A.6}$$

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
