# Peer review of "Higgs-Confinement Transitions in QCD from Symmetry Protected Topological Phases"

_SciPost Physics_

## Round 2 · Referee Report · Anonymous (Referee 1) · 2024-7-10

Strengths

  1. Highly innovative approach to the study of the QCD (and related QFTs) phase diagram, using previously un-noticed SPT information.

  2. Highly plausible new argument against a widely believed conjecture in the QCD literature, with possible interesting implications for neutron star physics.

Weaknesses

Only minor weaknesses.

  1. Paper is very long, making it difficult to digest.

  2. The arguments presented do not quite prove that there is a phase transition as a function of density. Having said that, they are very plausible.

Report

This paper points that gauge theories with fundamental-representation fermions, which can have Higgs and confining regimes, can move between different SPT phases as a function of parameters, and the Higgs and confining regimes can in fact be genuinely distinct phases of matter because they are distinct as SPTs. This is a very striking and surprising observation with wide-ranging applications. The paper is nicely written (although much longer than I personally would have preferred), and I think it will be a seminar reference in the field in the decades to come. SciPost should be proud to publish it.

Recommendation

Publish (surpasses expectations and criteria for this Journal; among top 10%)

---

## Round 2 · Referee Report · Anonymous (Referee 2) · 2024-8-8

Report

The paper under scrutiny addresses the problem of Higgs-confinement continuity in 4d gauge theories from the perspective of the novel topological approach to symmetries. It establishes the remarkable and original result that it is indeed possible to discriminate between a higgsed and a confined phase (in theories with trivial 1-form symmetry) when T-reversal is a preserved symmetry. The transition, that can be both first or second order, is generically beyond the Landau paradigm, and happens through the jump of a discrete SPT phase involving flavour and/or gravitational background fields. The consequences of this observation are considered for a model very close to real-world QCD, so that even applications to the physics of neutron stars are briefly contemplated.

The authors give a very detailed and convincing exposition of their results, which as already stated above, are very important and bring a new light to a problem that has been studied since the 70s. It is also interesting that the new topological approach to symmetries has been instrumental in achieving this new understanding.

I have a few remarks on some details that could improve the reading of the paper. However I have first of all a general remark on the presentation. The paper is very detailed, sometimes to the extent that it is a bit redundant. In other words, given the importance of the results, I would have expected a much sharper presentation. Many of the details are known to the experts, and can be easily rederived/found in relevant references. Surely, it can help many readers to have a nice exposition of the material needed for the results, but it hinders in my opinion the immediate transmission of the salient features of the paper. If I may add, the too many footnotes also make the reading very non-linear. Many of them can probably be eliminated without any loss of information, while others can be included in the main text. I am not requiring here that the authors dramatically alter the text, I am just expressing an opinion, the authors can then decide whether they agree or not, or even keep the remarks in mind for future papers.

Concerning specific points, I have the following remarks. The main one is about T-invariance. It is made abundantly clear how T-invariance is crucial for the phase transition to occur as an abrupt jump of the SPT. I would have appreciated to find a slightly longer discussion of what happens when T-invariance is broken, even in the simplest examples. For instance, in presence of a generic complex quark mass, it is said that then one can smoothly interpolate between the two values of the SPT. Should this be considered a crossover? More importantly, given also the amount of introductory material that is already present, I would have expected a more lengthy discussion of why would we expect T to be preserved, both in abstract and realistic models. For instance, there is no mention of T-invariance when discussing neutron stars. Another question I kept asking myself while reading is whether T-invariance might require tuning?

Lesser details are the following:

-the argument around (2.55) about the ’t Hooft anomaly is written in a confusing way. It seems one can choose to have an anomaly.

-the comment before (4.9) on breaking or not of $(-1)^F$ by $h$ is not formulated clearly, hence it is confusing. I am actually not sure I understand it. I agree that SU(2) QCD with 2 Weyl doublets is a bosonic theory, but I would say that the same theory to which we add a Higgs scalar doublet is no longer bosonic. Indeed there are fermionic gauge invariant composites, as emphasized in [24]. The element $-I$ of the SU(2) can no longer be identified with $(-1)^F$ since it also flips sign to $h$. This is of course a marginal issue, but I would like the authors to clarify this point.

-it is not clear if the gaps in (6.4) have to be compared to the quark masses, or not. If yes, then it seems that $\mu_B$ has to be hierarchically larger than $m_q$ in order for the gap to change the sign of the masses.

My recommendation is clearly to publish the paper. All my comments above can be taken as optional improvements. However I would be very happy if the authors indeed consider the suggested clarifications.

Recommendation

Publish (surpasses expectations and criteria for this Journal; among top 10%)

---

## Editorial Decision

resubmitted